# On the Generalization Properties of Diffusion Models

**Puheng Li***[*]
Department of Statistics
Stanford University
puhengli@stanford.edu

**Zhong Li***[†]
Machine Learning Group
Microsoft Research Asia
lzhong@microsoft.com

**Huishuai Zhang**[†]
Machine Learning Group
Microsoft Research Asia
huzhang@microsoft.com

**Jiang Bian**
Machine Learning Group
Microsoft Research Asia
jiabia@microsoft.com

## Abstract

Diffusion models are a class of generative models that serve to establish a stochastic transport map between an empirically observed, yet unknown, target distribution and a known prior. Despite their remarkable success in real-world applications, a theoretical understanding of their generalization capabilities remains underdeveloped. This work embarks on a comprehensive theoretical exploration of the generalization attributes of diffusion models. We establish theoretical estimates of the generalization gap that evolves in tandem with the training dynamics of score-based diffusion models, suggesting a polynomially small generalization error ($O(n^{-2/5} + m^{-4/5})$) on both the sample size $n$ and the model capacity $m$, evading the curse of dimensionality (i.e., not exponentially large in the data dimension) when *early-stopped*. Furthermore, we extend our quantitative analysis to a *data-dependent* scenario, wherein target distributions are portrayed as a succession of densities with progressively increasing distances between modes. This precisely elucidates the *adverse* effect of "*modes shift*" in ground truths on the model generalization. Moreover, these estimates are not solely theoretical constructs but have also been confirmed through numerical simulations. Our findings contribute to the rigorous understanding of diffusion models' generalization properties and provide insights that may guide practical applications.

## 1 Introduction

As an emerging family of deep generative models, diffusion models (DMs; [16, 45]) have experienced a surge in popularity, owing to their unparalleled performance in a wide range of applications ([22, 39, 68, 15, 3, 35, 4, 60, 33, 69, 64, 63, 53]). This has led to notable commercial successes, such as DALL·E ([34]), Imagen ([38]), and Stable Diffusion ([36]). Mathematically, diffusion models learn an unknown underlying distribution through a two-stage process: (i) first, successively and gradually injecting random noises (forward process); (ii) then reversing the forward process through denoising for sampling purposes (reverse process). To achieve this, an equivalent formulation of diffusion models called score-based generative models (SGMs; [50, 52]) is employed. SGMs implement the aforementioned two-stage process via the continuous dynamics represented by a joint group of coupled stochastic differential equations (SDEs) ([54, 20]).

---

[*]This work was done when Puheng Li was an undergraduate student at Peking University and a research intern at MSRA. The first two authors contributed equally to this work and are ordered alphabetically.
[†]Corresponding author.

37th Conference on Neural Information Processing Systems (NeurIPS 2023).

Despite their impressive empirical performance, the theoretical foundation of DMs/SGMs remains underexplored. Generally, fundamental theoretical questions can be categorized into several aspects. By considering machine learning models as mathematical function classes from certain spaces, one can identify three central aspects: approximation, optimization and generalization. At the forefront lies the generalization problem, which aims to characterize the learning error between the learned and ground truth distributions.

The development of generalization theory for diffusion models is pressing due to both theoretical and practical concerns:

- In theory, the generalization issues of generative modeling (or learning for distributions) may exhibit as the memorization phenomenon, if the modeled distribution is eventually trained to converge to the empirical distribution only associated with training samples. Intuitively, memorization arises from two reasons: (i) it is useful for the hypothesis space to be large enough to approximate highly complex underlying target distributions (universal convergence; [65]); (ii) the underlying distribution is unknown in practice, and one can only use a dataset with finite samples drawn from the target distribution. Rigorous mathematical characterizations of memorization are developed for bias potential models and GANs in [66] and [67], respectively. A natural question is, does a similar phenomenon occur for diffusion models? To answer this, a thorough investigation of generalization properties for DMs/SGMs is required.

- In practice, the generalization capability of diffusion models is also an essential requirement, as the memorization can lead to potential privacy and copyright risks when models are deployed. Similar to other generative models and large language models (LLMs) [5, 70, 19, 6], diffusion models can also memorize and leak training samples [5, 46], hence can be subsequently attacked using specific procedures and algorithms [28, 18, 62]. Although there are defense methods developed to meet privacy and copyright standards ([11, 14, 58]), these approaches are often heuristic, without providing sufficient quantitative understandings particularly on diffusion models. Therefore, a comprehensive investigation of the generalization foundation of diffusion models, including both theoretical and empirical aspects, is of utmost importance in improving principled tutorial guidance in practice.

The current work develops the generalization theory of diffusion models in a mathematically rigorous manner. Our main results include the following:

- We derive an upper bound of the generalization gap for diffusion models along the training dynamics. This result suggests, with *early-stopping*, the generalization error of diffusion models scales polynomially small on the sample size ($O(n^{-2/5})$) and the model capacity ($O(m^{-4/5})$). Notably, the generalization error also escapes from the curse of dimensionality.

- This "uniform" bound is further extended to a *data-dependent* setting, where a sequence of unidimensional Gaussian mixtures distributions with an increasing modes' distance is considered as the ground truth. This result characterizes the effect of "*modes shift*" quantitatively, which implies that the generalization capability of diffusion models is *adversely* affected by the distance between high-density regions of target distributions.

- The theoretical findings are numerically verified in simulations.

The rest of this paper is organized as follows. In Section 2, we discuss the related work on the convergence and training fronts of diffusion models and also the generalization aspects of other generative modeling methods. Section 3 is the central part, which includes the problem formulation, main results, and consequences. Section 4 includes numerical verifications on synthetic and real-world datasets[3]. All the details of proofs and experiments are found in the appendices.

## 2 Related Work

We review the related work on diffusion models concerning the central results in this paper.

- First, on the convergence theory, [7, 9] established elaborate error estimates between the modeled and target distribution given the discretization and time-dependent score matching

---

[3]Code is available at `https://github.com/lphLeo/Diffusion_Generalization`

$$\boldsymbol{x}(0) \xrightarrow{\hspace{2cm} d\boldsymbol{x} = \boldsymbol{f}(\boldsymbol{x},t)dt + g(t)d\boldsymbol{W}_t \hspace{2cm}} \boldsymbol{x}(T)$$

$$\boldsymbol{x}(0) \xleftarrow{\hspace{1cm} d\boldsymbol{x} = \left[\boldsymbol{f}(\boldsymbol{x},t) - g^2(t)\nabla_{\boldsymbol{x}}\log p_t(\boldsymbol{x})\right]dt + g(t)d\bar{\boldsymbol{W}}_t \hspace{1cm}} \boldsymbol{x}(T)$$

$$\approx s_{t,\boldsymbol{\theta}}(\boldsymbol{x}) := \frac{1}{m}\boldsymbol{A}\sigma(\boldsymbol{W}\boldsymbol{x} + \boldsymbol{U}e(t)), \boldsymbol{\theta} = \boldsymbol{A}$$

**Target**: Finitely-supported prob. & Gaussian mixtures

**Loss**: Time-dependent score matching (Eq. (7))

**Algorithm**: Gradient flow

**Notations**:

$t$: SDE time $\qquad$ $T$: maximal SDE time

$p_T \approx \pi$: a known prior

$\tau$: training time

Figure 1: Illustration of the problem formulation and important notations.

tolerance. Compared to the present work, they did not evolve the concrete training dynamics since the setting therein focuses on the properties of optimizers.

- Second, on the training front, [51] proposed a set of techniques to enhance the training performance of score-based generative models, scaling diffusion models to images of higher resolution, but without any characterization of possible generalization improvements. Similar to [7, 9], [49] also provided error estimates between the modeled and target distributions in a point-wise sense and again did not evolve the detailed training dynamics.

- Third, on the generalization and memorization side, corresponding theories are developed for bias potential models and GANs in [66] and [67], respectively, where the modeled distribution learns the ground truth with early-stopping and diverges or converges to the empirical distribution only associated with training samples after sufficiently long training time. The current work extends the mathematical analysis to the case of diffusion models under a data-dependent setting.

- As a supplement, we also discuss related literature regarding low-density learning. [50] illustrated the difficulty of learning from low-density regions with toy formulations and simulations, which motivates the sampling method of annealed Langevin dynamics as the predecessor of (score-based) diffusion models. [21] restudied similar problems under the pure score matching regime (without the denoising or time-dependent dynamics) and attributed the difficulty to increasing isoperimetry of distributions with modes shift. [41] defined the Hardness score and numerically justified that decreasing manifold densities leads to the increasing Hardness score, and consequently applied the Hardness score as the regularization to the sampling process to enhance synthetic images from low-density regions. As a comparison, this work establishes a mathematically rigorous estimate on the generalization gap that *quantitatively* depends on the range of low-density regions (between modes) in target distributions for (score-based) diffusion models that requires *denoising*.

## 3 Formulation and Results

In this section, we first introduce the problem setup. Next, we state the main theoretical results, subsequent consequences, and possible connections. The numerical illustration is provided at last.

### 3.1 Problem Formulation

Since DMs/SGMs have already grown into a large family of generative models with an enormous number of variants, there are various ways to define the parameterization of diffusion models. Here, we adopt (one of) the most fundamental architectures proposed in [54], where the forward perturbation and reverse sampling process are both implemented by a joint group of coupled (stochastic) differential equations. See Figure 1 for an illustration of the problem formulation.

**Forward perturbation.** We start with the setting of unsupervised learning. Given an unlabeled dataset $\mathcal{D}_{\boldsymbol{x}} = \{\boldsymbol{x}_i\}_{i=1}^n \subset \mathbb{R}^d$ with the sample $\boldsymbol{x}_i \overset{\text{i.i.d.}}{\sim} p_0(\boldsymbol{x})$, where $p_0$ denotes the underlying (ground truth or target) distribution, the forward diffusion process is defined as

$$d\boldsymbol{x} = \boldsymbol{f}(\boldsymbol{x}, t)dt + g(t)d\boldsymbol{W}_t, \quad \boldsymbol{x}(0) \sim p_0. \tag{1}$$

Here, the drift coefficient $\boldsymbol{f}(\cdot, t) : \mathbb{R}^d \mapsto \mathbb{R}^d$ is a time-dependent vector-valued function, and the diffusion coefficient $g(\cdot) : \mathbb{R}_{\geq 0} \mapsto \mathbb{R}$ is a scalar function, and $\boldsymbol{W}_t$ denotes the standard Wiener process (a.k.a., Brownian motion). The SDE (1) has a unique, strong solution under certain regularity conditions (i.e., globally Lipschitz coefficients in both state and time; see [31]). From now on, we denote by $p_t(\boldsymbol{x}(t))$ the marginal distribution of $\boldsymbol{x}(t)$, and let $p_{t|s}(\boldsymbol{x}(t)|\boldsymbol{x}(s))$ be the (perturbation) transition kernel from $\boldsymbol{x}(s)$ to $\boldsymbol{x}(t)$, $0 \leq s < t \leq T < \infty$ with $T$ as the time horizon. By appropriately selecting $\boldsymbol{f}$ and $g$, one can force the SDE (1) to converge to a prior distribution (typically a Gaussian). Common examples include the (time-rescaled) Ornstein–Uhlenbeck (OU) process,[4] which is a special case of the linear version of (1):

$$d\boldsymbol{x} = f(t)\boldsymbol{x}dt + g(t)d\boldsymbol{W}_t, \quad \boldsymbol{x}(0) \sim p_0. \tag{2}$$

**Reverse sampling.** According to [1] and [54], both the following reverse-time SDE and probability flow ODE share the same marginal distribution as the forward-time SDE (1):

$$d\boldsymbol{x} = \Big[\boldsymbol{f}(\boldsymbol{x}, t) - g^2(t)\nabla_{\boldsymbol{x}} \log p_t(\boldsymbol{x})\Big]dt + g(t)d\bar{\boldsymbol{W}}_t, \tag{3}$$

$$d\boldsymbol{x} = \Big[\boldsymbol{f}(\boldsymbol{x}, t) - \frac{1}{2}g^2(t)\nabla_{\boldsymbol{x}} \log p_t(\boldsymbol{x})\Big]dt, \tag{4}$$

where $\bar{\boldsymbol{W}}_t$ is a standard Wiener process when time flows backwards from $T$ to $0$, and $dt$ is an infinitesimal negative time step. With the initial condition $\boldsymbol{x}(T) \sim p_T \approx \pi$, where $\pi$ is a known prior distribution such as the Gaussian noise, one can (numerically) solve (3) or (4) to transform noises into samples from $p_0$, which is exactly the goal of generative modeling.

**Loss objectives.** The only remaining task is to estimate the unknown (Stein) score function $\nabla_{\boldsymbol{x}} \log p_t(\boldsymbol{x})$. This is achieved by minimizing the following weighted sum of denoising score matching ([57]) objectives:

$$\mathcal{L}(\boldsymbol{\theta}; \lambda(\cdot)) := \mathbb{E}_{t \sim \mathcal{U}(0,T)} \Big[\lambda(t) \cdot \mathbb{E}_{\boldsymbol{x}(0) \sim p_0} \big[\mathbb{E}_{\boldsymbol{x}(t) \sim p_{t|0}} \big[\|\boldsymbol{s}_{t,\boldsymbol{\theta}}(\boldsymbol{x}(t)) - \nabla_{\boldsymbol{x}(t)} \log p_{t|0}(\boldsymbol{x}(t)|\boldsymbol{x}(0))\|_2^2\big]\big]\Big], \tag{5}$$

with $\boldsymbol{\theta}^* := \arg\min_{\boldsymbol{\theta}} \mathcal{L}(\boldsymbol{\theta}; \lambda(\cdot))$, where $\mathcal{U}(0,T)$ denotes the uniform distribution over $[0,T]$, and $\lambda(t) : [0,T] \mapsto \mathbb{R}_+$ is a weighting function, which is typically selected as

$$\lambda(t) \propto 1/\sqrt{\mathbb{E}_{\boldsymbol{x}(t) \sim p_{t|0}}[\|\nabla_{\boldsymbol{x}(t)} \log p_{t|0}(\boldsymbol{x}(t)|\boldsymbol{x}(0))\|_2^2]} \tag{6}$$

according to ([54]). The score function $\boldsymbol{s}_{t,\boldsymbol{\theta}} : \mathbb{R}^d \mapsto \mathbb{R}^d$ is time-dependent and can be parameterized as a neural network (encoded with the time information) such as the U-net([37]) architecture commonly applied in the field of image segmentation. Alternatively, one can also define the time-dependent score matching loss

$$\tilde{\mathcal{L}}(\boldsymbol{\theta}; \lambda(\cdot)) := \mathbb{E}_{t \sim \mathcal{U}(0,T)} \big[\lambda(t) \cdot \mathbb{E}_{\boldsymbol{x}(t) \sim p_t} \big[\|\boldsymbol{s}_{t,\boldsymbol{\theta}}(\boldsymbol{x}(t)) - \nabla_{\boldsymbol{x}(t)} \log p_t(\boldsymbol{x}(t))\|_2^2\big]\big], \tag{7}$$

which is equivalent to (5) up to a constant independent of $\boldsymbol{\theta}$ by [57, 49].

In practice, expectations in the objective (5) can be respectively estimated with empirical means over time steps in $[0,T]$, data samples from $p_0$ and $p_{t|0}$, which is efficient when the drift coefficient $\boldsymbol{f}(\cdot, t)$ is linear. Specifically, if the forward-time SDE takes the form of (2), the transition kernel $p_{t|0}$ has a closed form ([40])

$$p_{t|0}(\boldsymbol{x}(t)|\boldsymbol{x}(0)) = \mathcal{N}(\boldsymbol{x}(t); r(t)\boldsymbol{x}(0), r^2(t)v^2(t)\boldsymbol{I}_d), \tag{8}$$

where $\mathcal{N}(\boldsymbol{x}; \boldsymbol{\mu}, \boldsymbol{\Sigma})$ denotes the multivariate Gaussian distribution evaluated at $\boldsymbol{x}$ with the expectation $\boldsymbol{\mu}$ and covariance $\boldsymbol{\Sigma}$, and $r(t) := e^{\int_0^t f(\zeta)d\zeta}$, $v(t) := \sqrt{\int_0^t \frac{g^2(\zeta)}{r^2(\zeta)}d\zeta}$.

---

[4]The OU process is the unique time-homogeneous Markov process which is also a Gaussian process, with the stationary distribution as the standard Gaussian distribution.

**Training.** We aim to investigate the gradient flow training dynamics over the empirical loss

$$\frac{d}{d\tau}\hat{\boldsymbol{\theta}}_n(\tau) = -\nabla_{\hat{\boldsymbol{\theta}}_n(\tau)}\hat{\mathcal{L}}_n(\hat{\boldsymbol{\theta}}_n(\tau);\lambda(\cdot)), \quad \hat{\boldsymbol{\theta}}_n(0) := \hat{\boldsymbol{\theta}}_n^0, \tag{9}$$

where $\hat{\mathcal{L}}_n$ is the Monte-Carlo estimation of $\mathcal{L}$ defined in (5) on the training dataset,[5] with an auxiliary gradient flow over the population loss

$$\frac{d}{d\tau}\boldsymbol{\theta}(\tau) = -\nabla_{\boldsymbol{\theta}(\tau)}\mathcal{L}(\boldsymbol{\theta}(\tau);\lambda(\cdot)), \quad \boldsymbol{\theta}(0) := \boldsymbol{\theta}^0 = \hat{\boldsymbol{\theta}}_n^0. \tag{10}$$

In both cases, the weighting function $\lambda(\cdot)$ is selected as in (6). Denote the score function learned at the training time $\tau$ evaluated at the SDE time $t$ with respect to the empirical loss and population loss as $\boldsymbol{s}_{t,\hat{\boldsymbol{\theta}}_n(\tau)}(\boldsymbol{x}(t))$ and $\boldsymbol{s}_{t,\boldsymbol{\theta}(\tau)}(\boldsymbol{x}(t))$, respectively. The corresponding density functions, denoted by $p_{t,\hat{\boldsymbol{\theta}}_n(\tau)}(\boldsymbol{x}(t))$ and $p_{t,\boldsymbol{\theta}(\tau)}(\boldsymbol{x}(t))$, are obtained by solving

$$\nabla_{\boldsymbol{x}(t)}\log p_{t,\hat{\boldsymbol{\theta}}_n(\tau)}(\boldsymbol{x}(t)) = \boldsymbol{s}_{t,\hat{\boldsymbol{\theta}}_n(\tau)}(\boldsymbol{x}(t)), \quad \nabla_{\boldsymbol{x}(t)}\log p_{t,\boldsymbol{\theta}(\tau)}(\boldsymbol{x}(t)) = \boldsymbol{s}_{t,\boldsymbol{\theta}(\tau)}(\boldsymbol{x}(t)), \tag{11}$$

and then normalizing, respectively.

**Score networks.** We parameterize the score function $\boldsymbol{s}_{t,\boldsymbol{\theta}}$ as the following random feature model

$$\boldsymbol{s}_{t,\boldsymbol{\theta}}(\boldsymbol{x}) := \frac{1}{m}\boldsymbol{A}\sigma(\boldsymbol{W}\boldsymbol{x} + \boldsymbol{U}\boldsymbol{e}(t)) = \frac{1}{m}\sum_{i=1}^{m}\boldsymbol{a}_i\sigma(\boldsymbol{w}_i^\top\boldsymbol{x} + \boldsymbol{u}_i^\top\boldsymbol{e}(t)), \tag{12}$$

where $\sigma$ is the ReLU activation function, $\boldsymbol{A} = (\boldsymbol{a}_1, \ldots, \boldsymbol{a}_m) \in \mathbb{R}^{d \times m}$ is the *trainable* parameter, while $\boldsymbol{W} = (\boldsymbol{w}_1, \ldots, \boldsymbol{w}_m)^\top \in \mathbb{R}^{m \times d}$ and $\boldsymbol{U} = (\boldsymbol{u}_1, \ldots, \boldsymbol{u}_m)^\top \in \mathbb{R}^{m \times d_e}$ are randomly initialized and *frozen* during training, and $\boldsymbol{e} : \mathbb{R}_{\geq 0} \mapsto \mathbb{R}^{d_e}$ is the embedding function concerning the time information. Assume that $\boldsymbol{a}_i$, $\boldsymbol{w}_i$ and $\boldsymbol{u}_i$ are i.i.d. sampled from an underlying distribution $\rho$. Then, as $m \to \infty$, we get

$$\boldsymbol{s}_{t,\boldsymbol{\theta}}(\boldsymbol{x}) \to \bar{\boldsymbol{s}}_{t,\bar{\boldsymbol{\theta}}}(\boldsymbol{x}) := \mathbb{E}_{(\boldsymbol{a},\boldsymbol{w},\boldsymbol{u})\sim\rho}\left[\boldsymbol{a}\sigma(\boldsymbol{w}^\top\boldsymbol{x} + \boldsymbol{u}^\top\boldsymbol{e}(t))\right]$$

$$= \mathbb{E}_{(\boldsymbol{w},\boldsymbol{u})\sim\rho_0}\left[\boldsymbol{a}(\boldsymbol{w},\boldsymbol{u})\sigma(\boldsymbol{w}^\top\boldsymbol{x} + \boldsymbol{u}^\top\boldsymbol{e}(t))\right], \tag{13}$$

with $\boldsymbol{a}(\boldsymbol{w},\boldsymbol{u}) := \frac{1}{\rho_0(\boldsymbol{w},\boldsymbol{u})}\int_{\mathbb{R}^d}\boldsymbol{a}\rho(\boldsymbol{a},\boldsymbol{w},\boldsymbol{u})d\boldsymbol{a}$ and $\rho_0(\boldsymbol{w},\boldsymbol{u}) := \int_{\mathbb{R}^d}\rho(\boldsymbol{a},\boldsymbol{w},\boldsymbol{u})d\boldsymbol{a}$. By the positive homogeneity property of the ReLU activation, we can assume that $\|\boldsymbol{w}\|_1 + \|\boldsymbol{u}\|_1 \leq 1$ w.l.o.g.

One can view $\bar{\boldsymbol{s}}_{t,\bar{\boldsymbol{\theta}}}(\boldsymbol{x})$ as a continuous version of the random feature model. Correspondingly, the optimal solution is denoted as $\bar{\boldsymbol{\theta}}^*$ when replacing the parameterized score function $\boldsymbol{s}_{t,\boldsymbol{\theta}}(\boldsymbol{x})$ in the loss objective (5) or (7) by $\bar{\boldsymbol{s}}_{t,\bar{\boldsymbol{\theta}}}(\boldsymbol{x})$. Define the kernel $k_{\rho_0}(\boldsymbol{x},\boldsymbol{x}') := \mathbb{E}_{(\boldsymbol{w},\boldsymbol{u})\sim\rho_0}\left[\sigma(\boldsymbol{w}^\top\boldsymbol{x} + \boldsymbol{u}^\top\boldsymbol{e}(t))\sigma(\boldsymbol{w}^\top\boldsymbol{x}' + \boldsymbol{u}^\top\boldsymbol{e}(t))\right]$, and let $\mathcal{H}_{k_{\rho_0}}$ be the induced reproducing kernel Hilbert space (RKHS; [2]), we have $\bar{\boldsymbol{s}}_{t,\bar{\boldsymbol{\theta}}} \in \mathcal{H}_{k_{\rho_0}}$ if the RKHS norm $\left\|\bar{\boldsymbol{s}}_{t,\bar{\boldsymbol{\theta}}}\right\|_{\mathcal{H}_{k_{\rho_0}}}^2 := \mathbb{E}_{(\boldsymbol{w},\boldsymbol{u})\sim\rho_0}[\|\boldsymbol{a}(\boldsymbol{w},\boldsymbol{u})\|_2^2] = \|\|\boldsymbol{a}\|_2\|_{L^2(\rho_0)}^2 < \infty$, and the corresponding discrete version can be defined by the empirical average, i.e., $\|\boldsymbol{s}_{t,\boldsymbol{\theta}}\|_{\mathcal{H}_{k_{\rho_0}}}^2 := \frac{1}{m}\|\boldsymbol{A}\|_F^2 = \frac{1}{m}\sum_{i=1}^{m}\|\boldsymbol{a}(\boldsymbol{w}_i,\boldsymbol{u}_i)\|_2^2$.

**Remark 1.** *There are more modern and complex mathematical tools such as neural tangent kernels (NTKs) and mean fields that can be selected as the score networks. Employing these modern tools is valuable at least for theoretical completeness and we leave these as the future work.*[6]

The goal is to measure and bound the generalization error evolving with the gradient flow training dynamics (9) between the learned distribution and target distribution, using the common Kullback–Leibler (KL) divergence.

**Definition 1** (KL divergence). *Given two distributions $p$ and $q$, the KL divergence from $q$ to $p$ is defined as $D_{\mathrm{KL}}(p\|q) = \int_{\mathbb{R}^d} p(\boldsymbol{x})\log\left(\frac{p(\boldsymbol{x})}{q(\boldsymbol{x})}\right)d\boldsymbol{x}$.*

Based on the above definitions, the generalization gap along the gradient flow training dynamics (9) is formulated as $D_{\mathrm{KL}}\left(p_0\|p_{0,\hat{\boldsymbol{\theta}}_n(\tau)}\right)$, which is only a function of the training time $\tau$. The goal is to estimate $D_{\mathrm{KL}}\left(p_0\|p_{0,\hat{\boldsymbol{\theta}}_n(\tau)}\right)$.

---

[5] Recall the dataset $\mathcal{D}_{\boldsymbol{x}} = \{\boldsymbol{x}_i\}_{i=1}^n \subset \mathbb{R}^d$, and we take $\mathcal{D}_t = \{t_i\}_{i=1}^n \subset [0, T]$ for convenience.

[6] For example, the extension to NTKs is possible, since the training of random feature models follows a specific NTK regime (only the last layer is updated) in the output space (instead of the parameter space), which can be properly analyzed in the infinite-width regime.

## 3.2 Main Results

In this section, we state the main results of the generalization capability of the (score-based) diffusion models and how it evolves as the training proceeds. Based on the formulation in Section 3.1, we theoretically derive several upper bounds to estimate $D_{\mathrm{KL}}\left(p_0\|p_{0,\hat{\boldsymbol{\theta}}_n(\tau)}\right)$ under different settings. The results cover both the positive and negative aspects: in the data-independent setting where the target distribution has finite support, the generalization gap is proved to be small; while in the data-dependent setting where the target distribution possesses shift modes, the generalization is adversely affected by the modes' distance.

### 3.2.1 Data-Independent Generalization Gap

In this section, we provide the characterization of the generalization capability for diffusion models given a target distribution defined on a finite domain. Generally, the KL divergence from the learned distribution $p_{0,\hat{\boldsymbol{\theta}}_n(\tau)}$ at the training time $\tau$ to the target distribution $p_0$ can be estimated as follows.

**Theorem 1.** *Suppose that the target distribution $p_0$ is continuously differentiable and has a compact support set, i.e., $\|\boldsymbol{x}\|_\infty$ is uniformly bounded, and there exists a reproducing kernel Hilbert space (RKHS) $\mathcal{H}$ ($:=\mathcal{H}_{k_{\rho_0}}$) such that $\bar{\boldsymbol{s}}_{0,\bar{\boldsymbol{\theta}}^*} \in \mathcal{H}$. Assume that the initial loss, trainable parameters, the embedding function $\boldsymbol{e}(t)$ and weighting function $\lambda(t)$ are all bounded. Then for any $\delta > 0$, $\delta \ll 1$, with the probability of at least $1 - \delta$, we have*

$$D_{\mathrm{KL}}\left(p_0\|p_{0,\hat{\boldsymbol{\theta}}_n(\tau)}\right) \lesssim \left[\frac{\tau^4}{mn} + \frac{\tau^3}{m^2} + \frac{1}{\tau}\right] + \left[\frac{1}{m} + \bar{\tilde{\mathcal{L}}}\left(\bar{\boldsymbol{\theta}}^*\right) + \tilde{\mathcal{L}}\left(\boldsymbol{\theta}^*\right)\right] + D_{\mathrm{KL}}\left(p_T\|\pi\right), \quad \tau \geq 1,$$

*where $\lesssim$ hides the term $d\log(d+1)$, the polynomials of $\log(1/\delta^2)$, finite RKHS norms and universal positive constants only depending on $T$.*

**Remark 2.** *Since $p_0$ is compactly supported, the target score function $\boldsymbol{s}_0(\boldsymbol{x}) = \nabla_{\boldsymbol{x}} \log p_0(\boldsymbol{x})$ is also defined on a compact domain. According to [12, 13], $\boldsymbol{s}_0$ is contained in the Barron function space with a finite Barron norm, and hence in a certain RKHS with a finite RKHS norm. Therefore, it is reasonable to require that the global minimizer $\boldsymbol{s}_{0,\boldsymbol{\theta}^*}$ or $\bar{\boldsymbol{s}}_{0,\bar{\boldsymbol{\theta}}^*}$ is also contained in some RKHS.*

**Proof sketch.** Theorem 1 is proved via the following procedure.

1. According to Theorem 1 in [49], the KL divergence on the left-hand side can be upper bounded by the population loss of the trained model up to a small error. That is,

$$D_{\mathrm{KL}}\left(p_0\|p_{0,\hat{\boldsymbol{\theta}}_n(\tau)}\right) \leq \tilde{\mathcal{L}}(\hat{\boldsymbol{\theta}}_n(\tau); g^2(\cdot)) + D_{\mathrm{KL}}\left(p_T\|\pi\right). \tag{14}$$

2. We use the model trained with respect to the population loss (10) to perform the decomposition:

$$\tilde{\mathcal{L}}(\hat{\boldsymbol{\theta}}_n(\tau)) = \left[\tilde{\mathcal{L}}(\hat{\boldsymbol{\theta}}_n(\tau)) - \tilde{\mathcal{L}}(\boldsymbol{\theta}(\tau))\right] + \tilde{\mathcal{L}}(\boldsymbol{\theta}(\tau))$$
$$\lesssim \left[\tilde{\mathcal{L}}(\hat{\boldsymbol{\theta}}_n(\tau)) - \tilde{\mathcal{L}}(\boldsymbol{\theta}(\tau))\right] + \bar{\tilde{\mathcal{L}}}(\bar{\boldsymbol{\theta}}(\tau)) + \text{Monte Carlo} \triangleq I_1 + I_2 + I_3, \tag{15}$$

where we omit the weighting function $g^2(\cdot)$ for simplicity. Here, $\bar{\tilde{\mathcal{L}}}$ is the loss objective obtained by replacing $\boldsymbol{s}_{t,\boldsymbol{\theta}}$ in $\tilde{\mathcal{L}}$ (defined in (7)) by $\bar{\boldsymbol{s}}_{t,\bar{\boldsymbol{\theta}}}$, and $I_3$ summarizes the resulting Monte Carlo error.

3. $I_3$ can be estimated via a similar argument as in [24] (Lemma 48).

4. $I_2$ can be upper bounded via a standard analysis on the gradient flow dynamics over convex objectives.

5. $I_1$ can be reduced as the norm product of $\boldsymbol{s}_{0,\hat{\boldsymbol{\theta}}_n(\tau)}$, $\boldsymbol{s}_{0,\boldsymbol{\theta}(\tau)}$ and their gap, then

   (a) the former can be bounded with a square-root rate growth via a general norm estimate of parameters trained under the gradient flow dynamics;

   (b) the latter can be estimated by the Rademacher complexity (see e.g., Chapter 26 in [43]).

Combining all above gives the desired result. The detailed proof is found in Appendix A.1.

**Discussion on error bounds.** The three error terms are further analyzed as follows.

- The first term is the main error, which implies an *early-stopping* generalization gap. In fact, if one selects an early-stopping time $\tau_{es}$ as $\tau_{es} = \Theta\left(n^{\frac{2}{5}}\right)$, and let $m \sim n$, we have

$$D_{KL}\left(p_0 \| p_{0,\hat{\boldsymbol{\theta}}_n(\tau_{es})}\right) \lesssim (1/n)^{\frac{2}{5}} + (1/m)^{\frac{4}{5}} . \tag{16}$$

- The second term is $m$-dependent and corresponds to the approximation error, which is $o(1)$ when $m \gg 1$. In fact, the random feature model is a universal approximator to Lipschitz continuous functions on a compact domain (Theorem 6 in [17]). See more details in Appendix A.1 (the last paragraph).

- The third term is exponentially small in $T$ since $\pi$ (e.g. the Gaussian density) is log-Sobolev, according to a classical result in e.g. [55] (Theorem 3.20, Theorem 3.24 and Remark 3.26).

**Remark 3.** *In practice, it is common to use the test error to evaluate the generalization performance. For diffusion models, a straightforward approach is to compute the negative log-likelihood (averaged in bits/dim; equivalent to the KL divergence) on the test dataset during training with the instantaneous change-of-variable formula ([8]) and probability flow ODE (defined in (4)), where the true score function $\nabla_{\boldsymbol{x}} \log p_t(\boldsymbol{x})$ is replaced by $\boldsymbol{s}_{t,\boldsymbol{\theta}(\tau)}(\boldsymbol{x})$.*

**Remark 4.** *Previous literature has established similar bounds for bias potential models ([66]) and GANs ([67]). Theorem 1 extends the corresponding results to the setting of diffusion models. Furthermore, this upper bound is finer in the sense that it incorporates the information regarding the model capacity (the hidden dimension $m$ in this case), which shows that more parameters benefit the learning and generalization, as expected.*

### 3.2.2 Data-Dependent Generalization Gap

In Section 3.2.1, we derive estimates on the generalization error for diffusion models along the training dynamics, where the target distribution is assumed to be finitely supported. In reality, this is often not the case, where target distributions usually possess distant multi-modes, from simple Gaussian mixtures to complicated Boltzmann distributions of physical systems ([29, 30]). Under these settings, the above analysis in Section 3.2.1 can not directly apply since the data domain is unbounded. It remains a problem to quantitatively characterize the generalization behavior of diffusion models given these target distributions with distant multi-modes or modes shift.

To provide a fine-grained demonstration of the generalization capability of diffusion models when applied to learn distributions with distant multi-modes, as an illustrating example, the Gaussian mixture with two modes is selected as the target distribution.

**Theorem 2.** *Suppose the target distribution $p_0$ is a one-dimensional 2-mode Gaussian mixture: $p_0(x) = q_1 \mathcal{N}(x; -\mu, 1) + q_2 \mathcal{N}(x; \mu, 1)$, where $\mu > \sqrt{\log(1/\delta^2)}$, $q_1$, $q_2 > 0$ with $q_1 + q_2 = 1$ are all constants. Under the conditions of Theorem 1 (except the uniform boundness of inputs), we have*

$$D_{KL}\left(p_0 \| p_{0,\hat{\boldsymbol{\theta}}_n(\tau)}\right) \lesssim \text{Poly}(\mu)\left[\frac{\tau^4}{mn} + \frac{\tau^3}{m^2}\right] + \frac{1}{\tau} + \left[\frac{\mu^2}{m} + \bar{\tilde{\mathcal{L}}}\left(\bar{\boldsymbol{\theta}}^*\right) + \tilde{\mathcal{L}}\left(\boldsymbol{\theta}^*\right)\right] + D_{KL}\left(p_T \| \pi\right),$$

*where $\tau \geq 1$, $\lesssim$ hides the polynomials of $\log(1/\delta^2)$, finite RKHS norms and universal positive constants only depending on $T$.*

**Proof sketch.** Theorem 2 is proved following a similar procedure with Theorem 1, except the input data $x$ does not have a uniform bound here. This problem mainly affects the last step (5 (b)) in the proof sketch of Theorem 1, and can be handled by using the fact

$$|x| \in [\mu - \sqrt{\log(1/\delta^2)}, \mu + \sqrt{\log(1/\delta^2)}] = \Theta(\mu) \tag{17}$$

given the target Gaussian mixture distribution. The detailed proof is found in Appendix A.2.

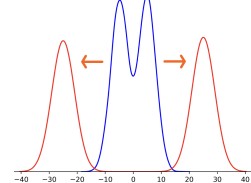

Figure 2: An illustration of modes shift.

**Remark 5.** *Theorem 2 indicates that, even for a simple target distribution (e.g. a one-dimensional 2-mode Gaussian mixture), the generalization error of diffusion models can be polynomially large regarding the modes' distance. Although Theorem 2 provides only an upper bound, the modes shift effect holds due to the model-target inconsistency (the last paragraph in Appendix A.2) and the following consistent experiments (Section 4.1.2).*

# 4 Numerical Verifications

In this section, we numerically verify the previous theoretical results and insights (early-stopping generalization and modes shift effect) on both synthetic datasets and real-world datasets.

## 4.1 Simulations on Synthetic Datasets

### 4.1.1 Early-Stopping Generalization

First, we illustrate the early-stopping generalization gap established in Theorem 1. We select the one-hidden-layer neural network with Swish activations as the score network, which is trained using the SGD optimizer with a fixed learning rate $0.5$. The target distribution is set to be a one-dimensional 2-mode Gaussian mixture with the modes' distance equalling 6, and the number of data samples is 1000. We measure the KL divergence from the trained model to the target distribution $D_{\mathrm{KL}}\left(p_0 \| p_{0,\hat{\boldsymbol{\theta}}_n(\tau)}\right)$ along with the training epochs.

From Figure 3, one can observe that the KL divergence achieves its minimum at approximately the 800-th training epoch, and it starts to increase after this turning point. The experimental results are consistent with Theorem 1 (over multiple runs), which states that there exist early-stopping times when diffusion models can generalize well, indicating the effectiveness of our upper bound. Further, the KL divergence begins to oscillate after the minimum point, which may suggest a phase transition in the training dynamics, and the transition point is around the (optimal) early-stopping time.

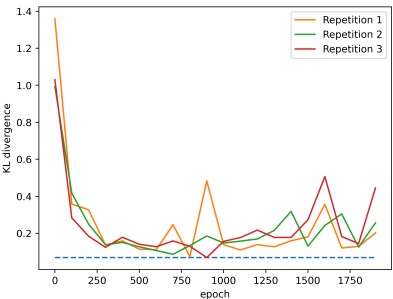

Figure 3: The KL divergence dynamics.

### 4.1.2 Modes Shift Effect

Next, we numerically test the relationship between the modes' distance and generalization (density estimation) performance. All the configurations remain the same as Section 4.1.1, except that the target Gaussian mixtures have different modes' distances.

In Figure 4, the modeled distributions exhibit the following two-stage dynamics: (i) first gradually fitting the two modes (epoch $= 100 \to 1000$); (ii) then diverging (epoch $= 1000 \to 1900$). This aligns with the KL divergence dynamics (Figure 3) and again verifies the corresponding theoretical results (Theorem 1). However, Figure 5 shows that when the modes are distant from each other, there is *difficulty* in the learning process. In Figure 5, the optimal generalization is achieved at epoch $= 100$, but is still far from well generalizing. As the training proceeds, the learned model is almost always a single-mode distribution (epoch $= 1000, 1900$). This phenomenon is aligned with the results established in Theorem 2, which states that when there are distant modes in the target distribution, the generalization performance is relatively poor.

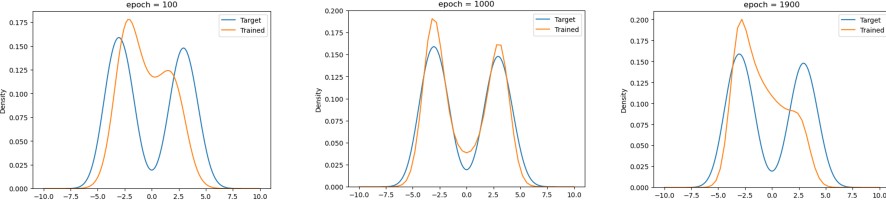

Figure 4: The training dynamics when the distance between two modes is 6 ($\mu = 3$).

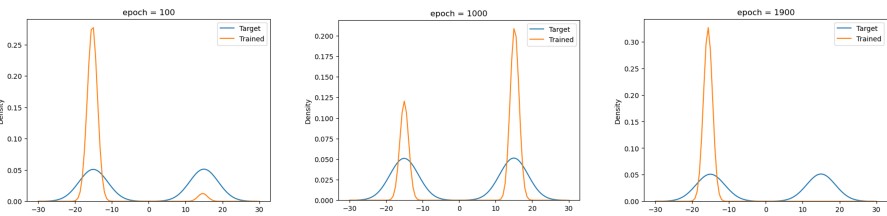

Figure 5: The training dynamics when the distance between two modes is 30 ($\mu = 15$).

## 4.2 Simulations on Real-World Datasets

In this subsection, we verify our results on the MNIST dataset using the standard U-net architecture as the score network, which suggests that the adverse effect of modes shift on the generalization performance of diffusion models also appears *in general*.

The setup is as follows. First, we perform a $K$-means clustering on $\mathcal{D}$ ($\mathcal{D}$ denote the MNIST dataset) to get $\mathcal{D} = \bigcup_{k=1}^{K} \mathcal{D}_k$, and $\bar{\boldsymbol{x}}_k$ as the center of $\mathcal{D}_k$, $k = 1, 2, \cdots, K$. Let $(i^*, j^*) := \arg\max_{i \neq j} \|\bar{\boldsymbol{x}}_i - \bar{\boldsymbol{x}}_j\|_2$, and $\mathcal{D}_{\text{farthest}} := \mathcal{D}_{i^*} \bigcup \mathcal{D}_{j^*}$. $\mathcal{D}_{\text{nearest}}$ is similarly constructed by $\arg\min$ indices. Then, by randomly selecting the same number of data samples and using the same configuration, we train two separate diffusion models on $\mathcal{D}_{\text{farthest}}$ and $\mathcal{D}_{\text{nearest}}$, respectively, and then perform inference (sampling). The training loss curves and sampling results are shown in Figure 6 and Figure 7, respectively. One can observe a significant performance gap: the diffusion model trained on $\mathcal{D}_{\text{farthest}}$ appears a higher learning loss and worse sampling quality compared to those of $\mathcal{D}_{\text{nearest}}$.

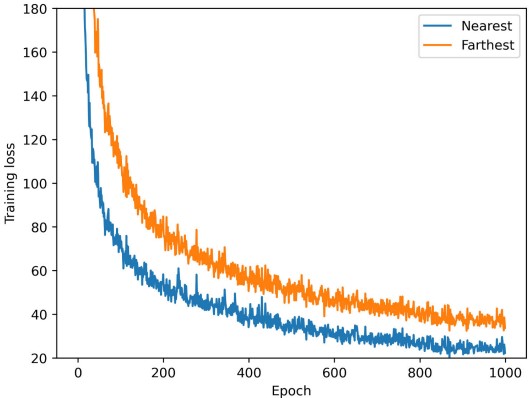

Figure 6: The training loss dynamics.

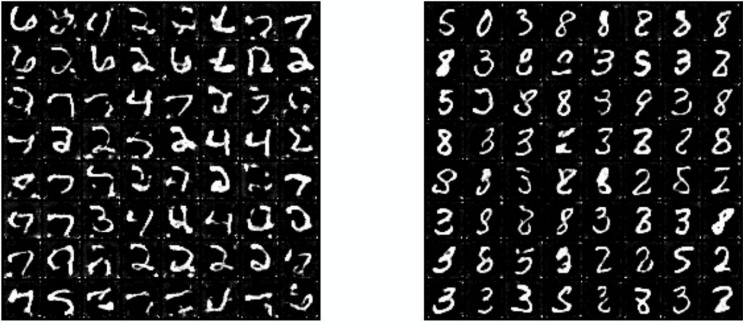

Figure 7: Sampling of the farthest (left) and nearest (right) clusters.

### 4.3 Discussion

We compare the results developed in this work with former corresponding literature as follows:

- The previous work [21] also studied the adverse effect of modes shift, which particularly reported a contrastive simulation indicating the degraded performance when modeling Gaussian mixtures with the increasing distance between modes (see Figure 2 in [21] and compare with Figure 4 and Figure 5). However, the results therein are established and tested under the "pure" score matching setting, without the denoising or time-dependent dynamics. As a comparison, Theorem 2 establishes a theoretical estimate on the generalization gap for diffusion models that requires *denoising*, and this upper bound *directly* depends on the distance between modes of target distributions, instead of a circuitous characterization in [21] to attribute the difficulty of learning modes shift to increased isoperimetry of corresponding target distributions.

- Similar adverse effect of modes shift has also been theoretically analyzed and numerically verified on recurrent neural networks (RNNs), see e.g., [23, 24]. There, the modes shift is understood as a type of long-term memory. This is the phenomenon of the "curse of memory": When there is long-term memory in the target, it requires a large number of parameters for the approximation. Meanwhile, the training process will suffer from severe slowdowns. Both of these effects can be exponentially more pronounced with increasing memory or modes shift.

- The very recent work [42] also considered the problem of learning Gaussian mixtures using the denoising diffusion probabilistic model (DDPM) objective, but under a *teacher-student* setting. That is, given the Gaussian mixtures target, [42] parametrized the score network model in the same form of the score function target, with the goal to identify true parameters. Consequently, there are all positive convergence results developed in [42], despite that the time and sample complexity increases with the distance between modes. As a comparison, Theorem 2 adopts a pre-selected score network model without incorporating any information from the ground truth, and hence establish negative results concerning the modes shift. In fact, if the goal is to identify only true positions of the target Gaussian mixture (this is exactly the setting of [42]), the teacher-student setup seems not necessary (see Figure 5, where the true position is also learned efficiently using the one-hidden-layer Swish neural network as the score network, but the modes are always weighed incorrectly). In addition, [42] did not take the whole denoising dynamics into account. That is, the gradient descent (GD) analysis therein was performed on the denoising score matching objective successively at only two time stages: a larger $t_1$ ("high noise") and a smaller $t_2$ ("low noise"), which is often not the case in practice.

## 5  Conclusion

In this paper, we provide a theoretical analysis of the fundamental generalization aspect of training diffusion models under both the data-independent and data-dependent settings, and early-stopping estimates of the generalization gap along the training dynamics are derived. Quantitatively, the data-independent results indicate a polynomially small generalization error that escapes from the curse of dimensionality, while the data-dependent results suggest the adverse effect of modes shift in target distributions. Numerical simulations have illustrated and verified these theoretical analyses. This work forms a basic starting point for understanding the intricacies of modern deep generative modeling and corresponding central concerns such as memorization, privacy, and copyright arising from practical applications and business products. More broadly, the approach here may have the potential to be extended to other variants in the diffusion models family, including a general SDE-based design space ([20]), consistency models ([48]), rectified flows ([27, 26]), Schrödinger bridges ([59, 56, 10, 44, 47, 25]), etc. These are certainly worthy of future exploration.

## Acknowledgements

We would like to thank Dr. Hongkang Yang for helpful discussions, and all the reviewers' valuable feedback and insightful suggestions to improve this work.

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

# A  Technical Results and Proofs

## A.1  Data-Independent Generalization Gap

To derive the theorem for the generalization error of this score-based generative model, we first give the following lemmas.

**Lemma 1** (Forward perturbation estimates). *Consider the forward diffusion process with the linear drift coefficient (2). For any $\delta > 0$, $\delta \ll 1$, with the probability of at least $1 - \delta$, we have*

$$\|\boldsymbol{x}(t)\|_\infty \lesssim C_T \left( \|\boldsymbol{x}(0)\|_\infty + \sqrt{\log(1/\delta^2)} \right), \tag{18}$$

*where $C_T := \max_{t \in [0,T]} \{r(t), r(t)v(t)\}$.*

*Proof.* When the drift coefficient $\boldsymbol{f}(\cdot, t) : \mathbb{R}^d \mapsto \mathbb{R}^d$ is linear to $\boldsymbol{x}$, i.e., $\boldsymbol{f}(\boldsymbol{x}, t) = f(t)\boldsymbol{x}$, the transition kernel $p_{t|0}$ has a closed form (8)

$$p_{t|0}(\boldsymbol{x}(t)|\boldsymbol{x}(0)) = \mathcal{N}(\boldsymbol{x}(t); r(t)\boldsymbol{x}(0), r^2(t)v^2(t)\boldsymbol{I}_d) \tag{19}$$

with $r(t) := e^{\int_0^t f(\zeta)d\zeta}$, $v(t) := \sqrt{\int_0^t \frac{g^2(\zeta)}{r^2(\zeta)}d\zeta}$. Hence, we have

$$\boldsymbol{x}(t) = r(t)\boldsymbol{x}(0) + r(t)v(t)\boldsymbol{z}, \quad \boldsymbol{z} \sim \mathcal{N}(\boldsymbol{0}, \boldsymbol{I}_d). \tag{20}$$

For any $\epsilon \sim \mathcal{N}(0, 1)$, $c > 1$, we have

$$\mathbb{P}\{\epsilon : |\epsilon| > c\} = 2 \int_c^{+\infty} \frac{1}{\sqrt{2\pi}} e^{-x^2/2} dx$$

$$\leq \frac{1}{\sqrt{2\pi}} \int_c^{+\infty} 2x e^{-x^2/2} dx = \frac{1}{\sqrt{2\pi}} \int_{c^2}^{+\infty} e^{-x/2} dx = \sqrt{\frac{2}{\pi}} e^{-c^2/2}.$$

Let $\delta = \Theta(e^{-c^2/2})$, we get

$$\mathbb{P}\{\epsilon : |\epsilon| \leq \sqrt{\log(1/\delta^2)}\} \geq 1 - \delta. \tag{21}$$

Hence, for any $\delta \in (0, 1)$ with $\delta \ll 1$, with the probability of at least $1 - \delta$, we have

$$\|\boldsymbol{x}(t)\|_\infty \lesssim C_T \left( \|\boldsymbol{x}(0)\|_\infty + \sqrt{\log(1/\delta^2)} \right) \tag{22}$$

with $C_T := \max_{t \in [0,T]} \{r(t), r(t)v(t)\}$. The proof is completed. $\qquad\square$

**Lemma 2** (Theorem 1 in [49]). *We have*

$$D_{\mathrm{KL}} \left( p_0 \| p_{0,\hat{\boldsymbol{\theta}}_n(\tau)} \right) \leq \tilde{\mathcal{L}}(\hat{\boldsymbol{\theta}}_n(\tau); g^2(\cdot)) + D_{\mathrm{KL}} \left( p_T \| \pi \right).$$

**Lemma 3.** *Both of the loss objectives $\tilde{\mathcal{L}}(\boldsymbol{\theta}; \lambda(\cdot))$ and $\bar{\tilde{\mathcal{L}}}(\bar{\boldsymbol{\theta}}; \lambda(\cdot))$ are quadratic (and hence convex).*

*Proof.* The convexity arises from the following points: (i) The score matching loss objectives $\tilde{\mathcal{L}}(\boldsymbol{\theta}; \lambda(\cdot))$ (defined in (7)) and $\bar{\tilde{\mathcal{L}}}(\bar{\boldsymbol{\theta}}; \lambda(\cdot))$ are $L^2$-metrics between the score network model and target score function; (ii) The score networks are defined as random feature models (see (12) and (13)) that are linear to trainable parameters. Therefore, using some trace techniques and basic variational calculations, it is not hard to derive the fact that the loss objectives are quadratic and hence convex with respect to trainable parameters.

(1) For $\tilde{\mathcal{L}}(\boldsymbol{\theta}; \lambda(\cdot))$, recall that $\boldsymbol{s}_{t,\boldsymbol{\theta}}(\boldsymbol{x}(t)) = \frac{1}{m}\boldsymbol{A}\sigma(\boldsymbol{W}\boldsymbol{x}(t) + \boldsymbol{U}\boldsymbol{e}(t))$, and let $\boldsymbol{s}_t(\boldsymbol{x}(t)) := \nabla_{\boldsymbol{x}(t)} \log p_t(\boldsymbol{x}(t))$, $\boldsymbol{h}_1(\boldsymbol{x}, t) := (\sqrt{\lambda(t)}/\sqrt{m})\sigma(\boldsymbol{W}\boldsymbol{x} + \boldsymbol{U}\boldsymbol{e}(t))$, $\boldsymbol{h}_2(\boldsymbol{x}, t) := \sqrt{\lambda(t)}\boldsymbol{s}_t(\boldsymbol{x})$, we have

$$\tilde{\mathcal{L}}(\boldsymbol{\theta}; \lambda(\cdot)) = \mathbb{E}_{t \sim \mathcal{U}(0,T)} \mathbb{E}_{\boldsymbol{x}(t) \sim p_t} \left[ \boldsymbol{h}_1^\top(\boldsymbol{x}(t), t)(\boldsymbol{A}/\sqrt{m})^\top (\boldsymbol{A}/\sqrt{m})\boldsymbol{h}_1(\boldsymbol{x}(t), t) \right.$$
$$\left. - 2\boldsymbol{h}_2^\top(\boldsymbol{x}(t), t)(\boldsymbol{A}/\sqrt{m})\boldsymbol{h}_1(\boldsymbol{x}(t), t) \right] + \text{constant}.$$

Since for any $\boldsymbol{h}, \bar{\boldsymbol{h}}, \boldsymbol{B}$, we have

$$\mathbb{E}_t \mathbb{E}_{\boldsymbol{x}(t)}[\boldsymbol{h}^\top(\boldsymbol{x}(t), t) \boldsymbol{B} \bar{\boldsymbol{h}}(\boldsymbol{x}(t), t)] = \mathbb{E}_t \mathbb{E}_{\boldsymbol{x}(t)}[\text{trace}(\boldsymbol{B} \bar{\boldsymbol{h}}(\boldsymbol{x}(t), t) \boldsymbol{h}^\top(\boldsymbol{x}(t), t))]$$
$$= \text{trace}(\boldsymbol{B} \mathbb{E}_t \mathbb{E}_{\boldsymbol{x}(t)}[\bar{\boldsymbol{h}}(\boldsymbol{x}(t), t) \boldsymbol{h}^\top(\boldsymbol{x}(t), t)]),$$

we further get

$$\tilde{\mathcal{L}}(\boldsymbol{\theta}; \lambda(\cdot)) = \frac{1}{m} \text{trace}(\boldsymbol{A}^\top \boldsymbol{A} \boldsymbol{B}_1) - \frac{2}{\sqrt{m}} \text{trace}(\boldsymbol{A} \boldsymbol{B}_2) + \text{constant},$$

where

$$\boldsymbol{B}_1 := \mathbb{E}_{t \sim \mathcal{U}(0,T)} \mathbb{E}_{\boldsymbol{x}(t) \sim p_t}[\boldsymbol{h}_1(\boldsymbol{x}(t), t) \boldsymbol{h}_1^\top(\boldsymbol{x}(t), t)], \quad \boldsymbol{B}_2 := \mathbb{E}_{t \sim \mathcal{U}(0,T)} \mathbb{E}_{\boldsymbol{x}(t) \sim p_t}[\boldsymbol{h}_1(\boldsymbol{x}(t), t) \boldsymbol{h}_2^\top(\boldsymbol{x}(t), t)].$$

Here, $\boldsymbol{B}_1$ is a positive semi-definite matrix, since $\boldsymbol{v}^\top \boldsymbol{B}_1 \boldsymbol{v} = \mathbb{E}_t \mathbb{E}_{\boldsymbol{x}(t)}[(\boldsymbol{v}^\top \boldsymbol{h}_1(\boldsymbol{x}(t), t))^2] \geq 0$ for any $\boldsymbol{v}$. Notice that for any $\boldsymbol{A}, \boldsymbol{B}$,

$$\text{trace}(\boldsymbol{A}^\top \boldsymbol{A} \boldsymbol{B}) = \text{trace}(\boldsymbol{A} \boldsymbol{B} \boldsymbol{A}^\top) = \sum_{i,j} \boldsymbol{B}_{ij} (\boldsymbol{A}_{:,j})^\top \boldsymbol{A}_{:,i} = \text{vec}(\boldsymbol{A})^\top (\boldsymbol{B} \otimes \boldsymbol{I}) \text{vec}(\boldsymbol{A}),$$

$$\text{trace}(\boldsymbol{A} \boldsymbol{B}) = \sum_j (\boldsymbol{A}_{:,j})^\top (\boldsymbol{B}^\top)_{:,j} = \text{vec}(\boldsymbol{A})^\top \text{vec}(\boldsymbol{B}^\top),$$

where $\otimes$ denotes the Kronecker product. Hence

$$\tilde{\mathcal{L}}(\boldsymbol{\theta}; \lambda(\cdot)) = \frac{1}{m} \text{vec}(\boldsymbol{A})^\top (\boldsymbol{B}_1 \otimes \boldsymbol{I}) \text{vec}(\boldsymbol{A}) - \frac{2}{\sqrt{m}} \text{vec}(\boldsymbol{B}_2^\top)^\top \text{vec}(\boldsymbol{A}) + \text{constant} \qquad (23)$$

is a quadratic function. It is straightforward to show that the eigenvalues of $\boldsymbol{B}_1 \otimes \boldsymbol{I}$ are the same as $\boldsymbol{B}_1$ but with multiplicity,[7] implying that $\boldsymbol{B}_1 \otimes \boldsymbol{I}$ is also positive semi-definite. Therefore,

$$\nabla_{\boldsymbol{\theta}}^2 \tilde{\mathcal{L}}(\boldsymbol{\theta}; \lambda(\cdot)) = \nabla_{\text{vec}(\boldsymbol{A})}^2 \tilde{\mathcal{L}}(\boldsymbol{\theta}; \lambda(\cdot)) = 2(\boldsymbol{B}_1 \otimes \boldsymbol{I})$$

is positive semi-definite, i.e., the loss is convex with respect to trainable parameters.

(2) For $\bar{\tilde{\mathcal{L}}}(\bar{\boldsymbol{\theta}}; \lambda(\cdot))$, notice that

$$\|\bar{\boldsymbol{s}}_{t, \bar{\boldsymbol{\theta}}}(\boldsymbol{x})\|_2^2 = \mathbb{E}_{(\boldsymbol{w}, \boldsymbol{u}) \sim \rho_0}\left[\sigma(\boldsymbol{w}^\top \boldsymbol{x} + \boldsymbol{u}^\top \boldsymbol{e}(t)) \boldsymbol{a}^\top(\boldsymbol{w}, \boldsymbol{u})\right] \mathbb{E}_{(\boldsymbol{w}', \boldsymbol{u}') \sim \rho_0}\left[\boldsymbol{a}(\boldsymbol{w}', \boldsymbol{u}') \sigma(\boldsymbol{w}'^\top \boldsymbol{x} + \boldsymbol{u}'^\top \boldsymbol{e}(t))\right]$$
$$= \mathbb{E}_{(\boldsymbol{w}, \boldsymbol{u}), (\boldsymbol{w}', \boldsymbol{u}') \sim \rho_0}\left[\boldsymbol{a}^\top(\boldsymbol{w}, \boldsymbol{u}) \sigma(\boldsymbol{w}^\top \boldsymbol{x} + \boldsymbol{u}^\top \boldsymbol{e}(t)) \sigma(\boldsymbol{w}'^\top \boldsymbol{x} + \boldsymbol{u}'^\top \boldsymbol{e}(t)) \boldsymbol{a}(\boldsymbol{w}', \boldsymbol{u}')\right].$$

Let $\boldsymbol{v} := (\boldsymbol{w}, \boldsymbol{u})$, $\boldsymbol{v}' := (\boldsymbol{w}', \boldsymbol{u}')$, $\boldsymbol{z}(t) := (\boldsymbol{x}^\top(t), \boldsymbol{e}^\top(t))^\top$, we get

$$\bar{\tilde{\mathcal{L}}}(\bar{\boldsymbol{\theta}}; \lambda(\cdot)) = \mathbb{E}_{t \sim \mathcal{U}(0,T)} \mathbb{E}_{\boldsymbol{x}(t) \sim p_t}\left[\lambda(t)\left(\|\bar{\boldsymbol{s}}_{t, \bar{\boldsymbol{\theta}}}(\boldsymbol{x}(t))\|_2^2 - 2\boldsymbol{s}_t^\top(\boldsymbol{x}(t)) \bar{\boldsymbol{s}}_{t, \bar{\boldsymbol{\theta}}}(\boldsymbol{x}(t)) + \|\boldsymbol{s}_t(\boldsymbol{x}(t))\|_2^2\right)\right]$$
$$= \mathbb{E}_{t \sim \mathcal{U}(0,T)} \mathbb{E}_{\boldsymbol{x}(t) \sim p_t}\left[\lambda(t)\left(\mathbb{E}_{\boldsymbol{v}, \boldsymbol{v}' \sim \rho_0}\left[\boldsymbol{a}^\top(\boldsymbol{v}) \sigma(\boldsymbol{v}^\top \boldsymbol{z}(t)) \sigma(\boldsymbol{v}'^\top \boldsymbol{z}(t)) \boldsymbol{a}(\boldsymbol{v}')\right]\right.\right.$$
$$\left.\left. - 2\boldsymbol{s}_t^\top(\boldsymbol{x}(t)) \mathbb{E}_{\boldsymbol{v} \sim \rho_0}\left[\boldsymbol{a}(\boldsymbol{v}) \sigma(\boldsymbol{v}^\top \boldsymbol{z}(t))\right]\right)\right] + \text{constant}$$
$$= \mathbb{E}_{\boldsymbol{v}, \boldsymbol{v}' \sim \rho_0}\left[\boldsymbol{a}^\top(\boldsymbol{v})\left(\mathbb{E}_{t \sim \mathcal{U}(0,T)} \mathbb{E}_{\boldsymbol{x}(t) \sim p_t}\left[\lambda(t) \sigma(\boldsymbol{v}^\top \boldsymbol{z}(t)) \sigma(\boldsymbol{v}'^\top \boldsymbol{z}(t))\right]\right) \boldsymbol{a}(\boldsymbol{v}')\right]$$
$$- 2\mathbb{E}_{\boldsymbol{v} \sim \rho_0}\left[\left(\mathbb{E}_{t \sim \mathcal{U}(0,T)} \mathbb{E}_{\boldsymbol{x}(t) \sim p_t}\left[\lambda(t) \boldsymbol{s}_t^\top(\boldsymbol{x}(t)) \sigma(\boldsymbol{v}^\top \boldsymbol{z}(t))\right]\right) \boldsymbol{a}(\boldsymbol{v})\right] + \text{constant}.$$

---

[7]In fact, if $\boldsymbol{A} \in \mathbb{R}^{n \times n}$, $\boldsymbol{B} \in \mathbb{R}^{m \times m}$ have the eigenvalues $\{\nu_i\}_{i=1}^n$, $\{\mu_j\}_{j=1}^m$, respectively, the eigenvalues of $\boldsymbol{A} \otimes \boldsymbol{B}$ are $\nu_i \mu_j$, $i = 1, \cdots, n$, $j = 1, \cdots, m$. This is due to the following argument. By Jordan–Chevalley decomposition, there exist invertible matrices $\boldsymbol{P}, \boldsymbol{Q}$ such that $\boldsymbol{A} = \boldsymbol{P} \boldsymbol{\Lambda} \boldsymbol{P}^{-1}$, $\boldsymbol{B} = \boldsymbol{Q} \boldsymbol{\Delta} \boldsymbol{Q}^{-1}$, where $\boldsymbol{\Lambda}, \boldsymbol{\Delta}$ are upper triangular matrices. Therefore, we have

$$\boldsymbol{A} \otimes \boldsymbol{B} = (\boldsymbol{P} \boldsymbol{\Lambda} \boldsymbol{P}^{-1}) \otimes (\boldsymbol{Q} \boldsymbol{\Delta} \boldsymbol{Q}^{-1}) = (\boldsymbol{P} \otimes \boldsymbol{Q})(\boldsymbol{\Lambda} \otimes \boldsymbol{\Delta})(\boldsymbol{P}^{-1} \otimes \boldsymbol{Q}^{-1}) = (\boldsymbol{P} \otimes \boldsymbol{Q})(\boldsymbol{\Lambda} \otimes \boldsymbol{\Delta})(\boldsymbol{P} \otimes \boldsymbol{Q})^{-1}.$$

That is, the matrices $\boldsymbol{A} \otimes \boldsymbol{B}$ and $\boldsymbol{\Lambda} \otimes \boldsymbol{\Delta}$ are similar. Notice that $\boldsymbol{\Lambda} \otimes \boldsymbol{\Delta}$ is still an upper triangular matrix with diagonal elements $\nu_i \mu_j$, $i = 1, \cdots, n$, $j = 1, \cdots, m$, we obatain the desired result.

Then for any $\phi \in L^2(\rho_0)$, by symmetry we have

$$\delta\bar{\bar{\mathcal{L}}}(\cdot;\lambda(\cdot))[\bar{\boldsymbol{\theta}},\phi] = \lim_{\epsilon\to 0}\frac{1}{\epsilon}\left(\bar{\bar{\mathcal{L}}}(\bar{\boldsymbol{\theta}}+\epsilon\phi;\lambda(\cdot)) - \bar{\bar{\mathcal{L}}}(\bar{\boldsymbol{\theta}};\lambda(\cdot))\right)$$

$$= \mathbb{E}_{\boldsymbol{v},\boldsymbol{v}'\sim\rho_0}\left[\phi^\top(\boldsymbol{v})\left(\mathbb{E}_{t\sim\mathcal{U}(0,T)}\mathbb{E}_{\boldsymbol{x}(t)\sim p_t}\left[\lambda(t)\sigma(\boldsymbol{v}^\top\boldsymbol{z}(t))\sigma(\boldsymbol{v}'^\top\boldsymbol{z}(t))\right]\right)\boldsymbol{a}(\boldsymbol{v}')\right.$$

$$+ \boldsymbol{a}^\top(\boldsymbol{v})\left(\mathbb{E}_{t\sim\mathcal{U}(0,T)}\mathbb{E}_{\boldsymbol{x}(t)\sim p_t}\left[\lambda(t)\sigma(\boldsymbol{v}^\top\boldsymbol{z}(t))\sigma(\boldsymbol{v}'^\top\boldsymbol{z}(t))\right]\right)\phi(\boldsymbol{v}')\Big]$$

$$- 2\mathbb{E}_{\boldsymbol{v}\sim\rho_0}\left[\left(\mathbb{E}_{t\sim\mathcal{U}(0,T)}\mathbb{E}_{\boldsymbol{x}(t)\sim p_t}\left[\lambda(t)\boldsymbol{s}_t^\top(\boldsymbol{x}(t))\sigma(\boldsymbol{v}^\top\boldsymbol{z}(t))\right]\right)\phi(\boldsymbol{v})\right]$$

$$= 2\mathbb{E}_{\boldsymbol{v},\boldsymbol{v}'\sim\rho_0}\left[\boldsymbol{a}^\top(\boldsymbol{v}')\left(\mathbb{E}_{t\sim\mathcal{U}(0,T)}\mathbb{E}_{\boldsymbol{x}(t)\sim p_t}\left[\lambda(t)\sigma(\boldsymbol{v}^\top\boldsymbol{z}(t))\sigma(\boldsymbol{v}'^\top\boldsymbol{z}(t))\right]\right)\phi(\boldsymbol{v})\right]$$

$$- 2\mathbb{E}_{\boldsymbol{v}\sim\rho_0}\left[\left(\mathbb{E}_{t\sim\mathcal{U}(0,T)}\mathbb{E}_{\boldsymbol{x}(t)\sim p_t}\left[\lambda(t)\boldsymbol{s}_t^\top(\boldsymbol{x}(t))\sigma(\boldsymbol{v}^\top\boldsymbol{z}(t))\right]\right)\phi(\boldsymbol{v})\right]$$

$$= 2\big\langle\mathbb{E}_{\boldsymbol{v}'\sim\rho_0}\left[K(\boldsymbol{v},\boldsymbol{v}';\lambda(\cdot))\boldsymbol{a}(\boldsymbol{v}')\right] - \boldsymbol{k}(\boldsymbol{v};\lambda(\cdot)), \phi(\boldsymbol{v})\big\rangle_{L^2(\rho_0)},$$

where

$$K(\boldsymbol{v},\boldsymbol{v}';\lambda(\cdot)) := \mathbb{E}_{t\sim\mathcal{U}(0,T)}\mathbb{E}_{\boldsymbol{x}(t)\sim p_t}\left[\lambda(t)\sigma(\boldsymbol{v}^\top\boldsymbol{z}(t))\sigma(\boldsymbol{v}'^\top\boldsymbol{z}(t))\right],$$

$$\boldsymbol{k}(\boldsymbol{v};\lambda(\cdot)) := \mathbb{E}_{t\sim\mathcal{U}(0,T)}\mathbb{E}_{\boldsymbol{x}(t)\sim p_t}\left[\lambda(t)\boldsymbol{s}_t(\boldsymbol{x}(t))\sigma(\boldsymbol{v}^\top\boldsymbol{z}(t))\right].$$

This yields

$$\frac{\delta\bar{\bar{\mathcal{L}}}(\cdot;\lambda(\cdot))}{\delta\bar{\boldsymbol{\theta}}} = 2\mathbb{E}_{\boldsymbol{v}'\sim\rho_0}\left[K(\boldsymbol{v},\boldsymbol{v}';\lambda(\cdot))\boldsymbol{a}(\boldsymbol{v}')\right] - 2\boldsymbol{k}(\boldsymbol{v};\lambda(\cdot)), \tag{24}$$

and

$$\bar{\bar{\mathcal{L}}}(\bar{\boldsymbol{\theta}}_1;\lambda(\cdot)) + \left\langle\frac{\delta\bar{\bar{\mathcal{L}}}(\cdot;\lambda(\cdot))}{\delta\bar{\boldsymbol{\theta}}}\bigg|_{\bar{\boldsymbol{\theta}}=\bar{\boldsymbol{\theta}}_1}, \bar{\boldsymbol{\theta}}_2 - \bar{\boldsymbol{\theta}}_1\right\rangle_{L^2(\rho_0)} - \bar{\bar{\mathcal{L}}}(\bar{\boldsymbol{\theta}}_2;\lambda(\cdot))$$

$$= \mathbb{E}_{\boldsymbol{v},\boldsymbol{v}'\sim\rho_0}\left[\boldsymbol{a}_1^\top(\boldsymbol{v})K(\boldsymbol{v},\boldsymbol{v}';\lambda(\cdot))\boldsymbol{a}_1(\boldsymbol{v}')\right] - 2\mathbb{E}_{\boldsymbol{v}\sim\rho_0}\left[\boldsymbol{k}^\top(\boldsymbol{v};\lambda(\cdot))\boldsymbol{a}_1(\boldsymbol{v})\right]$$

$$+ 2\big\langle\mathbb{E}_{\boldsymbol{v}'\sim\rho_0}\left[K(\boldsymbol{v},\boldsymbol{v}';\lambda(\cdot))\boldsymbol{a}_1(\boldsymbol{v}')\right] - \boldsymbol{k}(\boldsymbol{v};\lambda(\cdot)), \boldsymbol{a}_2(\boldsymbol{v}) - \boldsymbol{a}_1(\boldsymbol{v})\big\rangle_{L^2(\rho_0)}$$

$$- \mathbb{E}_{\boldsymbol{v},\boldsymbol{v}'\sim\rho_0}\left[\boldsymbol{a}_2^\top(\boldsymbol{v})K(\boldsymbol{v},\boldsymbol{v}';\lambda(\cdot))\boldsymbol{a}_2(\boldsymbol{v}')\right] + 2\mathbb{E}_{\boldsymbol{v}\sim\rho_0}\left[\boldsymbol{k}^\top(\boldsymbol{v};\lambda(\cdot))\boldsymbol{a}_2(\boldsymbol{v})\right]$$

$$= -\mathbb{E}_{\boldsymbol{v},\boldsymbol{v}'\sim\rho_0}\left[(\boldsymbol{a}_2(\boldsymbol{v})-\boldsymbol{a}_1(\boldsymbol{v}))^\top K(\boldsymbol{v},\boldsymbol{v}';\lambda(\cdot))(\boldsymbol{a}_2(\boldsymbol{v}')-\boldsymbol{a}_1(\boldsymbol{v}'))\right]$$

$$= -\mathbb{E}_{t\sim\mathcal{U}(0,T)}\mathbb{E}_{\boldsymbol{x}(t)\sim p_t}\left[\lambda(t)\mathbb{E}_{\boldsymbol{v},\boldsymbol{v}'\sim\rho_0}\left[(\boldsymbol{a}_2(\boldsymbol{v})-\boldsymbol{a}_1(\boldsymbol{v}))^\top\sigma(\boldsymbol{v}^\top\boldsymbol{z}(t))\sigma(\boldsymbol{v}'^\top\boldsymbol{z}(t))(\boldsymbol{a}_2(\boldsymbol{v}')-\boldsymbol{a}_1(\boldsymbol{v}'))\right]\right]$$

$$= -\mathbb{E}_{t\sim\mathcal{U}(0,T)}\mathbb{E}_{\boldsymbol{x}(t)\sim p_t}\left[\lambda(t)\left\|\mathbb{E}_{\boldsymbol{v}\sim\rho_0}\left[(\boldsymbol{a}_2(\boldsymbol{v})-\boldsymbol{a}_1(\boldsymbol{v}))\sigma(\boldsymbol{v}^\top\boldsymbol{z}(t))\right]\right\|_2^2\right] \leq 0,$$

hence the functional $\bar{\bar{\mathcal{L}}}(\bar{\boldsymbol{\theta}};\lambda(\cdot))$ is convex with respect to $\bar{\boldsymbol{\theta}}$ (given any positive weighting function $\lambda(\cdot)$). The proof is completed. $\qquad\square$

Since the weighting function is fixed in our analysis, we omit the notation $\lambda(\cdot)$ without ambiguity in the following contents.

**Lemma 4.** *For any $\tau > 0$ and $\boldsymbol{\theta}, \bar{\boldsymbol{\theta}}$, we have*

$$\bar{\bar{\mathcal{L}}}\left(\bar{\boldsymbol{\theta}}(\tau)\right) - \bar{\bar{\mathcal{L}}}\left(\bar{\boldsymbol{\theta}}\right) \lesssim \frac{\left\|\bar{\boldsymbol{s}}_{0,\bar{\boldsymbol{\theta}}_0}\right\|_{\mathcal{H}}^2 + \left\|\bar{\boldsymbol{s}}_{0,\bar{\boldsymbol{\theta}}}\right\|_{\mathcal{H}}^2}{\tau}, \quad \tilde{\mathcal{L}}\left(\boldsymbol{\theta}(\tau)\right) - \tilde{\mathcal{L}}\left(\boldsymbol{\theta}\right) \lesssim \frac{\left\|\boldsymbol{s}_{0,\boldsymbol{\theta}_0}\right\|_{\mathcal{H}}^2 + \left\|\boldsymbol{s}_{0,\boldsymbol{\theta}}\right\|_{\mathcal{H}}^2}{\tau}.$$

*Proof.* (1) For the loss objective $\bar{\bar{\mathcal{L}}}$, we define the Lyapunov function

$$\bar{E}(\tau) := \tau\left(\bar{\bar{\mathcal{L}}}\left(\bar{\boldsymbol{\theta}}(\tau)\right) - \bar{\bar{\mathcal{L}}}\left(\bar{\boldsymbol{\theta}}\right)\right) + \frac{1}{2}\left\|\boldsymbol{a}_\tau - \boldsymbol{a}\right\|_{L^2(\rho_0)}^2.$$

Then, we have

$$\frac{d}{d\tau}\bar{E}(\tau) = \left(\bar{\bar{\mathcal{L}}}\left(\bar{\boldsymbol{\theta}}(\tau)\right) - \bar{\bar{\mathcal{L}}}\left(\bar{\boldsymbol{\theta}}\right)\right) + \tau\cdot\frac{d}{d\tau}\bar{\bar{\mathcal{L}}}\left(\bar{\boldsymbol{\theta}}(\tau)\right) + \left\langle\boldsymbol{a}_\tau - \boldsymbol{a}, \frac{d}{d\tau}\boldsymbol{a}_\tau\right\rangle_{L^2(\rho_0)}$$

$$\leq \left(\bar{\bar{\mathcal{L}}}\left(\bar{\boldsymbol{\theta}}(\tau)\right) - \bar{\bar{\mathcal{L}}}\left(\bar{\boldsymbol{\theta}}\right)\right) - \left\langle\boldsymbol{a}_\tau - \boldsymbol{a}, \frac{\delta\bar{\bar{\mathcal{L}}}}{\delta\bar{\boldsymbol{\theta}}}\bigg|_{\bar{\boldsymbol{\theta}}=\bar{\boldsymbol{\theta}}(\tau)}\right\rangle_{L^2(\rho_0)},$$

where the last inequality holds since

$$\frac{d}{d\tau}\bar{\bar{\mathcal{L}}}\left(\bar{\boldsymbol{\theta}}(\tau)\right) = \left\langle \frac{\delta\bar{\bar{\mathcal{L}}}}{\delta\bar{\boldsymbol{\theta}}}\bigg|_{\bar{\boldsymbol{\theta}}=\bar{\boldsymbol{\theta}}(\tau)}, \frac{d}{d\tau}\bar{\boldsymbol{\theta}}(\tau)\right\rangle_{L^2(\rho_0)} = -\left\langle \frac{\delta\bar{\bar{\mathcal{L}}}}{\delta\bar{\boldsymbol{\theta}}}\bigg|_{\bar{\boldsymbol{\theta}}=\bar{\boldsymbol{\theta}}(\tau)}, \frac{\delta\bar{\bar{\mathcal{L}}}}{\delta\bar{\boldsymbol{\theta}}}\bigg|_{\bar{\boldsymbol{\theta}}=\bar{\boldsymbol{\theta}}(\tau)}\right\rangle_{L^2(\rho_0)} \leq 0.$$

By convexity, for any $\tau_1, \tau_2$, it holds that

$$\bar{\bar{\mathcal{L}}}\left(\bar{\boldsymbol{\theta}}(\tau_1)\right) + \left\langle \boldsymbol{a}_{\tau_2} - \boldsymbol{a}_{\tau_1}, \frac{\delta\bar{\bar{\mathcal{L}}}}{\delta\bar{\boldsymbol{\theta}}}\bigg|_{\bar{\boldsymbol{\theta}}=\bar{\boldsymbol{\theta}}(\tau_1)}\right\rangle_{L^2(\rho_0)} \leq \bar{\bar{\mathcal{L}}}\left(\bar{\boldsymbol{\theta}}(\tau_2)\right),$$

hence $\frac{d}{d\tau}\bar{E}(\tau) \leq 0$. We conclude that $\bar{E}(\tau) \leq \bar{E}(0)$, or equivalently

$$\tau\left(\bar{\bar{\mathcal{L}}}\left(\bar{\boldsymbol{\theta}}(\tau)\right) - \bar{\bar{\mathcal{L}}}\left(\bar{\boldsymbol{\theta}}\right)\right) + \frac{1}{2}\|\boldsymbol{a}_\tau - \boldsymbol{a}\|^2_{L^2(\rho_0)} \leq \frac{1}{2}\|\boldsymbol{a}_0 - \boldsymbol{a}\|^2_{L^2(\rho_0)}.$$

Therefore, note that $\left\|\bar{\boldsymbol{s}}_{0,\bar{\boldsymbol{\theta}}}\right\|^2_{\mathcal{H}} := \mathbb{E}_{\rho_0}[\|\boldsymbol{a}\|^2_2]$ (let $\mathcal{H} := \mathcal{H}_{k_{\rho_0}}$), we obtain

$$\bar{\bar{\mathcal{L}}}\left(\bar{\boldsymbol{\theta}}(\tau)\right) - \bar{\bar{\mathcal{L}}}\left(\bar{\boldsymbol{\theta}}\right) \lesssim \frac{\|\boldsymbol{a}_0 - \boldsymbol{a}\|^2_{L^2(\rho_0)}}{\tau} \lesssim \frac{\left\|\bar{\boldsymbol{s}}_{0,\bar{\boldsymbol{\theta}}_0}\right\|^2_{\mathcal{H}} + \left\|\bar{\boldsymbol{s}}_{0,\bar{\boldsymbol{\theta}}}\right\|^2_{\mathcal{H}}}{\tau},$$

which gives the desired estimate.

(2) For the loss objective $\tilde{\mathcal{L}}$, the argument is almost the same, except replacing the Lyapunov function by

$$E(\tau) := \tau\left(\tilde{\mathcal{L}}\left(\boldsymbol{\theta}(\tau)\right) - \tilde{\mathcal{L}}\left(\boldsymbol{\theta}\right)\right) + \frac{1}{2m}\|\boldsymbol{A}_\tau - \boldsymbol{A}\|^2_F,$$

and $\langle\cdot,\cdot\rangle_{L^2(\rho_0)}$ by $\langle\cdot,\cdot\rangle$. Note that $\boldsymbol{\theta} = \text{vec}(\boldsymbol{A})/\sqrt{m}$ and $\|\boldsymbol{s}_{0,\boldsymbol{\theta}}\|^2_{\mathcal{H}} = \|\boldsymbol{A}\|^2_F/m$, we obtain

$$\tilde{\mathcal{L}}\left(\boldsymbol{\theta}(\tau)\right) - \tilde{\mathcal{L}}\left(\boldsymbol{\theta}\right) \lesssim \frac{\|\boldsymbol{A}_0 - \boldsymbol{A}\|^2_F}{m\tau} \lesssim \frac{\|\boldsymbol{s}_{0,\boldsymbol{\theta}_0}\|^2_{\mathcal{H}} + \|\boldsymbol{s}_{0,\boldsymbol{\theta}}\|^2_{\mathcal{H}}}{\tau},$$

which completes the proof. $\qquad\square$

**Lemma 5.** *Suppose that the loss objectives $\tilde{\mathcal{L}}$, $\tilde{\mathcal{L}}^{(n)}$, $\bar{\bar{\mathcal{L}}}$, $\bar{\bar{\mathcal{L}}}^{(n)}$ are bounded at the initialization, then for any $\tau > 0$, we have*

$$\left\|\boldsymbol{s}_{0,\boldsymbol{\theta}(\tau)}\right\|_{\mathcal{H}}, \left\|\boldsymbol{s}_{0,\hat{\boldsymbol{\theta}}_n(\tau)}\right\|_{\mathcal{H}} \lesssim \left\|\boldsymbol{s}_{0,\boldsymbol{\theta}_0}\right\|_{\mathcal{H}} + \sqrt{\frac{\tau}{m}}, \quad \left\|\bar{\boldsymbol{s}}_{0,\bar{\boldsymbol{\theta}}(\tau)}\right\|_{\mathcal{H}}, \left\|\bar{\boldsymbol{s}}_{0,\hat{\bar{\boldsymbol{\theta}}}_n(\tau)}\right\|_{\mathcal{H}} \lesssim \left\|\bar{\boldsymbol{s}}_{0,\bar{\boldsymbol{\theta}}_0}\right\|_{\mathcal{H}} + \sqrt{\tau}.$$

*Proof.* It's sufficient to prove $\left\|\bar{\boldsymbol{s}}_{0,\bar{\boldsymbol{\theta}}(\tau)}\right\|_{\mathcal{H}} \lesssim \sqrt{\tau}$, and the rest part follows similarly.

Since

$$\|\boldsymbol{a}_\tau\|_{L^2(\rho_0)}\frac{d}{d\tau}\|\boldsymbol{a}_\tau\|_{L^2(\rho_0)} = \frac{1}{2}\frac{d}{d\tau}\|\boldsymbol{a}_\tau\|^2_{L^2(\rho_0)} = \left\langle \boldsymbol{a}_\tau, \frac{d}{d\tau}\boldsymbol{a}_\tau\right\rangle_{L^2(\rho_0)} = \left\langle \boldsymbol{a}_\tau, -\frac{\delta\bar{\bar{\mathcal{L}}}}{\delta\bar{\boldsymbol{\theta}}}\bigg|_{\bar{\boldsymbol{\theta}}=\bar{\boldsymbol{\theta}}(\tau)}\right\rangle_{L^2(\rho_0)},$$

applying Cauchy–Schwartz inequality yields

$$\frac{d}{d\tau}\|\boldsymbol{a}_\tau\|_{L^2(\rho_0)} = \left\langle \frac{\boldsymbol{a}_\tau}{\|\boldsymbol{a}_\tau\|_{L^2(\rho_0)}}, -\frac{\delta\bar{\bar{\mathcal{L}}}}{\delta\bar{\boldsymbol{\theta}}}\bigg|_{\bar{\boldsymbol{\theta}}=\bar{\boldsymbol{\theta}}(\tau)}\right\rangle_{L^2(\rho_0)}$$

$$\leq \left\|\frac{\delta\bar{\bar{\mathcal{L}}}}{\delta\bar{\boldsymbol{\theta}}}\bigg|_{\bar{\boldsymbol{\theta}}=\bar{\boldsymbol{\theta}}(\tau)}\right\|_{L^2(\rho_0)} = \sqrt{-\frac{d}{d\tau}\bar{\bar{\mathcal{L}}}\left(\bar{\boldsymbol{\theta}}(\tau)\right)}.$$

Thus, again by Cauchy–Schwartz inequality, for any $\tau > \tau_0 \geq 0$, we have

$$\|\boldsymbol{a}_\tau\|_{L^2(\rho_0)} - \|\boldsymbol{a}_{\tau_0}\|_{L^2(\rho_0)} \leq \int_{\tau_0}^\tau \sqrt{-\frac{d}{ds}\bar{\bar{\mathcal{L}}}\left(\bar{\boldsymbol{\theta}}(s)\right)}ds$$

$$\leq \sqrt{\tau - \tau_0}\sqrt{-\bar{\bar{\mathcal{L}}}\left(\bar{\boldsymbol{\theta}}(\tau)\right) + \bar{\bar{\mathcal{L}}}\left(\bar{\boldsymbol{\theta}}(\tau_0)\right)}$$

$$\leq \sqrt{\tau}\sqrt{\bar{\bar{\mathcal{L}}}\left(\bar{\boldsymbol{\theta}}(0)\right)}.$$

By choosing $\tau_0 = 0$, we have $\|\boldsymbol{a}_\tau\|_{L^2(\rho_0)} \lesssim \left\|\bar{\boldsymbol{s}}_{0,\bar{\boldsymbol{\theta}}(0)}\right\|_{\mathcal{H}} + \sqrt{\tau}$, hence $\left\|\bar{\boldsymbol{s}}_{0,\bar{\boldsymbol{\theta}}(\tau)}\right\|_{\mathcal{H}} \lesssim \left\|\bar{\boldsymbol{s}}_{0,\bar{\boldsymbol{\theta}}(0)}\right\|_{\mathcal{H}} +$
$\sqrt{\tau}$, which completes the proof. $\qquad\square$

**Lemma 6** (Monte Carlo estimates). *Define the Monte Carlo error*

$$\mathrm{Err}_{\mathrm{MC}} = \mathrm{Err}_{\mathrm{MC}}(\boldsymbol{\theta}, \bar{\boldsymbol{\theta}}; T, \lambda(\cdot)) := \mathbb{E}_{t\sim\mathcal{U}(0,T)} \left[\lambda(t) \cdot \mathbb{E}_{\boldsymbol{x}(t)\sim p_t} \left[\|\boldsymbol{s}_{t,\boldsymbol{\theta}}(\boldsymbol{x}(t)) - \bar{\boldsymbol{s}}_{t,\bar{\boldsymbol{\theta}}}(\boldsymbol{x}(t))\|_2^2\right]\right]. \tag{25}$$

*Suppose that $\|\boldsymbol{x}(0)\|_\infty \leq 1$, and the trainable parameter $\boldsymbol{a}$ and embedding function $\boldsymbol{e}(\cdot)$ are both bounded. Then, given any $\bar{\boldsymbol{\theta}}$, for any $\delta > 0$, $\delta \ll 1$, with the probability of at least $1 - \delta$, there exists $\boldsymbol{\theta}$ such that*

$$\mathrm{Err}_{\mathrm{MC}} \lesssim \frac{\log^2(1/\delta^2)}{m} d, \tag{26}$$

*where $\lesssim$ hides universal positive constants only depending on $T$.*

*Proof.* Fix any $\bar{\boldsymbol{\theta}}$. According to Lemma 1, for any $\delta > 0$, $\delta \ll 1$, with the probability of at least $1 - \delta$, we have

$$\|\boldsymbol{x}(t)\|_\infty \lesssim C_T \left(1 + \sqrt{\log(1/\delta^2)}\right) \triangleq C_{T,\delta}. \tag{27}$$

Based on the representation (13), for any $\boldsymbol{W} = (\boldsymbol{w}_1, \dots, \boldsymbol{w}_m)^\top \in \mathbb{R}^{m \times d}$, $\boldsymbol{U} = (\boldsymbol{u}_1, \dots, \boldsymbol{u}_m)^\top \in \mathbb{R}^{m \times d_e}$ with $(\boldsymbol{w}_i, \boldsymbol{u}_i) \sim \rho_0$, $i = 1, \cdots, m$, let $\boldsymbol{A} := (\boldsymbol{a}_1, \dots, \boldsymbol{a}_m) \in \mathbb{R}^{d \times m}$ with $\boldsymbol{a}_i := \boldsymbol{a}(\boldsymbol{w}_i, \boldsymbol{u}_i)$ for $i = 1, \cdots, m$, and

$$\boldsymbol{s}_{t,\boldsymbol{\theta}}(\boldsymbol{x}) := \frac{1}{m} \boldsymbol{A}\sigma(\boldsymbol{W}\boldsymbol{x} + \boldsymbol{U}\boldsymbol{e}(t)) = \frac{1}{m} \sum_{i=1}^m \boldsymbol{a}_i \sigma(\boldsymbol{w}_i^\top \boldsymbol{x} + \boldsymbol{u}_i^\top \boldsymbol{e}(t)), \tag{28}$$

then $\mathbb{E}_{\boldsymbol{W},\boldsymbol{U}}[\boldsymbol{s}_{t,\boldsymbol{\theta}}(\boldsymbol{x})] = \mathbb{E}_{(\boldsymbol{w},\boldsymbol{u})\sim\rho_0} \left[\boldsymbol{a}(\boldsymbol{w},\boldsymbol{u})\sigma(\boldsymbol{w}^\top \boldsymbol{x} + \boldsymbol{u}^\top \boldsymbol{e}(t))\right] = \bar{\boldsymbol{s}}_{t,\bar{\boldsymbol{\theta}}}(\boldsymbol{x})$. For $k = 1, \cdots, d$, let

$$Z_{t,k}(\boldsymbol{W}, \boldsymbol{U}) := \left\|s_{t,\boldsymbol{\theta},k}(\boldsymbol{x}) - \bar{s}_{t,\bar{\boldsymbol{\theta}},k}(\boldsymbol{x})\right\|_{L^2(p_t)} = \mathbb{E}_{\boldsymbol{x}\sim p_t}^{1/2} \left[\left|s_{t,\boldsymbol{\theta},k}(\boldsymbol{x}) - \bar{s}_{t,\bar{\boldsymbol{\theta}},k}(\boldsymbol{x})\right|^2\right]$$

$$= \mathbb{E}_{\boldsymbol{x}\sim p_t}^{1/2} \left[\left|\frac{1}{m} \sum_{i=1}^m a_{i,k}\sigma(\boldsymbol{w}_i^\top \boldsymbol{x} + \boldsymbol{u}_i^\top \boldsymbol{e}(t)) - \mathbb{E}_{(\boldsymbol{w},\boldsymbol{u})\sim\rho_0}\left[a_k(\boldsymbol{w},\boldsymbol{u})\sigma(\boldsymbol{w}^\top \boldsymbol{x} + \boldsymbol{u}^\top \boldsymbol{e}(t))\right]\right|^2\right].$$

If $(\tilde{\boldsymbol{W}}, \tilde{\boldsymbol{U}})$ is different from $(\boldsymbol{W}, \boldsymbol{U})$ at only one component indexed by $i$, we have

$$\left|Z_{t,k}(\boldsymbol{W}, \boldsymbol{U}) - Z_{t,k}(\tilde{\boldsymbol{W}}, \tilde{\boldsymbol{U}})\right|$$

$$= \left|\left\|s_{t,\boldsymbol{\theta},k}(\boldsymbol{x}) - \bar{s}_{t,\bar{\boldsymbol{\theta}},k}(\boldsymbol{x})\right\|_{L^2(p_t)} - \left\|s_{t,\tilde{\boldsymbol{\theta}},k}(\boldsymbol{x}) - \bar{s}_{t,\bar{\boldsymbol{\theta}},k}(\boldsymbol{x})\right\|_{L^2(p_t)}\right|$$

$$\overset{(i)}{\leq} \left\|s_{t,\boldsymbol{\theta},k}(\boldsymbol{x}) - s_{t,\tilde{\boldsymbol{\theta}},k}(\boldsymbol{x})\right\|_{L^2(p_t)}$$

$$= \frac{1}{m} \left\|a_{i,k}\sigma(\boldsymbol{w}_i^\top \boldsymbol{x} + \boldsymbol{u}_i^\top \boldsymbol{e}(t)) - \tilde{a}_{i,k}\sigma(\tilde{\boldsymbol{w}}_i^\top \boldsymbol{x} + \tilde{\boldsymbol{u}}_i^\top \boldsymbol{e}(t))\right\|_{L^2(p_t)}$$

$$\leq \frac{1}{m} \left(|a_{i,k}| \left\|\sigma(\boldsymbol{w}_i^\top \boldsymbol{x} + \boldsymbol{u}_i^\top \boldsymbol{e}(t))\right\|_{L^2(p_t)} + |\tilde{a}_{i,k}| \left\|\sigma(\tilde{\boldsymbol{w}}_i^\top \boldsymbol{x} + \tilde{\boldsymbol{u}}_i^\top \boldsymbol{e}(t))\right\|_{L^2(p_t)}\right)$$

$$\overset{(ii)}{\leq} \frac{1}{m} \left(|a_{i,k}| \left(\|\boldsymbol{w}_i\|_1 C_{T,\delta} + \|\boldsymbol{u}_i\|_1 \|\boldsymbol{e}(t)\|_\infty\right) + |\tilde{a}_{i,k}| \left(\|\tilde{\boldsymbol{w}}_i\|_1 C_{T,\delta} + \|\tilde{\boldsymbol{u}}_i\|_1 \|\boldsymbol{e}(t)\|_\infty\right)\right)$$

$$\overset{(iii)}{\leq} \frac{1}{m} \left(|a_{i,k}| + |\tilde{a}_{i,k}|\right) \left(C_{T,\delta} + \|\boldsymbol{e}(t)\|_\infty\right)$$

$$\overset{(iv)}{\lesssim} \frac{1}{m} \left(C_{T,\delta} + C_{T,\boldsymbol{e}}\right),$$

where (i) is from the triangle inequality, (ii) is due to the fact that $|\sigma(y)| = |\mathrm{ReLU}(y)| \leq |y|$ for any $y \in \mathbb{R}$, the triangle and Hölder's inequality and (27), (iii) follows from the positive homogeneity property of the ReLU activation, and (iv) is due to the boundness of the trainable parameter $\boldsymbol{a}$ and

embedding function $\boldsymbol{e}(\cdot)$. By McDiarmid's inequality (see e.g., Lemma 26.4 in [43]), for any $\delta > 0$, with the probability of at least $1 - \delta$, we have

$$|Z_{t,k}(\boldsymbol{W}, \boldsymbol{U}) - \mathbb{E}_{\boldsymbol{W}, \boldsymbol{U}}[Z_{t,k}(\boldsymbol{W}, \boldsymbol{U})]| \lesssim \frac{1}{m} \left(C_{T,\delta} + C_{T,e}\right) \sqrt{m \log(2/\delta)/2} \qquad (29)$$

$$\lesssim \left(C_{T,\delta} + C_{T,e}\right) \sqrt{\frac{\log(1/\delta)}{m}}. \qquad (30)$$

Since

$$\mathbb{E}_{\boldsymbol{W}, \boldsymbol{U}}\left[Z_{t,k}^2(\boldsymbol{W}, \boldsymbol{U})\right]$$

$$= \mathbb{E}_{\boldsymbol{W}, \boldsymbol{U}}\left[\mathbb{E}_{\boldsymbol{x} \sim p_t}\left[\left|s_{t,\boldsymbol{\theta},k}(\boldsymbol{x}) - \bar{s}_{t,\bar{\boldsymbol{\theta}},k}(\boldsymbol{x})\right|^2\right]\right]$$

$$\overset{(v)}{=} \mathbb{E}_{\boldsymbol{x} \sim p_t}\left[\mathbb{E}_{\boldsymbol{W}, \boldsymbol{U}}\left[\left|s_{t,\boldsymbol{\theta},k}(\boldsymbol{x}) - \bar{s}_{t,\bar{\boldsymbol{\theta}},k}(\boldsymbol{x})\right|^2\right]\right]$$

$$= \frac{1}{m^2} \mathbb{E}_{\boldsymbol{x} \sim p_t}\left[\mathbb{E}_{\boldsymbol{W}, \boldsymbol{U}}\left[\left|\sum_{i=1}^m \left(a_{i,k}\sigma(\boldsymbol{w}_i^\top \boldsymbol{x} + \boldsymbol{u}_i^\top \boldsymbol{e}(t)) - \mathbb{E}_{(\boldsymbol{w}, \boldsymbol{u}) \sim \rho_0}\left[a_k(\boldsymbol{w}, \boldsymbol{u})\sigma(\boldsymbol{w}^\top \boldsymbol{x} + \boldsymbol{u}^\top \boldsymbol{e}(t))\right]\right)\right|^2\right]\right]$$

$$= \frac{1}{m^2} \mathbb{E}_{\boldsymbol{x} \sim p_t}\left[\mathbb{E}_{\boldsymbol{W}, \boldsymbol{U}}\left[\sum_{i=1}^m \left(a_{i,k}\sigma(\boldsymbol{w}_i^\top \boldsymbol{x} + \boldsymbol{u}_i^\top \boldsymbol{e}(t)) - \mathbb{E}_{(\boldsymbol{w}, \boldsymbol{u}) \sim \rho_0}\left[a_k(\boldsymbol{w}, \boldsymbol{u})\sigma(\boldsymbol{w}^\top \boldsymbol{x} + \boldsymbol{u}^\top \boldsymbol{e}(t))\right]\right)^2\right]\right]$$

$$+ \frac{1}{m^2} \mathbb{E}_{\boldsymbol{x} \sim p_t}\left[\mathbb{E}_{\boldsymbol{W}, \boldsymbol{U}}\left[\sum_{i \neq j} \left(a_{i,k}\sigma(\boldsymbol{w}_i^\top \boldsymbol{x} + \boldsymbol{u}_i^\top \boldsymbol{e}(t)) - \mathbb{E}_{(\boldsymbol{w}, \boldsymbol{u}) \sim \rho_0}\left[a_k(\boldsymbol{w}, \boldsymbol{u})\sigma(\boldsymbol{w}^\top \boldsymbol{x} + \boldsymbol{u}^\top \boldsymbol{e}(t))\right]\right)\right.\right.$$

$$\left.\left. \times \left(a_{j,k}\sigma(\boldsymbol{w}_j^\top \boldsymbol{x} + \boldsymbol{u}_j^\top \boldsymbol{e}(t)) - \mathbb{E}_{(\boldsymbol{w}, \boldsymbol{u}) \sim \rho_0}\left[a_k(\boldsymbol{w}, \boldsymbol{u})\sigma(\boldsymbol{w}^\top \boldsymbol{x} + \boldsymbol{u}^\top \boldsymbol{e}(t))\right]\right)\right]\right]$$

$$= \frac{1}{m^2} \mathbb{E}_{\boldsymbol{x} \sim p_t}\left[\sum_{i=1}^m \mathbb{E}_{(\boldsymbol{w}, \boldsymbol{u}) \sim \rho_0}\left[\left(a_k(\boldsymbol{w}, \boldsymbol{u})\sigma(\boldsymbol{w}^\top \boldsymbol{x} + \boldsymbol{u}^\top \boldsymbol{e}(t)) - \mathbb{E}_{(\boldsymbol{w}, \boldsymbol{u}) \sim \rho_0}\left[a_k(\boldsymbol{w}, \boldsymbol{u})\sigma(\boldsymbol{w}^\top \boldsymbol{x} + \boldsymbol{u}^\top \boldsymbol{e}(t))\right]\right)^2\right]\right]$$

$$+ \frac{1}{m^2} \mathbb{E}_{\boldsymbol{x} \sim p_t}\left[\sum_{i \neq j} \mathbb{E}_{(\boldsymbol{w}_i, \boldsymbol{u}_i) \sim \rho_0}\left[\left(a_{i,k}\sigma(\boldsymbol{w}_i^\top \boldsymbol{x} + \boldsymbol{u}_i^\top \boldsymbol{e}(t)) - \mathbb{E}_{(\boldsymbol{w}, \boldsymbol{u}) \sim \rho_0}\left[a_k(\boldsymbol{w}, \boldsymbol{u})\sigma(\boldsymbol{w}^\top \boldsymbol{x} + \boldsymbol{u}^\top \boldsymbol{e}(t))\right]\right)\right]\right.$$

$$\left. \times \mathbb{E}_{(\boldsymbol{w}_j, \boldsymbol{u}_j) \sim \rho_0}\left[\left(a_{j,k}\sigma(\boldsymbol{w}_j^\top \boldsymbol{x} + \boldsymbol{u}_j^\top \boldsymbol{e}(t)) - \mathbb{E}_{(\boldsymbol{w}, \boldsymbol{u}) \sim \rho_0}\left[a_k(\boldsymbol{w}, \boldsymbol{u})\sigma(\boldsymbol{w}^\top \boldsymbol{x} + \boldsymbol{u}^\top \boldsymbol{e}(t))\right]\right)\right]\right]$$

$$= \frac{1}{m^2} \mathbb{E}_{\boldsymbol{x} \sim p_t}\left[\sum_{i=1}^m \mathbb{E}_{(\boldsymbol{w}, \boldsymbol{u}) \sim \rho_0}\left[\left(a_k(\boldsymbol{w}, \boldsymbol{u})\sigma(\boldsymbol{w}^\top \boldsymbol{x} + \boldsymbol{u}^\top \boldsymbol{e}(t)) - \mathbb{E}_{(\boldsymbol{w}, \boldsymbol{u}) \sim \rho_0}\left[a_k(\boldsymbol{w}, \boldsymbol{u})\sigma(\boldsymbol{w}^\top \boldsymbol{x} + \boldsymbol{u}^\top \boldsymbol{e}(t))\right]\right)^2\right]\right]$$

$$\leq \frac{1}{m} \mathbb{E}_{\boldsymbol{x} \sim p_t}\left[\mathbb{E}_{(\boldsymbol{w}, \boldsymbol{u}) \sim \rho_0}\left[\left(a_k(\boldsymbol{w}, \boldsymbol{u})\sigma(\boldsymbol{w}^\top \boldsymbol{x} + \boldsymbol{u}^\top \boldsymbol{e}(t))\right)^2\right]\right]$$

$$\overset{(ii)}{\leq} \frac{1}{m} \mathbb{E}_{\boldsymbol{x} \sim p_t}\left[\mathbb{E}_{(\boldsymbol{w}, \boldsymbol{u}) \sim \rho_0}\left[\left(|a_k(\boldsymbol{w}, \boldsymbol{u})| \left(\|\boldsymbol{w}\|_1 C_{T,\delta} + \|\boldsymbol{u}\|_1 \|\boldsymbol{e}(t)\|_\infty\right)\right)^2\right]\right]$$

$$\overset{(iii)}{\leq} \frac{1}{m} \mathbb{E}_{\boldsymbol{x} \sim p_t}\left[\mathbb{E}_{(\boldsymbol{w}, \boldsymbol{u}) \sim \rho_0}\left[\left(|a_k(\boldsymbol{w}, \boldsymbol{u})| \left(C_{T,\delta} + \|\boldsymbol{e}(t)\|_\infty\right)\right)^2\right]\right]$$

$$\overset{(iv)}{\lesssim} \frac{1}{m} \mathbb{E}_{\boldsymbol{x} \sim p_t}\left[\mathbb{E}_{(\boldsymbol{w}, \boldsymbol{u}) \sim \rho_0}\left[\left(C_{T,\delta} + C_{T,e}\right)^2\right]\right] \qquad (31)$$

$$\lesssim \left(C_{T,\delta} + C_{T,e}\right)^2 \frac{1}{m}, \qquad (32)$$

where (v) is due to Fubini's theorem, and (ii), (iii), (iv) is the same as before. By the triangle inequality, Jensen's inequality, (29) and (31), we obtain

$$\mathbb{E}_{\boldsymbol{x} \sim p_t}\left[\|\boldsymbol{s}_{t,\boldsymbol{\theta}}(\boldsymbol{x}) - \bar{\boldsymbol{s}}_{t,\bar{\boldsymbol{\theta}}}(\boldsymbol{x})\|_2^2\right] = \sum_{k=1}^d \mathbb{E}_{\boldsymbol{x} \sim p_t}\left[\left|s_{t,\boldsymbol{\theta},k}(\boldsymbol{x}) - \bar{s}_{t,\bar{\boldsymbol{\theta}},k}(\boldsymbol{x})\right|^2\right]$$

$$= \sum_{k=1}^{d} Z_{t,k}^2(\boldsymbol{W}, \boldsymbol{U})$$

$$\leq \sum_{k=1}^{d} \left( |Z_{t,k}(\boldsymbol{W}, \boldsymbol{U}) - \mathbb{E}_{\boldsymbol{W}, \boldsymbol{U}}[Z_{t,k}(\boldsymbol{W}, \boldsymbol{U})]| + |\mathbb{E}_{\boldsymbol{W}, \boldsymbol{U}}[Z_{t,k}(\boldsymbol{W}, \boldsymbol{U})]| \right)^2$$

$$\lesssim \sum_{k=1}^{d} \left( |Z_{t,k}(\boldsymbol{W}, \boldsymbol{U}) - \mathbb{E}_{\boldsymbol{W}, \boldsymbol{U}}[Z_{t,k}(\boldsymbol{W}, \boldsymbol{U})]|^2 + \mathbb{E}_{\boldsymbol{W}, \boldsymbol{U}}[Z_{t,k}^2(\boldsymbol{W}, \boldsymbol{U})] \right)$$

$$\lesssim \sum_{k=1}^{d} (C_{T,\delta} + C_{T,\boldsymbol{e}})^2 \frac{\log(1/\delta)}{m} = (C_{T,\delta} + C_{T,\boldsymbol{e}})^2 \frac{\log(1/\delta)}{m} d,$$

$$(33)$$

which gives

$$\text{Err}_{\text{MC}} = \frac{1}{T} \int_0^T \lambda(t) \cdot \mathbb{E}_{\boldsymbol{x} \sim p_t} \left[ \|\boldsymbol{s}_{t,\boldsymbol{\theta}}(\boldsymbol{x}) - \bar{\boldsymbol{s}}_{t,\bar{\boldsymbol{\theta}}}(\boldsymbol{x})\|_2^2 \right] dt$$

$$\lesssim (C_{T,\delta} + C_{T,\boldsymbol{e}})^2 \frac{\log(1/\delta)}{m} d \cdot \frac{1}{T} \int_0^T \lambda(t) dt \leq \frac{\log^2(1/\delta^2)}{m} d.$$

Obviously, $\frac{1}{m}\|\boldsymbol{A}\|_F^2 = \frac{1}{m}\sum_{i=1}^m \|\boldsymbol{a}(\boldsymbol{w}_i, \boldsymbol{u}_i)\|_2^2 = \|\boldsymbol{s}_{t,\boldsymbol{\theta}}\|_{\mathcal{H}_{k_{\rho_0}}}^2 < \infty$. The proof is completed. $\square$

Now we are ready to prove the main theorem.

*Proof of Theorem 1.* Based on Lemma 2, we have

$$D_{\text{KL}} \left( p_0 \| p_{0, \hat{\boldsymbol{\theta}}_n(\tau)} \right) \leq \tilde{\mathcal{L}}(\hat{\boldsymbol{\theta}}_n(\tau); g^2(\cdot)) + D_{\text{KL}} \left( p_T \| \pi \right).$$

To bound the term $\tilde{\mathcal{L}}(\hat{\boldsymbol{\theta}}_n(\tau); g^2(\cdot))$, we use the following decomposition:

$$\tilde{\mathcal{L}}(\hat{\boldsymbol{\theta}}_n(\tau)) = \left[ \tilde{\mathcal{L}}(\hat{\boldsymbol{\theta}}_n(\tau)) - \tilde{\mathcal{L}}(\boldsymbol{\theta}(\tau)) \right] + \tilde{\mathcal{L}}(\boldsymbol{\theta}(\tau))$$

$$\lesssim \left[ \tilde{\mathcal{L}}(\hat{\boldsymbol{\theta}}_n(\tau)) - \tilde{\mathcal{L}}(\boldsymbol{\theta}(\tau)) \right] + \bar{\tilde{\mathcal{L}}}(\bar{\boldsymbol{\theta}}(\tau)) + \text{Monte Carlo} \triangleq I_1 + I_2 + I_3.$$

According to Lemma 4, we obtain

$$I_2 \lesssim \bar{\tilde{\mathcal{L}}} \left( \bar{\boldsymbol{\theta}}^* \right) + \frac{1}{\tau} \left( \|\bar{\boldsymbol{s}}_{0, \bar{\boldsymbol{\theta}}_0}\|_{\mathcal{H}}^2 + \|\bar{\boldsymbol{s}}_{0, \bar{\boldsymbol{\theta}}^*}\|_{\mathcal{H}}^2 \right).$$

The term $I_3$ can be further divided into

$$I_3 := \mathbb{E}_{t \sim \mathcal{U}(0,T)} \left[ \lambda(t) \cdot \mathbb{E}_{\boldsymbol{x}(t) \sim p_t} \left[ \|\boldsymbol{s}_{t,\boldsymbol{\theta}(\tau)}(\boldsymbol{x}(t)) - \bar{\boldsymbol{s}}_{t,\bar{\boldsymbol{\theta}}(\tau)}(\boldsymbol{x}(t))\|_2^2 \right] \right]$$

$$\lesssim \mathbb{E}_{t \sim \mathcal{U}(0,T)} \left[ \lambda(t) \cdot \mathbb{E}_{\boldsymbol{x}(t) \sim p_t} \left[ \|\bar{\boldsymbol{s}}_{t,\bar{\boldsymbol{\theta}}(\tau)}(\boldsymbol{x}(t)) - \bar{\boldsymbol{s}}_{t,\bar{\boldsymbol{\theta}}^*}(\boldsymbol{x}(t))\|_2^2 \right] \right]$$

$$+ \mathbb{E}_{t \sim \mathcal{U}(0,T)} \left[ \lambda(t) \cdot \mathbb{E}_{\boldsymbol{x}(t) \sim p_t} \left[ \|\boldsymbol{s}_{t,\boldsymbol{\theta}^*}(\boldsymbol{x}(t)) - \bar{\boldsymbol{s}}_{t,\bar{\boldsymbol{\theta}}^*}(\boldsymbol{x}(t))\|_2^2 \right] \right]$$

$$+ \mathbb{E}_{t \sim \mathcal{U}(0,T)} \left[ \lambda(t) \cdot \mathbb{E}_{\boldsymbol{x}(t) \sim p_t} \left[ \|\boldsymbol{s}_{t,\boldsymbol{\theta}(\tau)}(\boldsymbol{x}(t)) - \boldsymbol{s}_{t,\boldsymbol{\theta}^*}(\boldsymbol{x}(t))\|_2^2 \right] \right]$$

$$=: I_{3,1} + I_{3,2} + I_{3,3},$$

where $\boldsymbol{\theta}^*$ is the Monte Carlo estimator of $\bar{\boldsymbol{\theta}}^*$. By Lemma 6, for any $\delta > 0$, $\delta \ll 1$, with the probability of at least $1 - \delta$, it holds that

$$I_{3,2} = \text{Err}_{\text{MC}}(\boldsymbol{\theta}^*, \bar{\boldsymbol{\theta}}^*; T, \lambda(\cdot)) \lesssim \frac{\log^2(1/\delta^2)}{m} d, \tag{34}$$

while Lemma 4 gives

$$I_{3,1} \lesssim \bar{\tilde{\mathcal{L}}}(\bar{\boldsymbol{\theta}}(\tau)) + \bar{\tilde{\mathcal{L}}}(\bar{\boldsymbol{\theta}}^*) = I_2 + \bar{\tilde{\mathcal{L}}}(\bar{\boldsymbol{\theta}}^*) \lesssim \bar{\tilde{\mathcal{L}}} \left( \bar{\boldsymbol{\theta}}^* \right) + \frac{1}{\tau} \left( \|\bar{\boldsymbol{s}}_{0, \bar{\boldsymbol{\theta}}_0}\|_{\mathcal{H}}^2 + \|\bar{\boldsymbol{s}}_{0, \bar{\boldsymbol{\theta}}^*}\|_{\mathcal{H}}^2 \right),$$

and similarly,

$$I_{3,3} \lesssim \tilde{\mathcal{L}}\left(\boldsymbol{\theta}^*\right) + \frac{1}{\tau}\left(\|\boldsymbol{s}_{0,\boldsymbol{\theta}_0}\|_{\mathcal{H}}^2 + \|\boldsymbol{s}_{0,\boldsymbol{\theta}^*}\|_{\mathcal{H}}^2\right).$$

Hence, for any $\delta > 0$, $\delta \ll 1$, with the probability of at least $1 - \delta$, it holds that

$$I_3 \lesssim \frac{\log^2(1/\delta^2)}{m}d + \bar{\tilde{\mathcal{L}}}\left(\bar{\boldsymbol{\theta}}^*\right) + \tilde{\mathcal{L}}\left(\boldsymbol{\theta}^*\right) + \frac{1}{\tau}\left(\|\bar{\boldsymbol{s}}_{0,\bar{\boldsymbol{\theta}}_0}\|_{\mathcal{H}}^2 + \|\bar{\boldsymbol{s}}_{0,\bar{\boldsymbol{\theta}}^*}\|_{\mathcal{H}}^2 + \|\boldsymbol{s}_{0,\boldsymbol{\theta}_0}\|_{\mathcal{H}}^2 + \|\boldsymbol{s}_{0,\boldsymbol{\theta}^*}\|_{\mathcal{H}}^2\right).$$

For the term $I_1$, we have

$$\sqrt{\tilde{\mathcal{L}}(\hat{\boldsymbol{\theta}}_n(\tau))} - \sqrt{\tilde{\mathcal{L}}(\boldsymbol{\theta}(\tau))}$$

$$= \left\{\mathbb{E}_{t\sim\mathcal{U}(0,T)}\left[\lambda(t) \cdot \mathbb{E}_{\boldsymbol{x}(t)\sim p_t}\left[\|\boldsymbol{s}_{t,\hat{\boldsymbol{\theta}}_n(\tau)}(\boldsymbol{x}(t)) - \nabla_{\boldsymbol{x}(t)}\log p_t(\boldsymbol{x}(t))\|_2^2\right]\right]\right\}^{\frac{1}{2}}$$

$$\quad - \left\{\mathbb{E}_{t\sim\mathcal{U}(0,T)}\left[\lambda(t) \cdot \mathbb{E}_{\boldsymbol{x}(t)\sim p_t}\left[\|\boldsymbol{s}_{t,\boldsymbol{\theta}(\tau)}(\boldsymbol{x}(t)) - \nabla_{\boldsymbol{x}(t)}\log p_t(\boldsymbol{x}(t))\|_2^2\right]\right]\right\}^{\frac{1}{2}}$$

$$= \left\{\mathbb{E}_{t\sim\mathcal{U}(0,T)}\mathbb{E}_{\boldsymbol{x}(t)\sim p_t}\left[\lambda(t)\|\boldsymbol{s}_{t,\hat{\boldsymbol{\theta}}_n(\tau)}(\boldsymbol{x}(t)) - \nabla_{\boldsymbol{x}(t)}\log p_t(\boldsymbol{x}(t))\|_2^2\right]\right\}^{\frac{1}{2}}$$

$$\quad - \left\{\mathbb{E}_{t\sim\mathcal{U}(0,T)}\mathbb{E}_{\boldsymbol{x}(t)\sim p_t}\left[\lambda(t)\|\boldsymbol{s}_{t,\boldsymbol{\theta}(\tau)}(\boldsymbol{x}(t)) - \nabla_{\boldsymbol{x}(t)}\log p_t(\boldsymbol{x}(t))\|_2^2\right]\right\}^{\frac{1}{2}}$$

$$\le \left\{\mathbb{E}_{t\sim\mathcal{U}(0,T)}\mathbb{E}_{\boldsymbol{x}(t)\sim p_t}\left[\lambda(t)\|\boldsymbol{s}_{t,\hat{\boldsymbol{\theta}}_n(\tau)}(\boldsymbol{x}(t)) - \boldsymbol{s}_{t,\boldsymbol{\theta}(\tau)}(\boldsymbol{x}(t))\|_2^2\right]\right\}^{\frac{1}{2}}$$

$$= \left\{\mathbb{E}_{t\sim\mathcal{U}(0,T)}\mathbb{E}_{\boldsymbol{x}(t)\sim p_t}\left[\lambda(t)\left\|\frac{1}{m}\sum_{i=1}^m \hat{\boldsymbol{a}}_i(\tau)\sigma(\boldsymbol{w}_i^\top \boldsymbol{x}(t) + \boldsymbol{u}_i^\top \boldsymbol{e}(t)) - \frac{1}{m}\sum_{i=1}^m \boldsymbol{a}_i(\tau)\sigma(\boldsymbol{w}_i^\top \boldsymbol{x}(t) + \boldsymbol{u}_i^\top \boldsymbol{e}(t))\right\|_2^2\right]\right\}^{\frac{1}{2}}.$$

Notice that

$$\left\|\frac{1}{m}\sum_{i=1}^m(\hat{\boldsymbol{a}}_i(\tau) - \boldsymbol{a}_i(\tau))\sigma(\boldsymbol{w}_i^\top \boldsymbol{x}(t) + \boldsymbol{u}_i^\top \boldsymbol{e}(t))\right\|_2^2$$

$$\le \frac{1}{m^2}\left(\sum_{i=1}^m \|\hat{\boldsymbol{a}}_i(\tau) - \boldsymbol{a}_i(\tau)\|_2 \left|\sigma(\boldsymbol{w}_i^\top \boldsymbol{x}(t) + \boldsymbol{u}_i^\top \boldsymbol{e}(t))\right|\right)^2$$

$$\le \frac{1}{m^2}\sum_{i=1}^m \|\hat{\boldsymbol{a}}_i(\tau) - \boldsymbol{a}_i(\tau)\|_2^2 \sum_{i=1}^m \left|\sigma(\boldsymbol{w}_i^\top \boldsymbol{x}(t) + \boldsymbol{u}_i^\top \boldsymbol{e}(t))\right|^2$$

$$\lesssim \frac{1}{m^2}\sum_{i=1}^m \|\hat{\boldsymbol{a}}_i(\tau) - \boldsymbol{a}_i(\tau)\|_2^2 \sum_{i=1}^m \left(\left|\boldsymbol{w}_i^\top \boldsymbol{x}(t)\right|^2 + \left|\boldsymbol{u}_i^\top \boldsymbol{e}(t)\right|^2\right)$$

$$\le \frac{1}{m^2}\sum_{i=1}^m \|\hat{\boldsymbol{a}}_i(\tau) - \boldsymbol{a}_i(\tau)\|_2^2 \sum_{i=1}^m \left(\|\boldsymbol{w}_i\|_1^2 \|\boldsymbol{x}(t)\|_\infty^2 + \|\boldsymbol{u}_i\|_1^2 \|\boldsymbol{e}(t)\|_\infty^2\right)$$

$$\lesssim \frac{1}{m^2}\sum_{i=1}^m \|\hat{\boldsymbol{a}}_i(\tau) - \boldsymbol{a}_i(\tau)\|_2^2 \sum_{i=1}^m \left(C_{T,\delta}^2 + C_{T,\boldsymbol{e}}^2\right),$$

which gives

$$\sqrt{\tilde{\mathcal{L}}(\hat{\boldsymbol{\theta}}_n(\tau))} - \sqrt{\tilde{\mathcal{L}}(\boldsymbol{\theta}(\tau))} \lesssim \left\{\frac{1}{m}\sum_{i=1}^m \|\hat{\boldsymbol{a}}_i(\tau) - \boldsymbol{a}_i(\tau)\|_2^2 \left(C_{T,\delta}^2 + C_{T,\boldsymbol{e}}^2\right)\right\}^{\frac{1}{2}}$$

$$\le (C_{T,\delta} + C_{T,\boldsymbol{e}})\left\{\frac{1}{m}\sum_{i=1}^m \|\hat{\boldsymbol{a}}_i(\tau) - \boldsymbol{a}_i(\tau)\|_2^2\right\}^{\frac{1}{2}}.$$

Here, the triangle inequality, Cauchy–Schwartz inequality, the fact that $|\sigma(y)| = |\text{ReLU}(y)| \le |y|$ for any $y \in \mathbb{R}$, Hölder's inequality, the positive homogeneity property of the ReLU activation, and

the boundness of the input data, embedding function $\boldsymbol{e}(t)$ and weighting function $\lambda(t)$. Thus, we get

$$\tilde{\mathcal{L}}(\hat{\boldsymbol{\theta}}_n(\tau)) - \tilde{\mathcal{L}}(\boldsymbol{\theta}(\tau))$$

$$\lesssim \frac{1}{m} \left(C_{T,\delta}^2 + C_{T,e}^2\right) \sum_{i=1}^m \|\hat{\boldsymbol{a}}_i(\tau) - \boldsymbol{a}_i(\tau)\|_2^2 + \sqrt{\tilde{\mathcal{L}}(\boldsymbol{\theta}(\tau))} \left(C_{T,\delta} + C_{T,e}\right) \left\{ \frac{1}{m} \sum_{i=1}^m \|\hat{\boldsymbol{a}}_i(\tau) - \boldsymbol{a}_i(\tau)\|_2^2 \right\}^{\frac{1}{2}}$$

$$\lesssim \sqrt{\tilde{\mathcal{L}}(\boldsymbol{\theta}^*) + \frac{1}{\tau} \left(\|\boldsymbol{s}_{0,\boldsymbol{\theta}_0}\|_{\mathcal{H}}^2 + \|\boldsymbol{s}_{0,\boldsymbol{\theta}^*}\|_{\mathcal{H}}^2\right)} \left(C_{T,\delta} + C_{T,e}\right) \left\{ \frac{1}{m} \sum_{i=1}^m \|\hat{\boldsymbol{a}}_i(\tau) - \boldsymbol{a}_i(\tau)\|_2^2 \right\}^{\frac{1}{2}}$$

$$+ \frac{1}{m} \left(C_{T,\delta}^2 + C_{T,e}^2\right) \sum_{i=1}^m \|\hat{\boldsymbol{a}}_i(\tau) - \boldsymbol{a}_i(\tau)\|_2^2,$$

where the last inequality follows from Lemma 4. We further deduce that

$$\frac{1}{m} \sum_{i=1}^m \|\hat{\boldsymbol{a}}_i(\tau) - \boldsymbol{a}_i(\tau)\|_2^2$$

$$= \frac{1}{m} \sum_{i=1}^m \left\| \int_0^\tau \frac{d}{d\tau_0} (\hat{\boldsymbol{a}}_i(\tau_0) - \boldsymbol{a}_i(\tau_0)) d\tau_0 \right\|_2^2$$

$$= \frac{1}{m} \sum_{i=1}^m \left\| \int_0^\tau \left( \nabla_{\boldsymbol{\theta}_i(\tau_0)} \tilde{\mathcal{L}}(\boldsymbol{\theta}(\tau_0)) - \nabla_{\hat{\boldsymbol{\theta}}_{n,i}(\tau_0)} \hat{\tilde{\mathcal{L}}}_n(\hat{\boldsymbol{\theta}}_n(\tau_0)) \right) d\tau_0 \right\|_2^2$$

$$= \frac{1}{m^2} \sum_{i=1}^m \left\| \int_0^\tau \left( 2\mathbb{E}_{t \sim \mathcal{U}(0,T)} \left[ \lambda(t) \mathbb{E}_{\boldsymbol{x}(t) \sim p_t} \left[ \left( \boldsymbol{s}_{t,\boldsymbol{\theta}(\tau_0)}(\boldsymbol{x}(t)) - \nabla_{\boldsymbol{x}(t)} \log p_t(\boldsymbol{x}(t)) \right) \sigma(\boldsymbol{w}_i^\top \boldsymbol{x}(t) + \boldsymbol{u}_i^\top \boldsymbol{e}(t)) \right] \right] \right.$$

$$\left. - 2\mathbb{E}_{t \sim \mathcal{U}(0,T)} \left[ \lambda(t) \mathbb{E}_{\boldsymbol{x}(t) \sim p_t^{(n)}} \left[ \left( \boldsymbol{s}_{t,\hat{\boldsymbol{\theta}}_n(\tau_0)}(\boldsymbol{x}(t)) - \nabla_{\boldsymbol{x}(t)} \log p_t(\boldsymbol{x}(t)) \right) \sigma(\boldsymbol{w}_i^\top \boldsymbol{x}(t) + \boldsymbol{u}_i^\top \boldsymbol{e}(t)) \right] \right] \right) d\tau_0 \right\|_2^2,$$

where $p_t^{(n)}$ denotes the empirical distribution of $p_t$. Note that

$$\|\boldsymbol{\theta}\|_2^2 = \|\text{vec}(\boldsymbol{A})\|_2^2/m = \|\boldsymbol{A}\|_F^2/m = \|\boldsymbol{s}_{0,\boldsymbol{\theta}}\|_{\mathcal{H}}^2, \tag{35}$$

by Lemma 5 we get $\|\boldsymbol{\theta}(\tau)\|_2 = \|\boldsymbol{s}_{0,\boldsymbol{\theta}(\tau)}\|_{\mathcal{H}} \lesssim \|\boldsymbol{s}_{0,\boldsymbol{\theta}_0}\|_{\mathcal{H}} + \sqrt{\tau/m}$. For any $t \in [0,T]$, define the function space

$$\mathcal{F}_t := \{ \boldsymbol{f}_1(\boldsymbol{x}(t); \boldsymbol{\theta}_1(\tau)) f_2(\boldsymbol{x}(t); \boldsymbol{\theta}_2) : \boldsymbol{f}_1 \in \mathcal{F}_{1,t}, f_2 \in \mathcal{F}_{2,t} \},$$

where

$$\mathcal{F}_{1,t} := \left\{ \boldsymbol{s}_{t,\boldsymbol{\theta}(\tau)}(\boldsymbol{x}(t)) - \nabla_{\boldsymbol{x}(t)} \log p_t(\boldsymbol{x}(t)) : \|\boldsymbol{\theta}(\tau)\|_2 \lesssim \|\boldsymbol{s}_{0,\boldsymbol{\theta}_0}\|_{\mathcal{H}} + \sqrt{\tau/m} \right\},$$

$$\mathcal{F}_{2,t} := \left\{ \sigma(\boldsymbol{w}^\top \boldsymbol{x}(t) + \boldsymbol{u}^\top \boldsymbol{e}(t)) : \|\boldsymbol{w}\|_1 + \|\boldsymbol{u}\|_1 \leq 1 \right\}.$$

Then, according to Theorem A.5 in [61], for any $\delta \in (0,1)$, with the probability at least $1 - \delta$ over the choice of the dataset $\mathcal{D}_{\boldsymbol{x}} = \{\boldsymbol{x}_i\}_{i=1}^n$, it holds that

$$\left| \mathbb{E}_{\boldsymbol{x}(t) \sim p_t} \left[ \left( \boldsymbol{s}_{t,\boldsymbol{\theta}(\tau)}(\boldsymbol{x}(t)) - \nabla_{\boldsymbol{x}(t)} \log p_t(\boldsymbol{x}(t)) \right) \sigma(\boldsymbol{w}_i^\top \boldsymbol{x}(t) + \boldsymbol{u}_i^\top \boldsymbol{e}(t)) \right] \right.$$

$$\left. - \mathbb{E}_{\boldsymbol{x}(t) \sim p_t^{(n)}} \left[ \left( \boldsymbol{s}_{t,\hat{\boldsymbol{\theta}}_n(\tau)}(\boldsymbol{x}(t)) - \nabla_{\boldsymbol{x}(t)} \log p_t(\boldsymbol{x}(t)) \right) \sigma(\boldsymbol{w}_i^\top \boldsymbol{x}(t) + \boldsymbol{u}_i^\top \boldsymbol{e}(t)) \right] \right|$$

$$\leq \left| \mathbb{E}_{\boldsymbol{x}(t) \sim p_t} \left[ \left( \boldsymbol{s}_{t,\boldsymbol{\theta}(\tau)}(\boldsymbol{x}(t)) - \nabla_{\boldsymbol{x}(t)} \log p_t(\boldsymbol{x}(t)) \right) \sigma(\boldsymbol{w}_i^\top \boldsymbol{x}(t) + \boldsymbol{u}_i^\top \boldsymbol{e}(t)) \right] \right.$$

$$\left. - \mathbb{E}_{\boldsymbol{x}(t) \sim p_t^{(n)}} \left[ \left( \boldsymbol{s}_{t,\boldsymbol{\theta}(\tau)}(\boldsymbol{x}(t)) - \nabla_{\boldsymbol{x}(t)} \log p_t(\boldsymbol{x}(t)) \right) \sigma(\boldsymbol{w}_i^\top \boldsymbol{x}(t) + \boldsymbol{u}_i^\top \boldsymbol{e}(t)) \right] \right|$$

$$+ \left| \mathbb{E}_{\boldsymbol{x}(t) \sim p_t^{(n)}} \left[ \left( \boldsymbol{s}_{t,\boldsymbol{\theta}(\tau)}(\boldsymbol{x}(t)) - \nabla_{\boldsymbol{x}(t)} \log p_t(\boldsymbol{x}(t)) \right) \sigma(\boldsymbol{w}_i^\top \boldsymbol{x}(t) + \boldsymbol{u}_i^\top \boldsymbol{e}(t)) \right] \right.$$

$$\left. - \mathbb{E}_{\boldsymbol{x}(t) \sim p_t^{(n)}} \left[ \left( \boldsymbol{s}_{t,\hat{\boldsymbol{\theta}}_n(\tau)}(\boldsymbol{x}(t)) - \nabla_{\boldsymbol{x}(t)} \log p_t(\boldsymbol{x}(t)) \right) \sigma(\boldsymbol{w}_i^\top \boldsymbol{x}(t) + \boldsymbol{u}_i^\top \boldsymbol{e}(t)) \right] \right|$$

$$\lesssim \widehat{\text{Rad}}_n(\mathcal{F}_t) + \sup_{\boldsymbol{f} \in \mathcal{F}_t, \boldsymbol{x}(t) \in [-C_{T,\delta}, C_{T,\delta}]^d} |\boldsymbol{f}(\boldsymbol{x}(t))| \sqrt{\frac{\log(2/\delta)}{n}}$$

$$+ \left| \mathbb{E}_{\boldsymbol{x}(t) \sim p_t^{(n)}} \left[ \left( \boldsymbol{s}_{t,\boldsymbol{\theta}(\tau)}(\boldsymbol{x}(t)) - \boldsymbol{s}_{t,\hat{\boldsymbol{\theta}}_n(\tau)}(\boldsymbol{x}(t)) \right) \sigma(\boldsymbol{w}_i^\top \boldsymbol{x}(t) + \boldsymbol{u}_i^\top \boldsymbol{e}(t)) \right] \right| =: J_1 + J_2 + J_3,$$

where $\widehat{\mathrm{Rad}}_n$ denotes the (empirical) Rademacher complexity of $\mathcal{F}_t$ on $\mathcal{D}_{\boldsymbol{x}} = \{\boldsymbol{x}_i\}_{i=1}^n$, and all the inequalities hold in the element-wise sense.

(i) For $J_1$, according to Lemma A.6 in [61], we have

$$\widehat{\mathrm{Rad}}_n(\mathcal{F}_t) \leq \left( \sup_{\boldsymbol{f}_1 \in \mathcal{F}_{1,t}, \, \boldsymbol{x}(t) \in [-C_{T,\delta}, C_{T,\delta}]^d} |\boldsymbol{f}_1(\boldsymbol{x}(t))| + \sup_{f_2 \in \mathcal{F}_{2,t}, \, \boldsymbol{x}(t) \in [-C_{T,\delta}, C_{T,\delta}]^d} |f_2(\boldsymbol{x}(t))| \right)$$
$$\cdot \left( \widehat{\mathrm{Rad}}_n(\mathcal{F}_{1,t}) + \widehat{\mathrm{Rad}}_n(\mathcal{F}_{2,t}) \right).$$

Note that

$$\left| \sigma(\boldsymbol{w}^\top \boldsymbol{x}(t) + \boldsymbol{u}^\top \boldsymbol{e}(t)) \right| \leq \left| \boldsymbol{w}^\top \boldsymbol{x}(t) + \boldsymbol{u}^\top \boldsymbol{e}(t) \right|$$
$$\lesssim \|\boldsymbol{w}\|_1 \|\boldsymbol{x}(t)\|_\infty + \|\boldsymbol{u}\|_1 \|\boldsymbol{e}(t)\|_\infty$$
$$\leq C_{T,\delta} + C_{T,\boldsymbol{e}}, \tag{36}$$

which yields

$$\left| \boldsymbol{s}_{t,\boldsymbol{\theta}(\tau)}(\boldsymbol{x}(t)) \right| = \left| \frac{1}{\sqrt{m}} \sum_{i=1}^m \boldsymbol{\theta}_i(\tau) \sigma(\boldsymbol{w}_i^\top \boldsymbol{x}(t) + \boldsymbol{u}_i^\top \boldsymbol{e}(t)) \right|$$
$$\leq \frac{1}{\sqrt{m}} \sum_{i=1}^m |\boldsymbol{\theta}_i(\tau)| \left| \sigma(\boldsymbol{w}_i^\top \boldsymbol{x}(t) + \boldsymbol{u}_i^\top \boldsymbol{e}(t)) \right|$$
$$\leq \frac{1}{\sqrt{m}} (C_{T,\delta} + C_{T,\boldsymbol{e}}) \sum_{i=1}^m |\boldsymbol{\theta}_i(\tau)|$$
$$\leq (C_{T,\delta} + C_{T,\boldsymbol{e}}) \|\boldsymbol{\theta}(\tau)\|_2$$
$$\lesssim (C_{T,\delta} + C_{T,\boldsymbol{e}}) \left( \|\boldsymbol{s}_{0,\boldsymbol{\theta}_0}\|_{\mathcal{H}} + \sqrt{\tau/m} \right), \tag{37}$$

and hence

$$|\boldsymbol{f}_1(\boldsymbol{x}(t))| = \left| \boldsymbol{s}_{t,\boldsymbol{\theta}(\tau)}(\boldsymbol{x}(t)) - \nabla_{\boldsymbol{x}(t)} \log p_t(\boldsymbol{x}(t)) \right|$$
$$\lesssim (C_{T,\delta} + C_{T,\boldsymbol{e}}) \left( \|\boldsymbol{s}_{0,\boldsymbol{\theta}_0}\|_{\mathcal{H}} + \sqrt{\tau/m} \right) + C'_{T,\delta},$$
$$|f_2(\boldsymbol{x}(t))| = \left| \sigma(\boldsymbol{w}^\top \boldsymbol{x}(t) + \boldsymbol{u}^\top \boldsymbol{e}(t)) \right| \lesssim C_{T,\delta} + C_{T,\boldsymbol{e}},$$

where $C'_{T,\delta} := \max_{\boldsymbol{x}(t) \in [-C_{T,\delta}, C_{T,\delta}]^d} \left| \nabla_{\boldsymbol{x}(t)} \log p_t(\boldsymbol{x}(t)) \right|$. This gives

$$\widehat{\mathrm{Rad}}_n(\mathcal{F}_t) \lesssim (C_{T,\delta} + C_{T,\boldsymbol{e}}) \left( \|\boldsymbol{s}_{0,\boldsymbol{\theta}_0}\|_{\mathcal{H}} + \sqrt{\tau/m} + 1 \right) \left( \widehat{\mathrm{Rad}}_n(\mathcal{F}_{1,t}) + \widehat{\mathrm{Rad}}_n(\mathcal{F}_{2,t}) \right).$$

Let

$$\mathcal{F}'_{1,t} := \left\{ \boldsymbol{s}_{t,\boldsymbol{\theta}(\tau)}(\boldsymbol{x}(t)) : \|\boldsymbol{\theta}(\tau)\|_2 \lesssim \|\boldsymbol{s}_{0,\boldsymbol{\theta}_0}\|_{\mathcal{H}} + \sqrt{\tau/m} \right\},$$

according to Lemma 26.6 in [43], we get $\widehat{\mathrm{Rad}}_n(\mathcal{F}_{1,t}) \leq \widehat{\mathrm{Rad}}_n(\mathcal{F}'_{1,t})$. Since

$$n \widehat{\mathrm{Rad}}_n(\mathcal{F}'_{1,t}) = \mathbb{E}_{\boldsymbol{\xi}} \left[ \sup_{\|\boldsymbol{\theta}(\tau)\|_2 \lesssim \|\boldsymbol{s}_{0,\boldsymbol{\theta}_0}\|_{\mathcal{H}} + \sqrt{\tau/m}} \sum_{j=1}^n \xi_j \boldsymbol{s}_{t,\boldsymbol{\theta}(\tau)}(\boldsymbol{x}_j(t)) \right]$$
$$= \mathbb{E}_{\boldsymbol{\xi}} \left[ \sup_{\|\boldsymbol{\theta}(\tau)\|_2 \lesssim \|\boldsymbol{s}_{0,\boldsymbol{\theta}_0}\|_{\mathcal{H}} + \sqrt{\tau/m}} \sum_{j=1}^n \xi_j \frac{1}{\sqrt{m}} \sum_{i=1}^m \boldsymbol{\theta}_i(\tau) \sigma(\boldsymbol{w}_i^\top \boldsymbol{x}_j(t) + \boldsymbol{u}_i^\top \boldsymbol{e}(t)) \right]$$
$$\leq \frac{1}{\sqrt{m}} \mathbb{E}_{\boldsymbol{\xi}} \left[ \sup_{\substack{\|\boldsymbol{\theta}(\tau)\|_2 \lesssim \|\boldsymbol{s}_{0,\boldsymbol{\theta}_0}\|_{\mathcal{H}} + \sqrt{\tau/m} \\ \|\boldsymbol{w}_i\|_1 + \|\boldsymbol{u}_i\|_1 \leq 1, \, i \in [n]}} \sum_{i=1}^m \boldsymbol{\theta}_i(\tau) \sum_{j=1}^n \xi_j \sigma(\boldsymbol{w}_i^\top \boldsymbol{x}_j(t) + \boldsymbol{u}_i^\top \boldsymbol{e}(t)) \right]$$

$$\leq \frac{1}{\sqrt{m}} \mathbb{E}_{\boldsymbol{\xi}} \left[ \sup_{\|\boldsymbol{\theta}(\tau)\|_2 \lesssim \left\|\boldsymbol{s}_{0,\boldsymbol{\theta}_0}\right\|_{\mathcal{H}} + \sqrt{\tau/m}} \sum_{i=1}^{m} |\boldsymbol{\theta}_i(\tau)| \sup_{\|\boldsymbol{w}_i\|_1 + \|\boldsymbol{u}_i\|_1 \leq 1,\, i \in [n]} \left| \sum_{j=1}^{n} \xi_j \sigma(\boldsymbol{w}_i^\top \boldsymbol{x}_j(t) + \boldsymbol{u}_i^\top \boldsymbol{e}(t)) \right| \right]$$

$$\leq \mathbb{E}_{\boldsymbol{\xi}} \left[ \sup_{\|\boldsymbol{\theta}(\tau)\|_2 \lesssim \left\|\boldsymbol{s}_{0,\boldsymbol{\theta}_0}\right\|_{\mathcal{H}} + \sqrt{\tau/m}} \|\boldsymbol{\theta}(\tau)\|_2 \sup_{\|\boldsymbol{w}\|_1 + \|\boldsymbol{u}\|_1 \leq 1} \left| \sum_{j=1}^{n} \xi_j \sigma(\boldsymbol{w}^\top \boldsymbol{x}_j(t) + \boldsymbol{u}^\top \boldsymbol{e}(t)) \right| \right]$$

$$\lesssim \left( \left\|\boldsymbol{s}_{0,\boldsymbol{\theta}_0}\right\|_{\mathcal{H}} + \sqrt{\tau/m} \right) \mathbb{E}_{\boldsymbol{\xi}} \left[ \sup_{\|\boldsymbol{w}\|_1 + \|\boldsymbol{u}\|_1 \leq 1} \left| \sum_{j=1}^{n} \xi_j \sigma(\boldsymbol{w}^\top \boldsymbol{x}_j(t) + \boldsymbol{u}^\top \boldsymbol{e}(t)) \right| \right],$$

where the $\{\xi_i\}_{i=1}^{n}$ are independent random variables with the distribution $\mathbb{P}(\xi_i = 1) = \mathbb{P}(\xi_i = -1) = 1/2$, and obviously,

$$\sup_{\|\boldsymbol{w}\|_1 + \|\boldsymbol{u}\|_1 \leq 1} \sum_{j=1}^{n} \xi_j \sigma(\boldsymbol{w}^\top \boldsymbol{x}_j(t) + \boldsymbol{u}^\top \boldsymbol{e}(t)) \geq \sum_{j=1}^{n} \xi_j \sigma(\boldsymbol{0}_d^\top \boldsymbol{x}_j(t) + \boldsymbol{0}_d^\top \boldsymbol{e}(t)) = 0,$$

we have

$$\sup_{\|\boldsymbol{w}\|_1 + \|\boldsymbol{u}\|_1 \leq 1} \left| \sum_{j=1}^{n} \xi_j \sigma(\boldsymbol{w}^\top \boldsymbol{x}_j(t) + \boldsymbol{u}^\top \boldsymbol{e}(t)) \right|$$

$$\leq \max \left\{ \sup_{\|\boldsymbol{w}\|_1 + \|\boldsymbol{u}\|_1 \leq 1} \sum_{j=1}^{n} \xi_j \sigma(\boldsymbol{w}^\top \boldsymbol{x}_j(t) + \boldsymbol{u}^\top \boldsymbol{e}(t)), \sup_{\|\boldsymbol{w}\|_1 + \|\boldsymbol{u}\|_1 \leq 1} \sum_{j=1}^{n} (-\xi_j) \sigma(\boldsymbol{w}^\top \boldsymbol{x}_j(t) + \boldsymbol{u}^\top \boldsymbol{e}(t)) \right\}$$

$$\leq \sup_{\|\boldsymbol{w}\|_1 + \|\boldsymbol{u}\|_1 \leq 1} \sum_{j=1}^{n} \xi_j \sigma(\boldsymbol{w}^\top \boldsymbol{x}_j(t) + \boldsymbol{u}^\top \boldsymbol{e}(t)) + \sup_{\|\boldsymbol{w}\|_1 + \|\boldsymbol{u}\|_1 \leq 1} \sum_{j=1}^{n} (-\xi_j) \sigma(\boldsymbol{w}^\top \boldsymbol{x}_j(t) + \boldsymbol{u}^\top \boldsymbol{e}(t)),$$

and by symmetry

$$\mathbb{E}_{\boldsymbol{\xi}} \left[ \sup_{\|\boldsymbol{w}\|_1 + \|\boldsymbol{u}\|_1 \leq 1} \left| \sum_{j=1}^{n} \xi_j \sigma(\boldsymbol{w}^\top \boldsymbol{x}_j(t) + \boldsymbol{u}^\top \boldsymbol{e}(t)) \right| \right]$$

$$\leq \mathbb{E}_{\boldsymbol{\xi}} \left[ \sup_{\|\boldsymbol{w}\|_1 + \|\boldsymbol{u}\|_1 \leq 1} \sum_{j=1}^{n} \xi_j \sigma(\boldsymbol{w}^\top \boldsymbol{x}_j(t) + \boldsymbol{u}^\top \boldsymbol{e}(t)) \right] + \mathbb{E}_{\boldsymbol{\xi}} \left[ \sup_{\|\boldsymbol{w}\|_1 + \|\boldsymbol{u}\|_1 \leq 1} \sum_{j=1}^{n} (-\xi_j) \sigma(\boldsymbol{w}^\top \boldsymbol{x}_j(t) + \boldsymbol{u}^\top \boldsymbol{e}(t)) \right]$$

$$= 2\mathbb{E}_{\boldsymbol{\xi}} \left[ \sup_{\|\boldsymbol{w}\|_1 + \|\boldsymbol{u}\|_1 \leq 1} \sum_{j=1}^{n} \xi_j \sigma(\boldsymbol{w}^\top \boldsymbol{x}_j(t) + \boldsymbol{u}^\top \boldsymbol{e}(t)) \right] = 2n\widehat{\mathrm{Rad}}_n(\mathcal{F}_{2,t}),$$

i.e., $\widehat{\mathrm{Rad}}_n(\mathcal{F}_{1,t}) \leq \widehat{\mathrm{Rad}}_n(\mathcal{F}'_{1,t}) \lesssim 2 \left( \left\|\boldsymbol{s}_{0,\boldsymbol{\theta}_0}\right\|_{\mathcal{H}} + \sqrt{\tau/m} \right) \widehat{\mathrm{Rad}}_n(\mathcal{F}_{2,t})$. According to Lemma 26.9 (contraction lemma) and Lemma 26.11 in [43], we have

$$\widehat{\mathrm{Rad}}_n(\mathcal{F}_{2,t}) \leq (\|\boldsymbol{x}(t)\|_\infty + \|\boldsymbol{e}(t)\|_\infty) \sqrt{\frac{2\log(4d)}{n}} \lesssim (C_{T,\delta} + C_{T,\boldsymbol{e}}) \sqrt{\frac{\log(d+1)}{n}}.$$

Combining above, we obtain

$$J_1 = \widehat{\mathrm{Rad}}_n(\mathcal{F}_t) \lesssim (C_{T,\delta} + C_{T,\boldsymbol{e}})^2 \left( \left\|\boldsymbol{s}_{0,\boldsymbol{\theta}_0}\right\|_{\mathcal{H}} + \sqrt{\tau/m} + 1 \right)^2 \sqrt{\frac{\log(d+1)}{n}}.$$

(ii) For $J_2$, by (36) and (37), we get

$$|\boldsymbol{f}(\boldsymbol{x}(t))| = \left| \boldsymbol{s}_{t,\boldsymbol{\theta}(\tau)}(\boldsymbol{x}(t)) - \nabla_{\boldsymbol{x}(t)} \log p_t(\boldsymbol{x}(t)) \right| \left| \sigma(\boldsymbol{w}^\top \boldsymbol{x}(t) + \boldsymbol{u}^\top \boldsymbol{e}(t)) \right|$$

$$\lesssim (C_{T,\delta} + C_{T,\boldsymbol{e}})^2 \left( \left\|\boldsymbol{s}_{0,\boldsymbol{\theta}_0}\right\|_{\mathcal{H}} + \sqrt{\tau/m} \right) + C'_{T,\delta} (C_{T,\delta} + C_{T,\boldsymbol{e}}),$$

which gives

$$J_2 \lesssim (C_{T,\delta} + C_{T,e})^2 \left( \|s_{0,\theta_0}\|_{\mathcal{H}} + \sqrt{\tau/m} + 1 \right) \sqrt{\frac{\log(1/\delta)}{n}}.$$

(iii) For $J_3$, we similarly have

$$\left| s_{t,\hat{\theta}_n(\tau)}(x(t)) \right| \lesssim (C_{T,\delta} + C_{T,e}) \left( \|s_{0,\theta_0}\|_{\mathcal{H}} + \sqrt{\tau/m} \right),$$

hence by (36) and (37), we get

$$J_3 \leq \mathbb{E}_{x(t) \sim p_t^{(n)}} \left| \left( s_{t,\theta(\tau)}(x(t)) - s_{t,\hat{\theta}_n(\tau)}(x(t)) \right) \sigma(w_i^\top x(t) + u_i^\top e(t)) \right|$$
$$\lesssim (C_{T,\delta} + C_{T,e})^2 \left( \|s_{0,\theta_0}\|_{\mathcal{H}} + \sqrt{\tau/m} \right).$$

Combining (i), (ii) and (iii), we obtain

$$\frac{1}{m} \sum_{i=1}^m \|\hat{a}_i(\tau) - a_i(\tau)\|_2^2 \lesssim \frac{1}{m^2} \sum_{i=1}^m \left\| \int_0^\tau \mathbb{E}_{t \sim \mathcal{U}(0,T)} \left[ \lambda(t)(J_1 + J_2 + J_3) \mathbf{1}_d \right] d\tau_0 \right\|_2^2$$
$$\lesssim \frac{1}{m^2} \sum_{i=1}^m \left\| \int_0^\tau (J_1 + J_2 + J_3) \mathbf{1}_d d\tau_0 \right\|_2^2$$
$$= (J_1 + J_2 + J_3)^2 \tau^2 \frac{d}{m}$$
$$\lesssim \tau^2 \frac{d}{m} (C_{T,\delta} + C_{T,e})^4 \left[ \left( \|s_{0,\theta_0}\|_{\mathcal{H}}^2 + \frac{\tau}{m} + 1 \right)^2 \right.$$
$$\left. \cdot \left( \sqrt{\frac{\log(d+1)}{n}} + \sqrt{\frac{\log(1/\delta)}{n}} \right)^2 + \left( \|s_{0,\theta_0}\|_{\mathcal{H}}^2 + \frac{\tau}{m} \right) \right],$$

which gives

$$I_1 = \tilde{\mathcal{L}}(\hat{\theta}_n(\tau)) - \tilde{\mathcal{L}}(\theta(\tau))$$
$$\lesssim (C_{T,\delta} + C_{T,e}) \left( \sqrt{\tilde{\mathcal{L}}(\theta^*)} + \frac{1}{\sqrt{\tau}} \left( \|s_{0,\theta_0}\|_{\mathcal{H}} + \|s_{0,\theta^*}\|_{\mathcal{H}} \right) \right) \left\{ \frac{1}{m} \sum_{i=1}^m \|\hat{a}_i(\tau) - a_i(\tau)\|_2^2 \right\}^{\frac{1}{2}}$$
$$+ (C_{T,\delta}^2 + C_{T,e}^2) \frac{1}{m} \sum_{i=1}^m \|\hat{a}_i(\tau) - a_i(\tau)\|_2^2$$
$$\lesssim (C_{T,\delta} + C_{T,e})^3 \left( \sqrt{\tilde{\mathcal{L}}(\theta^*)} + \frac{1}{\sqrt{\tau}} \left( \|s_{0,\theta_0}\|_{\mathcal{H}} + \|s_{0,\theta^*}\|_{\mathcal{H}} \right) \right)$$
$$\cdot \tau \sqrt{\frac{d}{m}} \left[ \left( \|s_{0,\theta_0}\|_{\mathcal{H}}^2 + \tau + 1 \right) \left( \sqrt{\frac{\log(d+1)}{n}} + \sqrt{\frac{\log(1/\delta)}{n}} \right) + \left( \|s_{0,\theta_0}\|_{\mathcal{H}} + \sqrt{\frac{\tau}{m}} \right) \right]$$
$$+ \tau^2 \frac{d}{m} (C_{T,\delta} + C_{T,e})^6 \left[ \left( \|s_{0,\theta_0}\|_{\mathcal{H}}^2 + \tau + 1 \right)^2 \left( \sqrt{\frac{\log(d+1)}{n}} + \sqrt{\frac{\log(1/\delta)}{n}} \right)^2 + \left( \|s_{0,\theta_0}\|_{\mathcal{H}}^2 + \frac{\tau}{m} \right) \right]$$
$$\lesssim (C_{T,\delta} + C_{T,e})^6 \tau \sqrt{\frac{d}{m}} \left[ (\tau + 1) \left( \sqrt{\frac{\log(d+1)}{n}} + \sqrt{\frac{\log(1/\delta)}{n}} \right) + \left( \frac{\|A_0\|_F}{\sqrt{m}} + \sqrt{\frac{\tau}{m}} \right) \right]$$
$$\cdot \left\{ \left( \sqrt{\tilde{\mathcal{L}}(\theta^*)} + \frac{1}{\sqrt{m\tau}} \left( \|A_0\|_F + \|A_*\|_F \right) \right) \right.$$
$$\left. + \tau \sqrt{\frac{d}{m}} \left[ (\tau + 1) \left( \sqrt{\frac{\log(d+1)}{n}} + \sqrt{\frac{\log(1/\delta)}{n}} \right) + \left( \frac{\|A_0\|_F}{\sqrt{m}} + \sqrt{\frac{\tau}{m}} \right) \right] \right\} \tag{38}$$

$$\lesssim \log^3(1/\delta^2)\tau\sqrt{\frac{d}{m}}\left[(\tau+1)\left(\sqrt{\frac{\log(d+1)}{n}}+\sqrt{\frac{\log(1/\delta)}{n}}\right)+\left(\frac{1}{\sqrt{m}}+\sqrt{\frac{\tau}{m}}\right)\right]$$
$$\cdot\left\{\left(\sqrt{\tilde{\mathcal{L}}(\boldsymbol{\theta}^*)}+\frac{1}{\sqrt{m\tau}}\right)+\tau\sqrt{\frac{d}{m}}\left[(\tau+1)\left(\sqrt{\frac{\log(d+1)}{n}}+\sqrt{\frac{\log(1/\delta)}{n}}\right)+\left(\frac{1}{\sqrt{m}}+\sqrt{\frac{\tau}{m}}\right)\right]\right\},$$

where $\lesssim$ hides universal positive constants only depending on $T$.

Combining all above estimates yields

$$D_{\mathrm{KL}}\left(p_0\|p_{0,\hat{\boldsymbol{\theta}}_n(\tau)}\right)$$
$$\lesssim I_1+I_2+I_3+D_{\mathrm{KL}}\left(p_T\|\pi\right)$$
$$\lesssim \log^3(1/\delta^2)\tau\sqrt{\frac{d}{m}}\left[(\tau+1)\left(\sqrt{\frac{\log(d+1)}{n}}+\sqrt{\frac{\log(1/\delta)}{n}}\right)+\frac{1}{\sqrt{m}}\left(1+\sqrt{\tau}\right)\right]$$
$$\cdot\left\{\left(\sqrt{\tilde{\mathcal{L}}(\boldsymbol{\theta}^*)}+\frac{1}{\sqrt{m\tau}}\right)+\tau\sqrt{\frac{d}{m}}\left[(\tau+1)\left(\sqrt{\frac{\log(d+1)}{n}}+\sqrt{\frac{\log(1/\delta)}{n}}\right)+\frac{1}{\sqrt{m}}\left(1+\sqrt{\tau}\right)\right]\right\}$$
$$+\left(\bar{\tilde{\mathcal{L}}}\left(\bar{\boldsymbol{\theta}}^*\right)+\tilde{\mathcal{L}}\left(\boldsymbol{\theta}^*\right)\right)+\frac{\log^2(1/\delta^2)}{m}d+\frac{1}{\tau}\left(\|\bar{\boldsymbol{s}}_{0,\bar{\boldsymbol{\theta}}_0}\|_{\mathcal{H}}^2+\|\bar{\boldsymbol{s}}_{0,\bar{\boldsymbol{\theta}}^*}\|_{\mathcal{H}}^2+\|\boldsymbol{s}_{0,\boldsymbol{\theta}_0}\|_{\mathcal{H}}^2+\|\boldsymbol{s}_{0,\boldsymbol{\theta}^*}\|_{\mathcal{H}}^2\right)+D_{\mathrm{KL}}\left(p_T\|\pi\right).$$

If one only focuses on the $(m,n)$-dependence, i.e. the dependence on the model capacity and sample size, this upper bound can be further simplified as

$$D_{\mathrm{KL}}\left(p_0\|p_{0,\hat{\boldsymbol{\theta}}_n(\tau)}\right)$$
$$\overset{\text{(vi)}}{\lesssim}\left(\frac{\tau^2}{\sqrt{mn}}+\frac{\tau\sqrt{\tau}}{m}\right)\cdot\left(\sqrt{\tilde{\mathcal{L}}(\boldsymbol{\theta}^*)}+\frac{1}{\sqrt{m\tau}}+\frac{\tau^2}{\sqrt{mn}}+\frac{\tau\sqrt{\tau}}{m}\right)$$
$$+\left(\bar{\tilde{\mathcal{L}}}\left(\bar{\boldsymbol{\theta}}^*\right)+\tilde{\mathcal{L}}\left(\boldsymbol{\theta}^*\right)\right)+\frac{1}{m}+\frac{1}{\tau}+D_{\mathrm{KL}}\left(p_T\|\pi\right)$$
$$\leq\left(\sqrt{\tilde{\mathcal{L}}(\boldsymbol{\theta}^*)}+\frac{1}{\sqrt{m\tau}}+\frac{\tau^2}{\sqrt{mn}}+\frac{\tau\sqrt{\tau}}{m}\right)^2+\left(\bar{\tilde{\mathcal{L}}}\left(\bar{\boldsymbol{\theta}}^*\right)+\tilde{\mathcal{L}}\left(\boldsymbol{\theta}^*\right)\right)+\frac{1}{m}+\frac{1}{\tau}+D_{\mathrm{KL}}\left(p_T\|\pi\right)$$
$$\lesssim\left(\tilde{\mathcal{L}}(\boldsymbol{\theta}^*)+\frac{1}{m\tau}+\frac{\tau^4}{mn}+\frac{\tau^3}{m^2}\right)+\left(\bar{\tilde{\mathcal{L}}}\left(\bar{\boldsymbol{\theta}}^*\right)+\tilde{\mathcal{L}}\left(\boldsymbol{\theta}^*\right)\right)+\frac{1}{m}+\frac{1}{\tau}+D_{\mathrm{KL}}\left(p_T\|\pi\right)$$
$$\lesssim\frac{\tau^4}{mn}+\frac{\tau^3}{m^2}+\frac{1}{\tau}+\frac{1}{m}+\left(\bar{\tilde{\mathcal{L}}}\left(\bar{\boldsymbol{\theta}}^*\right)+\tilde{\mathcal{L}}\left(\boldsymbol{\theta}^*\right)\right)+D_{\mathrm{KL}}\left(p_T\|\pi\right),$$

where we assume $\tau\geq 1$ for simplicity, and $\overset{\text{(vi)}}{\lesssim}$ hides the term $d\log(d+1)$, the polynomials of $\log(1/\delta^2)$ and finite RKHS norms $\|\cdot\|_{\mathcal{H}}$. The proof is completed. $\qquad\square$

**Approximation errors.** For the (universal) approximation, we discuss in two points:

- First, the approximation is a separate problem that can be analyzed independently of training and generalization. While it is beyond the scope of current work, we have included the approximation error in the final estimates. When the approximation by random feature models fails, the generalization error is supposed to be significant.

- In addition, the random feature model can approximate Lipschitz continuous functions on a compact domain (Theorem 6 in [17]). Notice that the forward diffusion process defines a random path $(\boldsymbol{x}(t),t)_{t\in[0,T]}$ contained in a rectangular domain $R_{T,\delta}:=[-C_{T,\delta},C_{T,\delta}]^d\times[0,T]\subset\mathbb{R}^{d+1}$ with $C_{T,\delta}:=C_T(C_{\boldsymbol{x}}+\sqrt{\log(1/\delta^2)})$ (use Lemma 1 and the boundness of inputs), one can apply Theorem 6 in [17] to bound (7) on the domain $R_{T,\delta}$ in $\mathbb{R}^{d+1}$ to obtain approximation results for Lipschitz continuous target score functions.

## A.2 Data-Dependent Generalization Gap

**Lemma 7** (Forward perturbation estimates, Gaussian mixtures). *Suppose that $x$ is sampled from a one-dimensional 2-mode Gaussian mixture: $p_0(x) = q_1 \mathcal{N}(x; -\mu, 1) + q_2 \mathcal{N}(x; \mu, 1)$, where $\mu > 0$, $q_1$, $q_2 > 0$ with $q_1 + q_2 = 1$ are all constants. Then for any $\delta > 0$, $\delta \ll 1$, $\mu > \sqrt{\log(1/\delta^2)}$, with the probability of at least $1 - \delta$, we have*

$$|x(t)| \lesssim C_T \left( \mu + \sqrt{\log(1/\delta^2)} \right) \triangleq C_{T,\mu,\delta}. \tag{39}$$

*Proof.* It is straightforward to verify that

$$\mathbb{P} \left( \{x : |x - \mu| \leq \sqrt{\log(1/\delta^2)}\} \cup \{x : |x + \mu| \leq \sqrt{\log(1/\delta^2)}\} \right)$$

$$= \mathbb{P}\{x : |x - \mu| \leq \sqrt{\log(1/\delta^2)}\} + \mathbb{P}\{x : |x + \mu| \leq \sqrt{\log(1/\delta^2)}\}$$

$$= \int_{\mu - \sqrt{\log(1/\delta^2)}}^{\mu + \sqrt{\log(1/\delta^2)}} p_0(x) dx + \int_{-\mu - \sqrt{\log(1/\delta^2)}}^{-\mu + \sqrt{\log(1/\delta^2)}} p_0(x) dx$$

$$\geq q_2 \int_{\mu - \sqrt{\log(1/\delta^2)}}^{\mu + \sqrt{\log(1/\delta^2)}} \mathcal{N}(x; \mu, 1) dx + q_1 \int_{-\mu - \sqrt{\log(1/\delta^2)}}^{-\mu + \sqrt{\log(1/\delta^2)}} \mathcal{N}(x; -\mu, 1) dx$$

$$= q_2 \int_{-\sqrt{\log(1/\delta^2)}}^{\sqrt{\log(1/\delta^2)}} \mathcal{N}(x; 0, 1) dx + q_1 \int_{-\sqrt{\log(1/\delta^2)}}^{\sqrt{\log(1/\delta^2)}} \mathcal{N}(x; 0, 1) dx$$

$$= (q_1 + q_2) \cdot \mathbb{P}\{\epsilon : |\epsilon| \leq \sqrt{\log(1/\delta^2)}\} \geq 1 - \delta,$$

where the last inequality applies (21). That is, for any $\delta > 0$, $\delta \ll 1$, with the probability of at least $1 - \delta$, we have

$$|x| \in [\mu - \sqrt{\log(1/\delta^2)}, \mu + \sqrt{\log(1/\delta^2)}] = \Theta(\mu). \tag{40}$$

Hence, Lemma 1 gives

$$|x(t)| \lesssim C_T \left( \mu + \sqrt{\log(1/\delta^2)} \right), \tag{41}$$

which completes the proof. $\qquad \square$

**Lemma 8** (Monte Carlo estimates, Gaussian mixtures). *Define the Monte Carlo error*

$$\mathrm{Err}_{\mathrm{MC}} = \mathrm{Err}_{\mathrm{MC}}(\boldsymbol{\theta}, \bar{\boldsymbol{\theta}}; T, \lambda(\cdot)) := \mathbb{E}_{t \sim \mathcal{U}(0,T)} \left[ \lambda(t) \cdot \mathbb{E}_{x(t) \sim p_t} \left[ \|s_{t,\boldsymbol{\theta}}(x(t)) - \bar{s}_{t,\bar{\boldsymbol{\theta}}}(x(t))\|_2^2 \right] \right]. \tag{42}$$

*Suppose that the trainable parameter $\boldsymbol{a}$ and embedding function $\boldsymbol{e}(\cdot)$ are both bounded, and $x$ is sampled from a one-dimensional 2-mode Gaussian mixture: $p_0(x) = q_1 \mathcal{N}(x; -\mu, 1) + q_2 \mathcal{N}(x; \mu, 1)$, where $\mu > 0$, $q_1$, $q_2 > 0$ with $q_1 + q_2 = 1$ are all constants. Then, given any $\boldsymbol{\theta}$, for any $\delta > 0$, $\delta \ll 1$, $\mu > \sqrt{\log(1/\delta^2)}$, with the probability of at least $1 - \delta$, there exists $\boldsymbol{\theta}$ such that*

$$\mathrm{Err}_{\mathrm{MC}} \lesssim \mu^2 \frac{\log(1/\delta)}{m}, \tag{43}$$

*where $\lesssim$ hides universal positive constants only depending on $T$.*

*Proof.* According to Lemma 7, we just need to follow the proof of Lemma 6 by replacing $C_{T,\delta}$ by $C_{T,\mu,\delta}$. Notably, based on (33) in the proof of Lemma 6, one can finally derive

$$\mathrm{Err}_{\mathrm{MC}} \lesssim \mu^2 \frac{\log(1/\delta)}{m},$$

which gives the desired estimates. $\qquad \square$

*Proof of Theorem 2.* We decompose the loss $\tilde{\mathcal{L}}(\hat{\boldsymbol{\theta}}_n(\tau))$ in the same way as in the proof of Theorem 1. In fact, Theorem 2 is similarly proved by replacing $C_{T,\delta}$ in the proof of Theorem 1 by $C_{T,\mu,\delta}$. Note that $C_{T,\mu,\delta} = C_T \left( \mu + \sqrt{\log(1/\delta^2)} \right) \lesssim \mu$ for $\mu \gg 1$, the proof is completed. $\qquad \square$

**Remark 6.** *A standard variance is used here for convenience. In general, if var $= o(\mu)$ as $\mu \to +\infty$ (e.g. a bounded var), similar analysis and results are supposed to hold. However, this is different when var $= \Theta(\mu)$ as $\mu \to +\infty$, since the modes are not separated in this case, and we can not characterize modes shift by simply varying $\mu$.*

**Remark 7.** *Simply scaling down inputs seems not to resolve the adverse effect of modes shift, since the ground truth $\mu$ is unknown. One can use the input scale to approximate $\mu$ on toy datasets, but it is not that trivial in practice, particularly for real-world applications with multiple high-dimensional modes in varied scales.*

**The model-target inconsistency.** Informally, there is inconsistency between the score network model and target score function. In fact, given the target Gaussian mixture $p_0(x) = q_1 \mathcal{N}(x; -\mu, 1) + q_2 \mathcal{N}(x; \mu, 1)$, the target score function is

$$
s_0(x) = -x + \frac{q_2 \mathcal{N}(x; \mu, 1) - q_1 \mathcal{N}(x; -\mu, 1)}{q_2 \mathcal{N}(x; \mu, 1) + q_1 \mathcal{N}(x; -\mu, 1)} \mu,
$$

which gives $s_0(x) \approx -x + \mu$, $x \geq 0$ and $s_0(x) \approx -x - \mu$, $x \leq 0$. While the score network model is

$$
s_{0,\theta}(x) \approx \left( \frac{1}{m} \sum_{i:w_i > 0} a_i w_i \right) x + \left( \frac{1}{m} \sum_{i:w_i > 0} a_i \boldsymbol{u}_i^\top \boldsymbol{e}(0) \right),
$$

where "$\approx$" holds since $|x| = \Theta(\mu)$ by (40) ($\mu \gg 1$). Both $s_0$ and $s_{0,\theta}$ are linear functions, but they have unmatched scales in slopes and intercepts: $O(1)$ and $O(\mu)$ for $s_0$, but both $O(a)$'s for $s_{0,\theta}$. That is, modeling Gaussian mixtures with large modes distances as random feature models is inconsistent.

**Remark 8.** *Recall that the goal of this work is to estimate $D_{\mathrm{KL}}\left(p_0 \| p_{0,\hat{\boldsymbol{\theta}}_n(\tau)}\right)$, which aims at the density estimation characterization. This is different from [32], where the equivalence of learned and target data manifolds (support sets) is studied.*

# B  Additional Experiments

## B.1  Early-Stopping Generalization

We illustrate the early-stopping generalization gap established in Theorem 1 using the Adam optimizer. All the configurations remain the same as Section 4.1 except that the learning rate is now $10^{-3}$.

From Figure 8, one can observe that the KL divergence achieves its minimum at the 1000th training epoch, and it starts to increase after this turning point. The plot is aligned with Theorem 1, which states that there exists an optimal early-stopping time when the model can generalize well, indicating the effectiveness of the upper bound. Further, the KL divergence begins to oscillate after the minimum point (1000th training epoch), which may suggest a phase transition in the KL divergence dynamics, and the transition point is around the optimal early-stopping time. The finding is aligned with SGD setting.

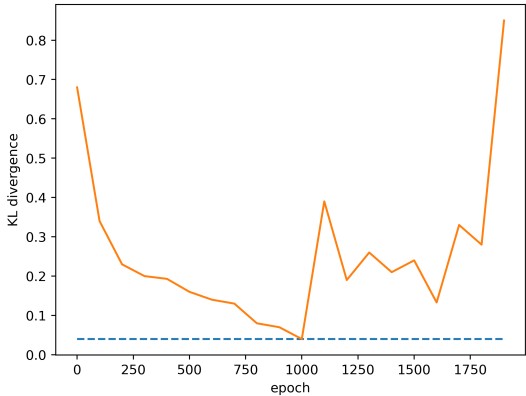

Figure 8: The KL divergence dynamics under the Adam optimizer.

## B.2 Modes Shift Effect

We further test the relationship between the modes' distance and the generalization performance using the Adam optimizer under the same configurations.

In Figure 9 and Figure 10, it is shown that the training of modeled distributions exhibits the same two-stage dynamics as the SGD setting, indicating that *the modes shift effect holds not particularly for a certain optimizer*.

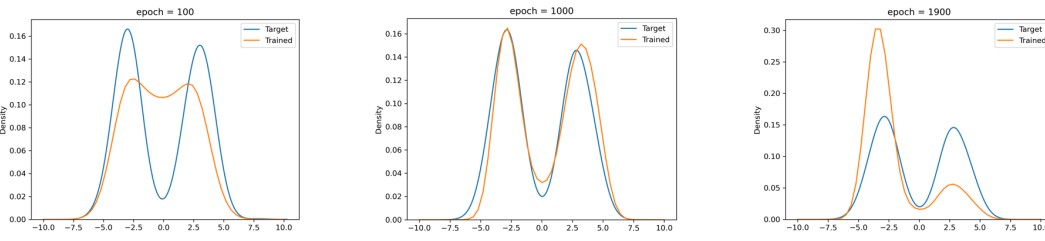

Figure 9: The Adam training dynamics when the distance between two modes is 6 ($\mu = 3$).

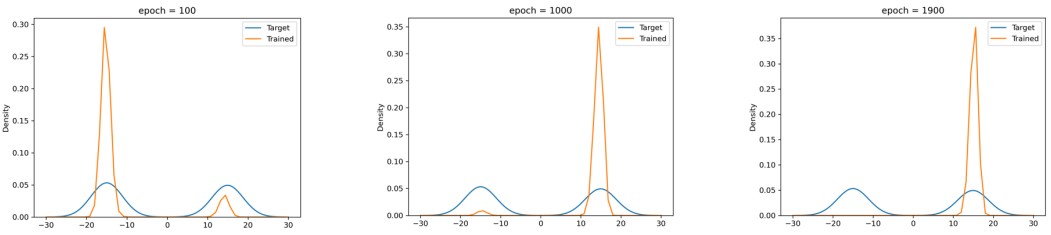

Figure 10: The Adam training dynamics when the distance between two modes is 30 ($\mu = 15$).

## B.3 Model Capacity Dependency

We also numerically study the dependency of generalization on the model capacity. Following the same configurations in Section 4.1.1, we conduct experiments for different hidden dimensions varying from $2^1$ to $2^{11}$.

In Figure 11, the left plot shows the KL divergence from the trained distribution at the 1000th training epoch to the target distribution, and the right plot shows the time duration of the modeled distribution to generalize. Here, the generalization criterion we select is $D_{\mathrm{KL}} \leq 10^{-1}$, and we stop training at epoch $= 10000$. Both the two plots in Figure 11 indicate that increasing model capacity benefits the generalization, which also verifies the corresponding theoretical results ($m$-dependency in Theorem 1 and Theorem 2).

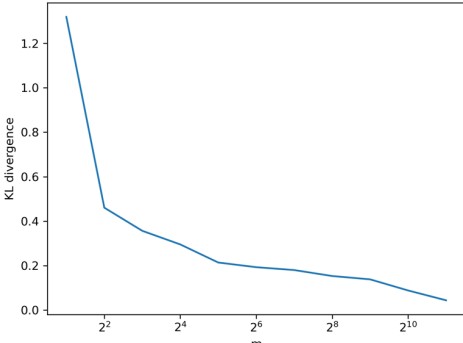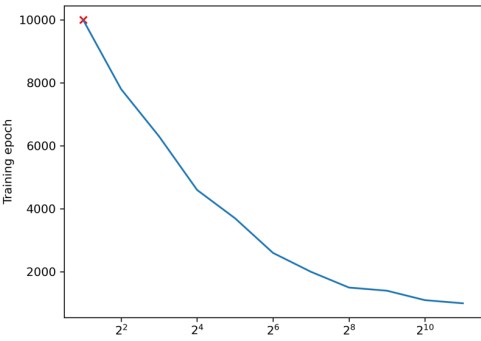

Figure 11: Left: The KL divergence from the model trained after 1000 epochs to the target distribution under different hidden dimensions. Right: The earliest training epoch when $D_{\mathrm{KL}} \leq 10^{-1}$. Here, "$\times$" means that the model does not generalize when the training is stopped at the maximum training epoch 10000.

