# OpenReview forum: "On the Generalization Properties of Diffusion Models"
_NeurIPS.cc/2023/Conference — NeurIPS 2023 poster_

### Official Review · Reviewer_uaj6 · 2023-07-03

**Soundness:** 3 good
**Presentation:** 3 good
**Contribution:** 3 good
**Rating:** 7
**Confidence:** 3

**Summary:**

This paper studies theoretical bounds on score-matching diffusion models' generalization ability, in terms of KL divergence between the true distribution and the learned distribution generated by the model. It is assumed that the score model is parametrized as a time-dependent (time refers to $t$ in the diffusion process) 2-layer random feature model, and that the training process is the continuous-time gradient flow with respect to the score matching loss $\tilde{\mathcal{L}}$. The authors characterize the generalization bound as a decreasing function of $n$ (the sample size) and $m$ (the model capacity; the hidden layer size), and shows that the error decreases polynomially in $n$ with appropriately chosen early-stopping time $\tau$, under the setup where the target distribution has compact support. They also study the effect of distance between the modes on the theoretical bound in the toy case of fitting a 2-Gaussian-mixture.

**Strengths:**

- The paper successfully builds the setup where the generalization of diffusion models is theoretically analyzable in terms of sample size, model capacity and training time, using common machineries used in ML theory.
- The analysis carefully deals with the necessary technical components for bounding the generalization error for diffusion models and the proofs seem sound (legitimate arguments assigned to each component of the analysis).
- Overall, I think the paper provides a good starting point for the theoretical study on the generalization of diffusion models and identifies some interesting questions on diffusion model training such as dimension dependency or the choice of optimal early stopping time.

**Weaknesses:**

- Of course, the investigated setup is limited (dealing with 2-layer random feature models) and is far away from what is being used in practice. To some extent this is understandable, considering that the similar limitation is shared among most of the relevant works in the field. However, one can still argue that the setup is obsolete as there are now more 'modern' tools like the neural tangent kernels or mean field limits for performing theoretical analyses on neural networks.
- I think the paper has insufficient discussion regarding the optimal solution $\bar{\boldsymbol{\theta}}^\ast$. It does not discuss, e.g., the universal approximation property of the random feature parametrization with respect to the score-matching loss $\tilde{\mathcal{L}}$ which is a sort of modified $L^2$ norm. So it is confusing whether the authors are assuming that the true score function is representable as a continuous random feature model not (so, if $\bar{\tilde{\mathcal{L}}}(\bar{\boldsymbol{\theta}}^\ast)=0$ or not); the term $\bar{\tilde{\mathcal{L}}}(\bar{\boldsymbol{\theta}}^\ast)$ is present in Theorems 1, 2 while absent in Corollaries 1, 2. It is also confusing whether $\mathcal{H}$ in Theorem 1 coincides with the RKHS induced by $k_{\rho_{0}}$.
- The value of Section 3.2.2 is unclear to me. It deals only with the toy case of 1D Gaussian mixture with two modes. Additionally, it provides an upper bound that is adversely affected by $\mu$, but there is no guarantee of tightness so it does not sufficiently "explain" the modes shift effect (to do so, one might rather need some sort of lower bound).
- Experiments are minimal and loosely designed. It is performed over a single toy case (1D 2-Gaussian-mixture) and it is not reported how many repetitions were tried and whether the results were consistent. Also, it is inconsistent with the theory, which deals with the continuous-time gradient flow. Why use Adam optimizer for the experiments?

**Questions:**

- What is the role of the time embedding function $\mathbf{e}$? Does it have any meaningful effect on the analysis?
- In Lemma 3, the authors use the convexity of the diffusion loss $\bar{\tilde{\mathcal{L}}}$ with respect to the trainable parameter $\bar{\mathbf{a}}$, and maybe this is why they consider the random feature model. But won't similar analyses be possible in more general setups using e.g., neural tangent kernels?
- Please clarify the second point from the "Weaknesses" section.
- In Theorem 2, how does variance of the Gaussian distributions comprising the mixture affect the bound? If the variance is not important, does simply scaling down all the data resolve the adverse modes shift effect claimed in Section 3.2.2?
- In the equation right before Line 511 in the appendix, is the minus sign missing in front of $\frac{d}{d\tau} \tilde{\mathcal{L}}(\boldsymbol{\theta}(s))$? I think the subsequent lines using the integral Cauchy-Schwarz inequality also share the sign issue.
- In the experiments, do the authors observe that their theory provide quantitative predictions on the optimal early-stopping time (the "transition point" in the training dynamics), or is it just a post-hoc observation? Were the results consistent over multiple runs?

---

> ### Author Rebuttal · Authors · 2023-08-09
>
> # Response to Reviewer uaj6
>
> Thank you for your comprehensive review and valuable feedback to help improve the paper. We detail our response below, and please kindly let us know whether our response addresses your concerns.
>
> ---
>
> **Weakness 1**:  There are two points regarding the setup concern:
>
> - First, we choose the random feature model (RFM) as the score network because it is simple to analyze, and still possesses universal approximation capability ([1]). We want to emphasize the insights gained from diffusion models rather than the neural network parametrization, which is relatively more developed in both theory and practice.
>
> - In addition, while the mathematical tools such as NTKs and mean fields may seem modern, they are more complex and require the infinite-width regime to achieve meaningful results, which is less practical in real-world applications. Nevertheless, employing these modern tools in the future work is valuable at least for theoretical completeness.
>
> ---
>
> **Weakness 2**: $\||\cdot\||_\mathcal{H}:=\||\cdot\||\_{\mathcal{H}\_{k\_{\rho\_0}}}$ (refer to **A3** in **Response to Reviewer ab6x** for more details). For the universal approximation, we discuss in two points:
>
> - First, the approximation is a separate problem that can be analyzed independently of training and generalization. While it is beyond the scope of current work, we have included the approximation error in the final estimates. When approximation by RFMs fails, the generalization error is supposed to be significant.
>
> - In addition, the RFM can approximate Lipschitz continuous functions on a compact domain (Theorem 6 in [1]). Notice that the forward diffusion process defines a random path $(\pmb{x}(t),t)\_{t \in [0,T]}$ contained in a rectangular domain $R_{T,\delta}:=[-C\_{T, \delta}, C\_{T, \delta}]^d \times [0,T] \subset \mathbb{R}^{d+1}$ with $C\_{T, \delta}:=C\_T (C_{\pmb{x}} + \sqrt{\log (1/\delta^2)})$ (use Lemma 1 and the boundness of inputs), one can apply Theorem 6 in [1] to bound Eq. (7) on the domain $R_{T,\delta}$ in $\mathbb{R}^{d+1}$ to obtain approximation results.
>
> ---
>
> **Weakness 3**: Three points:
>
> - Informally, there may be inconsistency between the score network model and target score function. In fact, given the target Gaussian mixture $p_0(x) = q_1 \mathcal{N}(x; -\mu, 1) + q_2 \mathcal{N}(x; \mu, 1)$, the score function is
> $$s_0(x)=-x+\frac{q_2 \mathcal{N}(x; \mu, 1)-q_1 \mathcal{N}(x; -\mu, 1)}{q_2 \mathcal{N}(x; \mu, 1)+q_1 \mathcal{N}(x; -\mu, 1)}\mu,
> $$
> which gives
> $s_0(x) \approx -x+\mu$, $x\ge 0$ and $s_0(x) \approx -x-\mu$, $x\le 0$. While the score network is
> $$
> s_{0,\theta}(x)
> \approx \left(\frac{1}{m} \sum\_{i: w\_i>0} a_i w_i\right) x+\left(\frac{1}{m} \sum\_{i: w\_i>0} a_i\pmb{u}_i^{\top} \pmb{e}(0)\right),
> $$
> where "$\approx$" holds since $x=\Theta(\mu)$ (Eq. (38)). Both $s_0$ and $s\_{0,\theta}$ are linear functions, but they have unmatched scales in slopes and intercepts: $O(1)$ and $O(\mu)$ for $s_0$, but both $O(a)$'s for $s\_{0,\theta}$. That is, modeling Gaussian mixtures with large modes distances as RFMs may be inconsistent (a rigorous lower bound is needed in the future).
>
> - Our experimental results (Figures 3 and 4) demonstrate a significant gap in the density estimation performance for different modes shifts.
>
> - Our new simulation on MNIST with the U-net score network (refer to **Point 1** in **Summary of new results (as required)** and **Figure 1** in the **pdf attachment** for details) suggests that modes shift may adversely affect the performance of diffusion models in general.
>
> ---
>
> **Weakness 4**: The experiment is strengthened on a real dataset (see above). For consistency, we reproduce original figures trained with (S)GD for many repetitions, including Figure 2 (studying the KL divergence dynamics), and Figure 3, 4 (studying the modes shift effect). Refer to **Figure 2, 3, 4** in the **pdf attachment**. It is shown that our experimental results are consistent with the theory and replicated over multiple runs. A different optimizer (e.g. Adam) is used only for faster training.
>
> ---
>
> **Question 1**: The embedding function $\pmb{e}(t)$ is required to be bounded on a compact interval, and hence has no meaningful effect on the analysis.
>
> ---
>
> **Question 2**: The extension to NTKs is possible, since the training of RFMs follows a specific NTK regime (only the last layer is updated) in the output space (instead of parameter space). We leave these as the future work.
>
> ---
>
> **Question 3**: Please refer to the response to **Weakness 2**.
>
> ---
>
> **Question 4**: A standard variance is used now for convenience. In general, if $\text{var} = o(\mu)$ as $\mu \to +\infty$ (e.g. a bounded $\text{var}$), similar analysis and results are supposed to hold. However, this is different when $\text{var} = \Theta(\mu)$ as $\mu \to +\infty$, since the modes are not *separated* in this case, and we can not characterize modes shift by simply varying $\mu$.
>
> Simply scaling down inputs seems not to resolve the adverse modes shift effect, since the ground truth $\mu$ is unknown. One may use the input scale to approximate $\mu$ on toy datasets, but it is not that trivial in practice, particularly for real-world problems with multiple high-dimensional modes in varied scales.
>
> ---
>
> **Question 5**: The typo is fixed.
>
> ---
>
> **Question 6**: Currently, it is difficult to predict the optimal early-stopping time quantitatively due to upper bound estimates and unknown universal constants. However, the reproduced plot on the KL divergence dynamics shows consistency in transition points (refer to **Figure 2** in the **pdf attachment**). For other consistency concerns, refer to the response to **Weakness 4**.
>
> ---
>
> **References**
>
> [1] Daniel Hsu, Clayton H Sanford, Rocco Servedio, Emmanouil Vasileios Vlatakis-Gkaragkounis. On the approximation power of two-layer networks of random ReLUs. *Proceedings of Thirty Fourth Conference on Learning Theory*, PMLR 134:2423-2461, 2021.
>
> ---

---

> > ### Comment · Reviewer_uaj6 · 2023-08-17
> > **A remaining question on specifying the role of approximation error**
> >
> > I have read the rebuttal, and I thank the authors for the detailed response, including additional experiments to appropriately support the theory. I find most of the responses adequate. One remaining question is: although I acknowledge that universal approximation may be a separate problem, in the statements of Corollary 1 and Corollary 2, the bound (right hand side of $\lesssim$) only involves $(\log(d+1)/n)^{\frac{1}{4}}$ and lacks the term $\bar{\tilde{\mathcal{L}}}(\bar{\boldsymbol{\theta}}^*)$ (and also the $m$-dependent term and KL divergence term between $p^T$ and $\pi$).
> > This is confusing, and it is the reason why I initially thought the authors are implicitly assuming universal approximation and infinite width, etc..
> >
> > Therefore, unless I am missing something, I insist that the authors should carefully clarify what they are omitting in Corollaries 1 and 2, and precisely under which conditions/assumptions one can expect the *KL divergence between $p^0$ and the generated $p^{0, \hat{\boldsymbol{\theta}}_n (\tau^{es})}$ actually goes to zero, polynomially in $n$*.

---

> > > ### Author Response · Authors · 2023-08-17
> > > **Further discussion on error bounds**
> > >
> > > Thanks for your detailed suggestion. We agree with you, and we omit other errors in the original version only to highlight the early-stopping estimates. Based on your suggestion, we plan to replace Corollary 1 (and 2) by a corresponding paragraph named "Discussion on error bounds" in the updated version, to discuss conditions under which the error terms in Theorem 1 (and 2) are negligible:
> > > - The first two terms: use original contents in Corollary 1 (and 2), which derive the early-stopping times and corresponding errors. They are core terms and we will highlight them as before.
> > > - The 3rd term: $m$-dependent, which is $o(1)$ when $m \gg 1$.
> > > - The 4th term: approximation error. We will include contents in above response (refer to the response to Weakness 2, Point 2) and the above-mentioned reference [1].
> > > - The 5th term: KL divergence between $p_T$ and $\pi$. We cite a classical result in e.g. [2] (Theorem 3.20, Theorem 3.24 and Remark 3.26), which states that if $\pi$ satisfies the log-Sobolev inequality (e.g. $\pi$ is a Gaussian density), the KL divergence between $p_T$ and $\pi$ is exponentially small in $T$.
> > >
> > > Note that the conditions for 4th and 5th error term (Lipschitz continuous target score functions and log-Sobolev stationary distribution, respectively) are sufficient but *not necessary*, we only list the most common conditions for demonstration.
> > >
> > > ---
> > >
> > > **References**
> > >
> > > [2] Ramon van Handel. Probability in high dimension. Technical report, Princeton University, 2014.

---

> > > > ### Comment · Reviewer_uaj6 · 2023-08-18
> > > >
> > > > I appreciate the authors' effort to clarify my final questions. Given that the paper has solid results that could serve as a basis for future theoretical studies on the generalization of diffusion models, I think it deserves publication once the author-reviewer discussion and the additional materials provided by the authors during the rebuttal period gets appropriately blended into revision. I decide to increase my rating from 6 to 7.

---

> > > > > ### Author Response · Authors · 2023-08-18
> > > > > **Thank you**
> > > > >
> > > > > Thank you for your reply! We are happy to see that our response addressed your concerns. Thanks again for your valuable feedback to help us improve our paper and for increasing your rating.

---

### Official Review · Reviewer_WQ2A · 2023-07-04

**Soundness:** 3 good
**Presentation:** 3 good
**Contribution:** 2 fair
**Rating:** 6
**Confidence:** 3

**Summary:**

This paper provides some generalization bounds for Diffusion Models when seen as score-based models. These bounds are similar to previous literature for GANs and consider two scenarios. In the first one, the target distribution has compact support, and in the second case, it corresponds to a one-dimensional 2-mode Gaussian mixture.


**Strengths:**

This paper is well-written and provides some generalization bounds on a general and toy problem and a numerical experiment. The sketches of the proofs are well detailed and explained.


**Weaknesses:**

Some of the motivations of the paper can seem obscure and a bit misleading. For instance, from the introduction, it feels that the authors will provide a general result for mode shift distribution while they only provide a theorem for the case of a 1-dimensional mixture of 2 Gaussian. It seems fine to study that toy problem, but it should be advertised as is. Also, the first two paragraphs on Page 2 were a bit obscure.

**Questions:**

"early-stopping estimates" better generalize, do you have a practical approach to early stopping here?

**Limitations:**

yes

---

> ### Author Rebuttal · Authors · 2023-08-08
>
> # Response to Reviewer WQ2A
>
> Thank you for your comprehensive review and your valuable feedback to help us improve the paper. We detail our response below, and please kindly let us know whether our response addresses your concerns.
>
> ---
>
> > **Q1**: Some of the motivations of the paper can seem obscure and a bit misleading. For instance, from the introduction, it feels that the authors will provide a general result for mode shift distribution while they only provide a theorem for the case of a 1-dimensional mixture of 2 Gaussian. It seems fine to study that toy problem, but it should be advertised as is. Also, the first two paragraphs on Page 2 were a bit obscure.
>
> **A1**: We answer the questions on introduction as follows:
>
> - For the analysis on modes shift distributions, we indeed investigate a specific setting in theory, with additional experiments on density estimation comparison (Figure 3 and Figure 4). Notably, we provide a new simulation result on a real dataset (refer to **Point 1** in **Summary of new results (as required)** and  **Figure 1** in the **pdf attachment** for more details), which suggests that the adverse effect of modes shift on the performance of diffusion models may appear in general. But anyway, this is still not a general rigorous mathematical analysis on the practical setting, and we will follow your advice to tone down a bit in the corresponding statements. We plan to modify lines 65-67 to:
> "This "uniform" bound is further extended to a *data-dependent* setting, where a sequence of unidimensional Gaussian mixtures distributions with an increasing modes' distance $\mu$ is considered as the ground truth.
> This result characterizes the effect of ``modes shift'' quantitatively, which implies that..." in the final version.
>
> - The first two paragraphs on Page 2 mainly conveys the information that generative models may possess the memorization property, which has been already proved for bias potential models and GANs in theory and shown for large language models in applications, leading to potential privacy and copyright risks. This is our motivation to investigate the generalization properties of diffusion models. To clarify, we plan to modify these two paragraphs to the following.
>     - "In theory, the generalization issues of generative modeling (or learning for distributions) may exhibit as the *memorization* phenomenon, if the modeled distribution is eventually trained to converge to the empirical distribution only associated with training samples. Intuitively, memorization arises from two reasons: (i) it is useful for the hypothesis space to be large enough to approximate highly complex underlying target distributions (universal convergence; [59]); (ii) the underlying distribution is unknown in practice, and one can only use a dataset with finite samples drawn from the target distribution. Rigorous mathematical characterizations of memorization are developed for bias potential models and GANs in [60] and [61], respectively. A natural question is, does a similar phenomenon occur for diffusion models? To answer this, a thorough investigation of generalization properties for DMs/SGMs is required."
>     - "In practice, the generalization capability of diffusion models is also an essential requirement, as the memorization can lead to potential privacy and copyright risks when models are deployed. Similar to other generative models and large language models (LLMs) ([5, 64, 17, 6]), diffusion models can also memorize and leak training samples ([5, 42]), hence can be subsequently attacked using specific procedures and algorithms ([26, 16, 56]). Although there are defense methods developed to meet privacy and copyright standards ([10, 13, 55 53]), these approaches are often heuristic, without providing sufficient quantitative understandings particularly on diffusion models. Therefore, a comprehensive investigation of the generalization foundation of diffusion models, including both theoretical and empirical aspects, is of utmost importance in improving principled tutorial guidance in practice."
>
> ---
>
> > **Q2**: "Early-stopping estimates" better generalize, do you have a practical approach to early stopping here?
>
> **A2**: It is a common practice to use the test error to evaluate the generalization performance. For diffusion models, a straightforward approach is to compute the negative log-likelihood (averaged in bits/dim) on the test dataset *during training* with the instantaneous change-of-variable formula ([1]) and probability flow ODE Eq. (4), where the true score function $\nabla\_{\pmb{x}} \log p\_{t}(\pmb{x})$ is replaced by $\pmb{s}\_{t, \pmb{\theta}(\tau)}(\pmb{x})$.
>
> ---
>
> **References**
>
> [1] Ricky T. Q. Chen, Yulia Rubanova, Jesse Bettencourt, David K. Duvenaud. Neural ordinary
> differential equations. In *Advances in Neural Information Processing Systems*, pp. 6571–6583,
> 2018.
>
> ---

---

### Official Review · Reviewer_4Zu3 · 2023-07-05

**Soundness:** 2 fair
**Presentation:** 1 poor
**Contribution:** 2 fair
**Rating:** 4
**Confidence:** 3

**Summary:**

The paper aims to characterize the generalization gap of score-based diffusion models. The setting is as follows. The input is $n$ samples from some unknown distribution $p_0$, and the goal is to approximate $p_0$ by a distribution $p_{0,\hat{\theta}_n}$
where the parameter $\hat{\theta}_n$  is a minimizer of the empirical loss function obtained by running gradient descent up to some time $\tau.$

The loss function is the expected difference between the gradients of the log density function of distributions along the trajectories of 2 SDEs (see equation (2)), one initialized at $p_0$ and the other at $p_{0,\hat{\theta}_n}.$

The paper proves a bound on the KL divergence between the true distribution $p_0$ and the learned distribution $p_{0,\hat{\theta}_n}$ in terms of the training time $\tau$ for gradient descent, the sample size $n$ and model parameter $m.$ The bound is obtained by using [Song, Durkan, Murray, Ermon’21]’s main result to bound the aforementioned KL divergence in terms of the empirical loss plus a negligible term, then bound the empirical loss by the population loss. The paper states a general result (theorem 1), then shows how to apply it to the specific case when the target distribution is a mixture of two 1-dimensional Gaussians with the same covariance. They quantitatively show that the error grows when the distance between the two modes of the Gaussians increases, the KL divergence between the true distribution and the learned one becomes worse, which matches the qualitative result given in [Koehler, Heckett, Risteski—ICLR’23].

**Strengths:**

The paper explicitly bounds the generalization error, which is defined as the KL divergence between the true distribution and the learned one, in the model parameters and the sample size. Similar bounds are shown before by [Koehler, Heckett, Risteski—ICLR’23] but this previous work doesn’t make explicit the dependency on the sample size and the model parameters, and further requires that the learned distribution. The results appear to be interesting, if correct.

**Weaknesses:**

The paper might be incorrect, or at least currently doesn’t do a good job of convincing me otherwise. I believe the statement of theorem 1/theorem 3 in the appendix lacks the important assumption that the loss function is convex (see the top line of page 15 of the supplement (proof of Lemma 3) and my 1st question for details). However, the loss function of mixtures of Gaussians appears non-convex (or at least it’s unclear to me why it should be so), so lemma 3 couldn’t be used in the proof of theorem 2 as the paper currently does.
Aside from that, the paper is quite hard to understand, and the definition of many terms are not easy to find. For ease of understanding, the author should make clear the difference between $T$, the time to run the diffusion models so that $p_T$ is close to the known prior distribution $\pi$ (e.g. Gaussian), and $\tau,$ the time to train the neural net for gradient estimation. These definitions, along with the definition of $p_T$ and $\pi$, should be clearly stated in the statement of theorem 1. The preconditions of the main theorem (theorem 1) should be stated upfront in the main text instead of in the appendix of the supplement.

**Questions:**

1/ In proof of lemma 3, the top line of page 15 of the supplement, does “by convexity” refer to the fact that the loss function is a convex function? If so, this assumption should be clearly stated as part of theorem 1 or theorem 3.
2/ In the neural net setup, lines 156 and 157 of the supplement, does A essentially play the role of theta? If so, this should be clearly stated.
3/ If lemma 3 requires the assumption that the loss function is convex, then can the authors justify that the loss function for Gaussian mixtures is convex? Else, theorem 2 cannot use lemma 3 like it currently does.

**Limitations:**

The paper's proof might be incorrect.

---

> ### Author Rebuttal · Authors · 2023-08-08
>
> # Response to Reviewer 4Zu3
>
> Thank you for your detailed review and feedback to help us improve the paper. We detail our response below, and please kindly let us know whether our response addresses your concerns.
>
> ---
>
> > **Weakness 1**: Convexity concern.
>
> **A1**: The convexity arises from the following two points:
>
> 1. We analyze the score-matching loss Eq. (7), which is a $L^2$-metric between the score network model and target score function;
>
> 2. The score network is defined as a random feature model Eq. (12) that is linear to trainable parameters.
>
> Therefore, using some trace techniques, it is not hard to derive the fact that the loss is quadratic and hence convex w.r.t. trainable parameters $\text{vec}(\pmb{A})$ (i.e. the vectorization of $\pmb{A}$).
>
> Concretely,  recall $\pmb{s}\_{t, \pmb{\theta}}(\pmb{x}(t)) = \pmb{A} \sigma(\pmb{W}\pmb{x}(t) + \pmb{U} \pmb{e}(t))$, and let $\pmb{s}\_{t}(\pmb{x}(t)):=\nabla\_{\pmb{x}(t)} \log p\_{t}(\pmb{x}(t))$, $\pmb{h}_1(\pmb{x},t):=\sqrt{\lambda (t)}\sigma(\pmb{W}\pmb{x} + \pmb{U} \pmb{e}(t))$,  $\pmb{h}_2(\pmb{x},t):=\sqrt{\lambda (t)}\pmb{s}\_{t}(\pmb{x})$, we have
> $$
> \tilde{\mathcal{L}} (\pmb{\theta};\lambda(\cdot)) =
>     \mathbb{E}\_{t \sim \mathcal{U} (0,T)}
>     \mathbb{E}\_{\pmb{x}(t) \sim p\_{t}}
>     [\pmb{h}_1^{\top}(\pmb{x}(t),t)\pmb{A}^{\top} \pmb{A} \pmb{h}_1(\pmb{x}(t),t) - 2\pmb{h}_2^{\top}(\pmb{x}(t),t)\pmb{A}\pmb{h}_1(\pmb{x}(t),t)] + \text{constant}.
> $$
> Since for any $\pmb{h}, \bar{\pmb{h}}, \pmb{B}$, we have
> $$
> \begin{aligned}
> \mathbb{E}\_{t}
> \mathbb{E}\_{\pmb{x}(t)}
>     [\pmb{h}^{\top}(\pmb{x}(t),t) \pmb{B} \bar{\pmb{h}}(\pmb{x}(t),t)]
> &= \mathbb{E}\_{t} \mathbb{E}\_{\pmb{x}(t)}
>     [\text{trace}(\pmb{B} \bar{\pmb{h}}(\pmb{x}(t),t)\pmb{h}^{\top}(\pmb{x}(t),t))] \\\\
> &= \text{trace}(\pmb{B} \mathbb{E}\_{t}
> \mathbb{E}\_{\pmb{x}(t)}
>     [\bar{\pmb{h}}(\pmb{x}(t),t)\pmb{h}^{\top}(\pmb{x}(t),t)]),
> \end{aligned}
> $$
> we further obtain
> $$
> \tilde{\mathcal{L}} (\pmb{\theta};\lambda(\cdot))
> = \text{trace}(\pmb{A}^{\top}\pmb{A} \pmb{B}_1)
> -2 \text{trace}(\pmb{A} \pmb{B}_2)  + \text{constant},
> $$
> where $\pmb{B}_1:=\mathbb{E}\_{t}
> \mathbb{E}\_{\pmb{x}(t)}
>     [\pmb{h}_1(\pmb{x}(t),t)\pmb{h}_1^{\top}(\pmb{x}(t),t)]$ and $\pmb{B}_2:=\mathbb{E}\_{t}
> \mathbb{E}\_{\pmb{x}(t)}
>     [\pmb{h}_1(\pmb{x}(t),t)\pmb{h}_2^{\top}(\pmb{x}(t),t)]$. Here, $\pmb{B}_1$ is a positive semi-definite matrix, since $\pmb{v}^{\top}\pmb{B}_1\pmb{v}=\mathbb{E}\_{t}
> \mathbb{E}\_{\pmb{x}(t)}
>     [(\pmb{v}^{\top}\pmb{h}_1(\pmb{x}(t),t))^2]\ge 0$ for any $\pmb{v}$. Notice that for any $\pmb{A}, \pmb{B}$,
> $$
> \begin{aligned}
> \text{trace}(\pmb{A}^{\top}\pmb{A} \pmb{B})
> &=\text{trace}(\pmb{A} \pmb{B}\pmb{A}^{\top})
> =\sum\_{i,j}\pmb{B}\_{ij}(\pmb{A}\_{:,j})^{\top}\pmb{A}\_{:,i}
> =\text{vec}(\pmb{A})^{\top} (\pmb{B} \otimes \pmb{I}) \text{vec}(\pmb{A}), \\\\
> \text{trace}(\pmb{A} \pmb{B})
> &= \sum\_{j} (\pmb{A}\_{:,j})^{\top}(\pmb{B}^{\top})\_{:,j}
> =\text{vec}(\pmb{A})^{\top} \text{vec}(\pmb{B}^{\top}),
> \end{aligned}
> $$
> where $\otimes$ denotes the Kronecker product. Hence
> $$
> \tilde{\mathcal{L}} (\pmb{\theta};\lambda(\cdot))
> = \text{vec}(\pmb{A})^{\top} (\pmb{B}_1 \otimes \pmb{I}) \text{vec}(\pmb{A}) - 2 \text{vec}(\pmb{B}_2^{\top})^{\top}\text{vec}(\pmb{A}) + \text{constant}.
> $$
> It is straightforward to show that the eigenvalues of $\pmb{B}_1 \otimes \pmb{I}$ are the same as $\pmb{B}_1$ but with multiplicity (see Lemma 0 below), implying that $\pmb{B}_1 \otimes \pmb{I}$ is also positive semi-definite. Therefore,
> $$
> \nabla\_{\pmb{\theta}}^2 \tilde{\mathcal{L}} (\pmb{\theta};\lambda(\cdot))
> = \nabla\_{\text{vec}(\pmb{A})}^2 \tilde{\mathcal{L}} (\pmb{\theta};\lambda(\cdot))
> =2(\pmb{B}_1 \otimes \pmb{I})
> $$
> is positive semi-definite, i.e., the loss is convex to trainable parameters.
>
> **Lemma 0**: Let $A\in \mathbb R^{n\times n}$, $B\in \mathbb R^{m\times m}$ have the eigenvalues $\\{\nu_i\\}_{i=1}^n$, $\\{\mu_j\\}\_{j=1}^m$, respectively. Then, the eigenvalues of $A \otimes B$ are $\nu_i\mu_j$, $i=1,\cdots,n$, $j=1,\cdots,m$.
>
> *Proof.* By Jordan–Chevalley decomposition, there exist invertible matrices $P, Q$ such that $A=P\Lambda P^{-1}$, $B=Q\Delta Q^{-1}$, where $\Lambda, \Delta$ are upper triangular matrices. Therefore,
> $$
> \begin{aligned}
> A \otimes B
> = (P\Lambda P^{-1}) \otimes (Q\Delta Q^{-1})
> = (P \otimes Q) (\Lambda \otimes \Delta) (P^{-1} \otimes Q^{-1})
> = (P \otimes Q) (\Lambda \otimes \Delta) (P \otimes Q)^{-1}.
> \end{aligned}
> $$
> That is, $A \otimes B$ and $\Lambda \otimes \Delta$ are similar. Notice that $\Lambda \otimes \Delta$ is still an upper triangular matrix with diagonal elements $\nu_i\mu_j$, $i=1,\cdots,n$, $j=1,\cdots,m$, we complete the proof.
>
> ---
>
> > **Weakness 2**: Definitions of notations and theorem statement.
>
> **A2**: To clarify, we have added a figure (refer to **Figure 5** in the **pdf attachment**) to illustrate the problem formulation and important notations, as is also suggested by **Reviewer ab6x** (see **Q1** and **A1** there). We hope this plot will further improve the readability. We will also add the omitted conditions in Theorem 1 back from the appendix in the updated version.
>
> ---
>
> > **Question 1**: Convexity.
>
> **A3**: As is discussed in **A1**, the loss function is a convex function w.r.t. trainable parameters ($\text{vec}(\pmb{A})$). We do not need an additional assumption on convexity.
>
> ---
>
> > **Question 2**: Trainable parameters.
>
> **A4**: Only the outer layer is trainable in the setting of random feature models. We will clarify and emphasize this point when introducing score networks in the updated version.
>
> ---
>
> > **Question 3**: Convexity with different target distributions.
>
> **A5**: As is discussed in **A1**, the loss function is a quadratic and hence convex function w.r.t. trainable parameters ($\text{vec}(\pmb{A})$), and this fact holds for any given target score function, including the Gaussian mixture.
>
> ---

---

> > ### Comment · Reviewer_4Zu3 · 2023-08-18
> >
> > Thank you the explanation. I realize that the convexity was due to the way the loss function is parametrized. However, I think a separate lemma and some brief explanation on this point should be added to the paper. As I have said in my previous comments, the writing stands to be improved and I don't think the paper is completely ready for publications at this point.
> > I raise my overall score to 4 and reduce my confidence level to 3.

---

> > > ### Author Response · Authors · 2023-08-19
> > > **Specific suggestions**
> > >
> > > Thanks for your comment. We are delighted to see that your (original) major concern on convexity has been resolved. We agree with your point to add a separate lemma and some brief explanation. Since all the contents including the explanation and proofs have been provided in the above response (refer to **A1**), we are ready to place them in the appendix before original Lemma 3 in the updated version.
> > >
> > > On the writing concern, as is shown in the above response (refer to **A2**), we have added a "formulation" figure (refer to **Figure 5** in the **pdf attachment**) to illustrate the setup and key notations, including
> > >
> > > - basic elements to perform a fundamental generalization analysis: the hypothesis space (diffusion process + random feature score network), concept space (different types of target distributions), loss objective (time-dependent score matching) and training algorithm (gradient flow);
> > >
> > > - mentioned (important) notations, such as the SDE time $t$ and its maximum $T$, training time $\tau$, and terminal state $p_T \approx \pi$ (a known prior).
> > >
> > > We feel that Figure 5 is clear enough to cover the details raised in your previous comments on the writing aspect. We would appreciate it if you can further provide more *specific* suggestions on the writing.

---

> > > > ### Comment · Reviewer_4Zu3 · 2023-08-20
> > > >
> > > > Regarding the writing, I did not mean to comment on the content provided by your rebuttal, but to repeat my assessment of the submitted paper. I apologize for causing confusion. This year, there are no options to submit a revision in the discussion phase, and rule 3 of the discussion phase says that the text in the rebuttal's attached PDF shouldn't be read. For context, the authors attached a pdf containing a figure consisting fully of text that explains some of the technical terms I mention. I don't think putting text inside a figure is a good-faith way of circumventing this rule. If this is indeed allowed, everyone can include a figure containing the scanned of a revised paper.

---

> > > > > ### Author Response · Authors · 2023-08-21
> > > > > **Further response**
> > > > >
> > > > > Thank you for the feedback. We are more than happy to improve the paper writing with all the reviewers' constructive suggestions. However, we respectfully disagree with the above judgement.
> > > > >
> > > > > **First, we did not violate the policy.** We submitted a 1-page PDF attachment containing 5 figures. There are some words serving as legends and captions, which are for the convenience of understanding the contents of figures. This is neither "full-of-texts" nor "a figure containing the scan of a revised paper". We believe that the reviewer's judgement is not valid.
> > > > >
> > > > > **Second, Figure 5 as an illustration of the problem formulation is suggested by Reviewer ab6x.** We believe that it is a good idea to improve the clarity of the presentation.
> > > > >
> > > > > Finally, we want to emphasize that the 1-page pdf attachment is allowed and used to show that the required materials are *now ready*. We can include all of them immediately in the  revised version.

---

### Official Review · Reviewer_ab6x · 2023-07-18

**Soundness:** 3 good
**Presentation:** 3 good
**Contribution:** 3 good
**Rating:** 6
**Confidence:** 3

**Summary:**


 This paper proves generalization bounds for diffusion models. With specific stopping, they show that the generalization error goes to zero, with a specific upper bound on the rate that scales polynomially with the sample size and the model capacity. Theorem 1 is the main convergence result, while in Theorem 2 they extend their results to the data-dependent setting. The paper is concluded with some supporting experiments.




**Strengths:**


- well-motivated problem and novel formulation
- nice theoretical results
- enough citations to prior works


**Weaknesses:**


- while it's generally well-written, it could've been better (e.g., a figure could've been used for problem formulation to make it better readable)



**Questions:**


This is a nice theoretical result. Here are some comments/questions.


- Equation 3: there are two $dt$'s there, is that right, or it's a typo? Please make it clear


- Theorem 1: do you need to know that RKHS to run the algorithm? Or do you just use it in the proofs? This is not that much clear from the context


- it is stressed that the upper bound on the generalization is dimension-free, while there is dependence on $d$ in the bound (Equation 18)

---

> ### Author Rebuttal · Authors · 2023-08-08
>
> # Response to Reviewer ab6x
>
> Thank you for your comprehensive review and your valuable feedback to help us improve the paper. We detail our response below, and please kindly let us know whether our response addresses your concerns.
>
> ---
>
> > **Q1**: While it's generally well-written, it could've been better (e.g., a figure could've been used for problem formulation to make it better readable).
>
> **A1**: This is a useful suggestion to further improve the readability. We have followed this suggestion to plot a figure (refer to **Figure 5** in the **pdf attachment**) to illustrate our problem formulation, including the diffusion process, (score matching) loss objectives, the random feature score network model, *different* target distributions, the gradient flow training dynamics and other important notations. We will also include this figure in the final version.
>
> ---
>
> > **Q2**: Equation 3: there are two $dt$'s there, is that right, or it's a typo? Please make it clear.
>
> **A2**: Yes, it is a typo. We have fixed it in the updated version.
>
> ---
>
> > **Q3**: Theorem 1: do you need to know that RKHS to run the algorithm? Or do you just use it in the proofs? This is not that much clear from the context.
>
> **A3**: The RKHS norm $\||\cdot\||_\mathcal{H}$ in Theorem 1 is the shorthand of $\||\cdot\||\_{\mathcal{H}\_{k\_{\rho\_0}}}$ , which is defined in lines 163-167. In short, $\rho_0$ is the distribution to initialize inner parameters $(\pmb{w},\pmb{u})$, $k\_{\rho\_0}$ is the induced kernel, and $\mathcal{H}\_{k\_{\rho\_0}}$ is the induced RKHS. We certainly use it in the proofs, and notice that the RKHS norm is just a weighted $L^2$-norm averaged on $\rho_0$ (see the definition in lines 166-167), and hence can be easily estimated by the Monte Carlo method (e.g. an empirical mean) when specifying $\rho_0$.
>
> ---
>
> > **Q4**: It is stressed that the upper bound on the generalization is dimension-free, while there is dependence on $d$ in the bound (Equation 18).
>
> **A4**: By saying "dimension-independent", we mean escaping from the curse of dimensionality (CoD), i.e., the upper bound does not exponentially depend on the data dimension. We will remove the term "dimension-independent" to avoid misunderstanding in the final version.
>
> ---

---

> > ### Comment · Reviewer_ab6x · 2023-08-14
> > **Response**
> >
> > I acknowledge the response provided by the authors, specifically the idea of adding new figures. Please also add the clarifications suggested in my comments to the new version of the paper (as promised here). I decided to keep my score unchanged.

---

> > > ### Author Response · Authors · 2023-08-14
> > > **Thank you**
> > >
> > > Thank you for acknowledging our response and considering our proposed changes, including the addition of new figures. We appreciate your feedback and your valuable comments. We will incorporate all the suggestions you provided into the new version of the paper. Thank you once again for your time and insightful reviews.

---

### Author Rebuttal · Authors · 2023-08-08

# Summary of new results (as required)

We sincerely appreciate all reviewers for their insightful and constructive feedback. Besides answering questions in detail to address all of the reviewers’ comments, we want to summarize and highlight new key results as follows, and all of the results are included in the **pdf attachment** and will be added in the final version.

1. We provide a new simulation result (refer to **Figure 1** in the **pdf attachment**) on the MNIST dataset with modeling the score function as the commonly-used U-net architecture, which suggests that the adverse effect of modes shift on the performance of diffusion models may appear in general. The setup procedure is as follows: (i) construct datasets from MNIST with different distances between modes; (ii) train diffusion models and evaluate respective performance on different scales of modes distances.

    - Concretely, let $\mathcal{D}$ denote the whole MNIST dataset, we first perform a $K$-Means clustering on $\mathcal{D}$ to get $\mathcal{D}=\bigcup_{k=1}^K \mathcal{D}_k$, and $\bar{\pmb{x}}_k$ as the center of $\mathcal{D}_k$, $k=1,\cdots,K$. Let $(i^*,j^*):=\arg\max\_{i \ne j} \||\bar{\pmb{x}}_i-\bar{\pmb{x}}_j\||$, and $\mathcal{D}\_{\text{farthest}}:=\mathcal{D}\_{i^*} \cup \mathcal{D}\_{j^*}$ ($\mathcal{D}\_{\text{nearest}}$ is similarly defined by corresponding $\arg\min$ indices). By randomly selecting the same number of data samples and using the same (hyper-parameters) configuration, we train two separate diffusion models on $\mathcal{D}\_{\text{farthest}}$ and $\mathcal{D}\_{\text{nearest}}$, respectively, and then perform inference (sampling). The training loss curves are shown in **Figure 1 (a)**, and the sampling results are shown in **Figure 1 (b)** for $\mathcal{D}\_{\text{farthest}}$ and **Figure 1 (c)** for $\mathcal{D}\_{\text{nearest}}$. One can observe a clear performance gap: the diffusion model trained on  $\mathcal{D}\_{\text{farthest}}$ appears a higher learning loss and worse sampling quality compared to those of $\mathcal{D}\_{\text{nearest}}$.

2. We reproduce original Figure 2 (studying the KL divergence dynamics) in the current paper by using (S)GD for training with many repetitions (refer to **Figure 2** in the **pdf attachment**), as is suggested by Reviewer uaj6. It is shown that the original experimental results are consistent with the theory and over multiple runs.

3. We reproduce original Figure 3 and Figure 4 (studying the modes shift effect) in the current paper by using (S)GD for training (refer to **Figure 3** and **Figure 4** in the **pdf attachment**), as is suggested by Reviewer uaj6. It is shown that the original experimental results are consistent with the theory.

4. We add a figure to illustrate our problem
formulation (refer to **Figure 5** in the **pdf attachment**), as is suggested by Reviewer ab6x.

---

### Decision · Program_Chairs · 2023-09-21

**Decision:**

Accept (poster)

**Comment:**

The paper presents some interesting novel theoretical results on the KL divergence between the true distribution and the learned distribution of scored based diffusion models, which might inspire an avenue of future work on the generalization of diffusion models.

The author responses to the reviewers comments satisfactorily addresses key concerns. We therefore urge the authors to incorporate the additional material and clarifying points provided in their feedback.

In particular it would be important to add a remark to prevent potential confusion on convexity, incorporate the novel experiment results and figure depicting the overall formulation, and include the discussion on the error bounds.